# Electron shuttling promotes denitrification and mitigates nitrous oxide emissions in lakes

Kang Song [1,2,3,4,8] ✉, Yanlin Xiao[1,5,8], Yuren Wang[1,4], Min Deng [1,3] ✉, Shuni Zhou[1,4], Yongxia Huang[1], Senbati Yeerken[6], Lu Li[1,2] & Fengchang Wu [7] ✉

Eutrophication is an emerging global issue that is becoming increasingly severe due to the rising nutrient inputs and limited availability of electron donors for nitrogen removal. In sediments where redox conditions fluctuate dramatically, extracellular electron transfer (EET) critically supports microbial metabolism. However, the biogeochemical significance of EET-coupled denitrification and its EET mechanisms remain unclear. Here, through field investigations and laboratory $^{15}N$ isotope experiments, we reveal that humic substance (HS)-mediated electron shuttling significantly enhances denitrification primarily by stimulating bacterial outer membrane *c*-type cytochrome. Specifically, EET mitigates the emission of greenhouse gas nitrous oxide by enriching *nosZII*-type reducers. Notably, the efficacy of exogenous HS amendment attenuates in sediment with high native HS concentration. Metagenomic binning further reveals multiple cytochromes forming a complete EET-coupled denitrification electron transport chain. Our findings elucidate the microbial mechanisms underlying electron shuttling-driven denitrification in lakes, thereby expanding the understanding of biogeochemical cycles.

Eutrophication has emerged as a significant global issue, primarily driven by increasing anthropogenic nutrient inputs, particularly the disproportionate rise in nitrogen from industrial and domestic wastewater, chemical fertilizers, and urban runoff[1,2]. This process often results in the proliferation of harmful algal blooms, which in turn lead to hypoxic conditions, the death of aquatic organisms, and even human health risks[1]. Presently, more than 63% of the world's large inland lakes are affected by eutrophication due to excessive nitrate concentrations[3]. Denitrification, the key biological mechanism for permanently removing bioavailable nitrate from ecosystems, is

essential for mitigating eutrophication[4]. However, incomplete denitrification also contributes significantly to the accumulation of toxic nitrite and the emission of nitrous oxide ($N_2O$), a potent greenhouse gas with a global warming potential ~310 times that of carbon dioxide[5–7]. Insufficient electron supply has been identified as a major factor constraining denitrification efficiency and elevated $N_2O$ emissions[3].

The extracellular electron transfer (EET) process provides supplementary electrons for biogeochemical cycling, playing a crucial role in sedimentary and aquatic systems. In soils, long-distance (> 10 cm)

[1]State Key Laboratory of Lake and Watershed Science for Water Security, Institute of Hydrobiology, Chinese Academy of Sciences, Wuhan, China. [2]Southern Marine Science and Engineering Guangdong Laboratory (Guangzhou), Guangzhou, China. [3]National-Regional Joint Engineering Research Center for Soil Pollution Control and Remediation in South China, Guangdong Key Laboratory of Integrated Agro-environmental Pollution Control and Management, Institute of Eco-environmental and Soil Sciences, Guangdong Academy of Sciences, Guangzhou, China. [4]University of Chinese Academy of Sciences, Beijing, China. [5]School of Environmental Studies, China University of Geosciences, Wuhan, Hubei, China. [6]College of Ecology and Environment, Xinjiang University, Urumqi, China. [7]State Key Laboratory of Environmental Criteria and Risk Assessment, Chinese Research Academy of Environmental Sciences, Beijing, China. [8]These authors contributed equally: Kang Song, Yanlin Xiao. ✉e-mail: sk@ihb.ac.cn; dengmin@ihb.ac.cn; wufengchang@vip.skleg.cn

EET facilitates an estimated electron flux of up to 6.73 μmol e⁻ cm⁻² day⁻¹ [8]. Similarly, in sediments, cable bacteria, which are filamentous microorganisms capable of conducting electricity over long distances, enable electron transfer across distances of 1–2 cm in the suboxic zone[9,10]. These bacteria link the oxidation of sulfides and organic matter in deeper sediment layers to 30%–42% of oxygen consumption at the surface[9,10]. These long-distance EET processes represent a significant but underappreciated source of remote electrons. Microbes often utilize electron shuttles to facilitate long-distance, indirect interspecies electron transfer. In EET mediated by electron shuttles, electrons produced through intracellular metabolic processes are transported to extracellular electron acceptors via redox-active electron shuttles[11]. Among these electron shuttles, humic substances (HS) stand out as the most abundant natural electron shuttles, forming 50%–80% of natural organic matter in water, soil, and sediments[12–14]. These organic polymers, derived from the decomposition and recombination of plant and microbial residues, play a vital role in supporting EET. Anaerobic organisms in aquatic sediments can use reduced HS to convert nitrate into nitrogen gas ($N_2$)[15]. Furthermore, the addition of HS to wetland sediments has been shown to enhance the reduction of $N_2O$, thereby mitigating greenhouse gas emissions[16]. However, the microbial coupling mechanisms between EET bacteria and denitrifying bacteria remain poorly understood, leaving a critical knowledge gap that requires further investigation.

The EET process involves three distinct functional proteins (Supplementary Fig. S1). Outer membrane cytochrome B (OmcB) is a multi-heme outer membrane cytochrome that transfers electrons from the intracellular environment to the outer membrane surface[17]. Similarly, outer membrane cytochrome S (OmcS), another multi-heme cytochrome, is located on the surface of conductive pili[18,19]. Pilin A (PilA) serves as the structural protein at the core of extracellular conductive pili, forming a physical pathway for electron conduction[20]. At present, the role of HS in influencing various EET processes and in situ denitrification rates remains insufficiently explored. Furthermore, while considerable progress has been made in elucidating how EET bacteria transfer electrons to extracellular electron acceptors, the microbial pathways enabling denitrifiers to utilize extracellular electrons remain poorly understood.

The EET-coupled denitrification process is a fundamental yet poorly understood mechanism in biogeochemical cycling. Here, we employed mass spectrometry, X-ray photoelectron spectroscopy (XPS), water chemistry analysis, bioelectrochemical techniques, and molecular biology approaches to investigate this process comprehensively. We confirmed that outer membrane cytochromes, in conjunction with electron shuttles, play a dominant role in the EET-coupled denitrification pathway in sediments. Notably, the efficacy of exogenous electron shuttle amendments declines significantly in sediments where native HS concentrations exceed 50 mg g⁻¹ sediment. Additionally, metagenomic binning revealed detailed electron transfer pathways from EET bacteria to HS and subsequently to denitrifiers, highlighting the extracellular electron uptake pathways utilized by denitrifiers. These findings offer valuable guidance for urban managers to design and implement targeted strategies aimed at mitigating environmental impacts effectively.

## Results and discussion

### Characteristics of HS determine electron transfer capacity in sediment

The concentration and composition of dissolved HS in sediment are presented in Fig. 1. By establishing sampling sites across different land-use areas and submerged plant zones, we successfully obtained a wide range of HS concentrations (9.5 ~ 230.1 mg g⁻¹ sediment) (Fig. 1b; and Supplementary Table S1). This range encompasses the previously reported HS concentrations in shallow, eutrophic lakes (14.5 ~ 78.5 mg g⁻¹ sediment)[21]. HA was the predominant form of HS, comprising

92.8 ± 9.4% of the total dissolved HS (Fig. 1a), which was consist with the HS composition in other lakes[21]. As dominant refractory organic compounds in sediment, HS also serve as natural electron shuttles[14,22]. Consequently, the electron transfer capacity of in situ sediments was assessed using maximum power density (MPD) measured by microbial fuel cells (MFC) (Supplementary Fig. S2)[23]. The MPD values spanned a wide range (0.05 ~ 76.65 μW m⁻²), comparable to those measured in aquaculture pond sediment MFC (52.2 ~ 99.5 μW m⁻²)[24] and plant MFC (0.28 ~ 37.0 μW m⁻²)[25,26]. However, these values were significantly lower than those reported for lake Taihu sediment MFC (3150 ~ 5700 μW m⁻²)[27–29], which were attribute to the formation of mature biofilm on anode surfaces after 15 to 60 days of cultivation in prior studies. Thus, the MPD measured in the present study (within 1 h after sediment addition) better reflects in situ characteristics. Notably, MPD exhibited a significant linear increase with HS concentration ($R^2 = 0.7355$, $p = 0.0004$) (Fig. 1, and Supplementary Fig. S3). These results indicate that natural electron shuttles such as dissolved HS can substantially enhance electron transfer capacity, similar to the effects observed with artificial electron shuttles[23].

### Enhanced electron transfer capacity improves lake denitrification performance while reduces $N_2O$ emissions

The physicochemical characteristics of overlying water and sediment are shown in Supplementary Table S2. Inorganic nitrogen concentrations in the water column aligned with prior Lake Taihu studies, while sediment inorganic nitrogen concentrations and dissolved $N_2$ concentration exceeded reported ranges[30–33]. The relationships between environmental factors and in situ denitrification performance are shown in Supplementary Fig. S4. Spearman correlation analysis revealed that water-column dissolved oxygen and pH, as well as sediment $NO_3^-$-N, $NH_4^+$-N, MPD, and HA, exhibited significant positive correlations with dissolved $N_2$ or $\Delta N_2$ concentrations ($p < 0.05$) (Fig. 1d; and Supplementary Fig. S4). Linear regression analysis further confirmed that MPD exhibited the strongest explanatory power for variations in dissolved $N_2$ and $\Delta N_2$ concentrations ($R^2 \geq 0.7926$) (Supplementary Fig. S5). Furthermore, stepwise multiple linear regression analysis identified MPD as the only significant predictor for both dissolved $N_2$ ($p < 0.05$) and $\Delta N_2$ ($p < 0.01$) concentrations (Supplementary Table S3–S4). The standardized coefficient of MPD was significantly higher than those of other environmental factors (Fig. 1c). Additionally, including predictors other than MPD did not significantly improve the model's goodness-of-fit for dissolved $N_2$ or $\Delta N_2$ concentrations ($p > 0.05$) (Supplementary Table S5-S6). Hence, MPD was the key environmental factor driving $N_2$ emissions. Elevated $\Delta N_2$ concentrations suggest higher complete denitrification rates at sites with increased MPD[33,34]. Notably, MPD is negatively correlated with all measured $N_2O$ emission characteristics, especially with $N_2O/(N_2O + N_2)$ ($p < 0.05$) (Fig. 1d). Previous laboratory incubation experiments using isotope analysis have demonstrated that the addition of HS or other exogenous electron shuttles to paddy soil and wetland sediments enhances $N_2O$ reduction[16,35]. Similarly, biochar, an electron shuttle, has been widely reported to reduce $N_2O$ emissions from denitrification in agricultural soils, such as paddy fields, acidic tea gardens, and vegetable fields, by 35%-99.9%[36–38]. Overall, an extra electron supply from EET facilitates complete denitrification, particularly enhancing the $N_2O$ reduction in sediment.

### Electron transfer-related functional groups of HS

Given that MPD significantly affects in situ denitrification potential (Fig. 1c), the samples were categorized into three representative groups to analyze the functional group characteristics of HS under varying MPD conditions (L-MPD: 4.1 ± 6.1 μW m⁻², M-MPD: 44.0 ± 3.5 μW m⁻², and H-MPD: 73.3 ± 3.6 μW m⁻²) (Supplementary Table S7). These sampling sites exhibited different dissolved $N_2$ and $\Delta N_2$ concentrations (Supplementary Table S7). Excitation-emission matrix (EEM) fluorescence

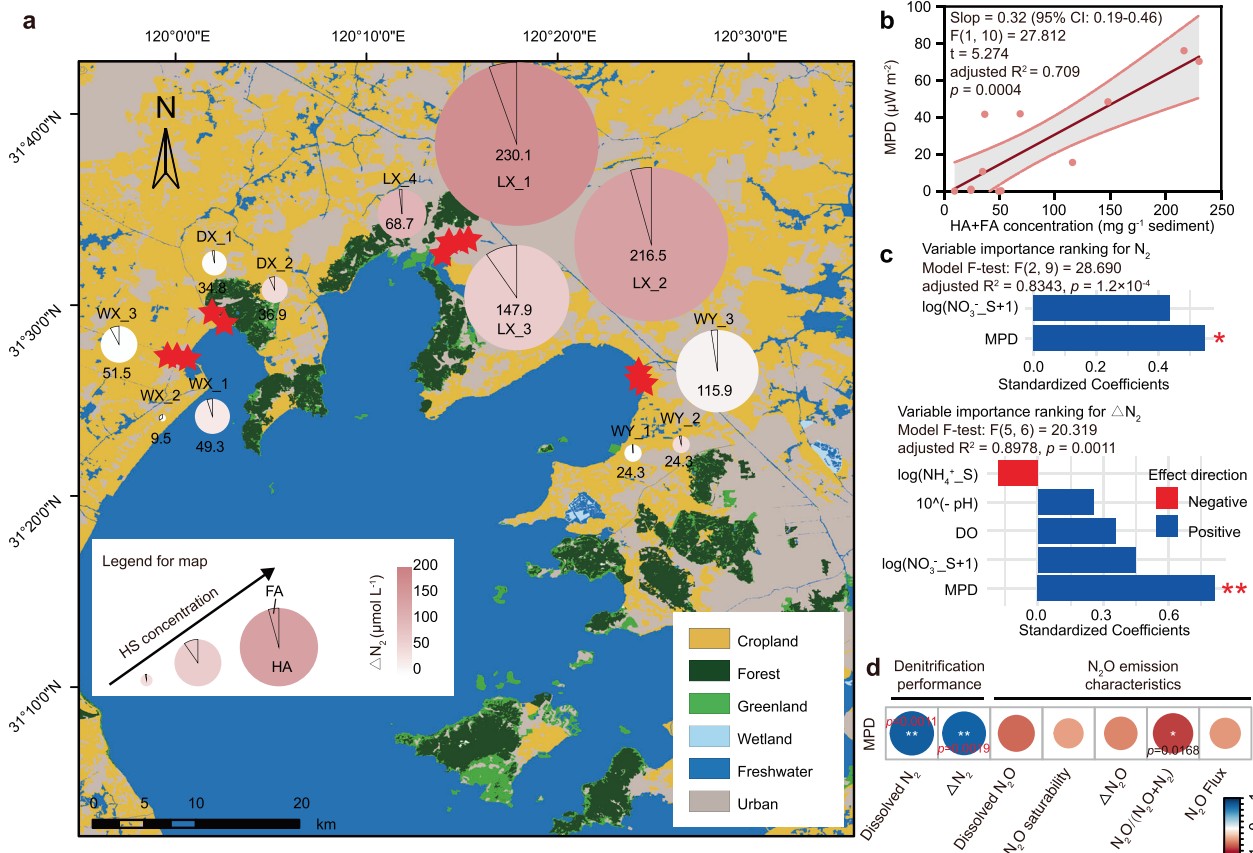

**Fig. 1 | Sampling sites, characteristics of electron shuttles, and in situ nitrogen removal performance. a** Distribution of sampling sites (red stars, $n = 12$ biological replicates). The pie charts indicate the concentrations and proportions of dissolved electron shuttles, i.e., humic acid (HA) and fulvic acid (FA). The size of each pie chart is proportional to the concentration of humic substances (HS), while the color represents the excess $N_2$ ($\Delta N_2$) concentration at the respective sites. **b** Simple linear regression analyses (two-sided) revealed a significant relationship between the concentrations of HS (HA + FA) and the electron transfer capacity, represented by maximum power density (MPD). The dark-red solid line denotes the linear regression fit. Light gray shading with red borders indicates 95% confidence intervals (CI). **c** Standardized coefficients of environmental factors derived from multiple linear regression model predicting dissolved $N_2$ concentration and $\Delta N_2$ concentration. Significance of each standardized coefficient was assessed using two-sided t-tests against the null hypothesis that the true coefficient is zero, derived from the regression model. * and ** indicate $p$ value = 0.0227 (standardized coefficient = 0.547 (95% CI: 0.096, 0.998), t = 2.745, Cohen's d = 1.830) and $p$ value = 0.0039 (standardized coefficient = 0.812 (95% CI: 0.375, 1.249), t = 4.546, Cohen's d = 3.712), respectively (two-sided $t$-test). **d** Spearman correlations between the MPD of flesh sediments and the in situ emission characteristics of $N_2$ and $N_2O$. Blue and red circles denote positive and negative correlations, respectively, with color saturation representing correlation strength. $n = 12$ biological replicates. Statistical significance was defined as a $p$ value < 0.05.

spectroscopy analysis confirmed that the extracted HS were HA and FA analogs[39] (Supplementary Fig. S6). Next, XPS analysis of functional groups in HA and FA from these representative MPD sediments revealed that the high-resolution C 1 $s$ spectra contain C-C, C-OH, C = O, and -COOH groups as the predominant carbon-based compounds (Fig. 2)[40]. Among these, phenolic C-OH groups primarily contribute to the electron-donating capacity (EDC), while quinone-type C = O groups support the electron-accepting capacity (EAC)[40,41]. In sediments, the continuous microbial transformation of C-OH and C = O groups determines the electron exchange capacity of HS, which is governed by the total atomic percentages of C-OH and C = O functional groups[41]. In this study, the total atomic percentages of C-OH and C = O functional groups in both HA and FA increased with the MPD (Fig. 2). Overall, the increased concentrations of HA and FA, combined with the rise in electron exchange-related functional groups, could be beneficial for enhancing MPD in sediments.

**Key EET processes affecting denitrification and $N_2O$ emissions**

The relationships between EET functional genes and denitrifying functional genes are shown in Fig. 3 and Supplementary Fig. S7. In this study, the absolute abundance of all EET functional genes exhibited positive correlations with denitrifying functional genes (i.e., *nirS* and *nirK*), particularly *omcB* ($R^2 \geq 0.6663$, $p < 0.05$) and *omcS* ($R^2 \geq 0.6818$, $p \leq 0.01$) (Supplementary Fig. S7). However, the absolute abundances of *nirS*, *omcB*, and *omcS* genes showed significant positive correlations with total bacterial abundance, characterized by high goodness-of-fit ($R^2 \geq 0.5304$, $p < 0.05$) (Supplementary Fig. S8). This suggests that the significant positive correlation between *nirS* and EET genes (*omcB* and *omcS*) (Fig. 3a) might stem from overall bacterial community growth in sediment. Hence, the linear correlation between 16S rRNA gene-standardized functional genes were further performed (Fig. 3). The results indicated all standardized EET genes were significantly positively correlated with denitrifying genes (Fig. 3a). However, only *omcB* showed a significant positive correlation with dissolved $N_2$ and $\Delta N_2$ concentrations ($R^2 \geq 0.6988$) (Fig. 3d). Additionally, only *omcB* demonstrated strong positive correlations with HA and FA concentrations ($R^2 \geq 0.5205$, $p < 0.05$) (Fig. 3c; and Supplementary Fig. S9). Furthermore, the addition of HS in sediment (HS_C group) significantly increased the expression level ratio (HS_C/Blank) of the *omcB* gene by 31.5 times, notably higher than the increases observed for *omcS* (2.0 times) and *pilA* (3.3 times) genes (Fig. 4e).

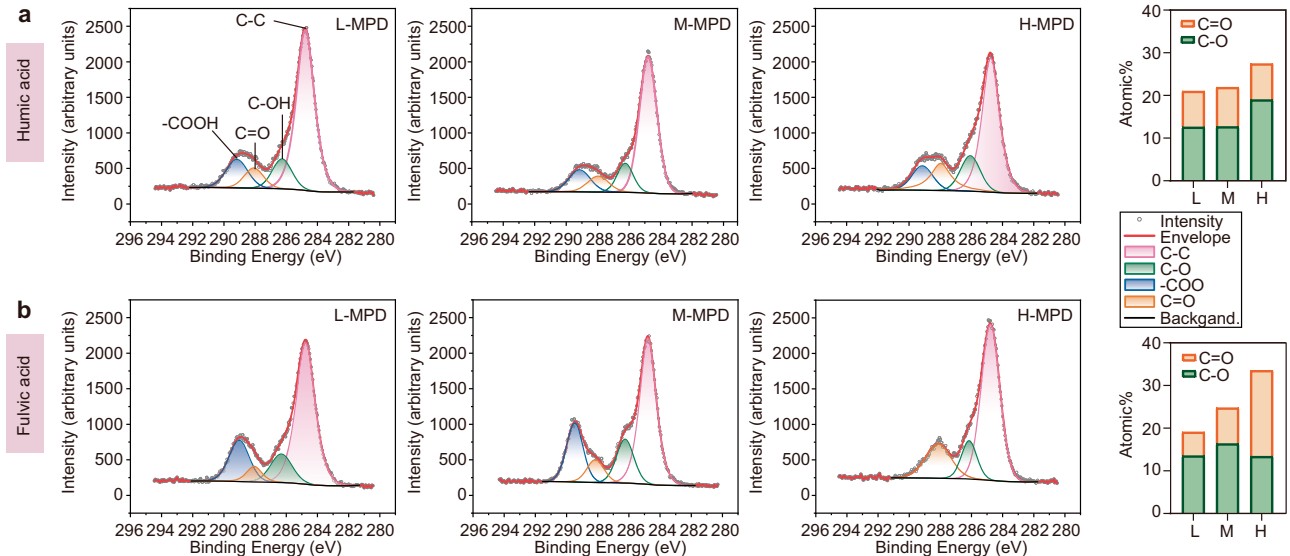

**Fig. 2 | The functional groups of extracted electron shuttles. a, b** X-ray photo-electron spectroscopy (XPS) measured C 1s spectrum and atomic% of humic acid (**a**) and fulvic acid (**b**) in representative L-MPD, M-MPD, and H-MPD samples. DX_1, LX_3, and LX_1 were randomly selected as representative L-MPD, M-MPD, and H-MPD samples, respectively (Supplementary Table S1). MPD, maximum power density.

Our results indicated that humic-type electron shuttles primarily enhance EET processes across the cell membrane pathway. The *omcB*, *omcS*, and *pilA* genes encode proteins essential for outer membrane c-type cytochrome[42], nanowire-forming cytochrome[43], and pilin protein[20], respectively (Supplementary Fig. S1). EET is facilitated by these pathways, which promote the activity of strictly anaerobic *nirS*-type denitrifiers[44] and facultatively anaerobic *nirK*-type denitrifiers[45]. However, only *omcB* exhibited a strong linear relationship ($R^2 > 0.52$, $p < 0.05$) with both HS and the denitrification end product ($N_2$). This correlation may be due to the larger contact area between the cell outer membrane and extracellular electron shuttles compared to nanowires or pilin. Overall, the outer membrane EET process regulated by *omcB* predominantly governs all types of denitrification-based nitrogen removal processes in sediments.

Furthermore, EET processes primarily enhance $N_2O$ reduction by stimulating *nosZII*-type $N_2O$ reducers. All EET functional genes showed significantly positive correlations with *nosZII* gene ($R^2 \geq 0.5204$, $p < 0.05$), while only the *pilA* gene correlated significantly with the *nosZI* gene ($R^2 = 0.6948$, $p = 0.0014$) (Fig. 3b). When comparing sediment gene expression between HS_C (HS-added) and Blank (no HS addition), the relative expression ratio (HS_C/Blank) of the *nosZII* gene (31.5-fold) was 4.1-fold higher than that of the *nosZI* gene (7.7-fold) (Fig. 4d). The *nosZI* and *nosZII* genes regulate the final step of denitrification and are the only known microbial pathways for reducing $N_2O$ to $N_2$[46]. The heightened abundance and expression of *nosZII* gene are particularly effective at mitigating $N_2O$ emissions, as *nosZII*-type reducers outcompete *nosZI*-type reducers in utilizing low $N_2O$ concentrations[46]. These findings suggested that EET can lower $N_2O$ emissions by enhancing $N_2O$ reduction, specifically via *nosZII*-type $N_2O$-reducers (Fig. 1d; Fig. 3b). This effect likely results from the increased electron availability, which favors complete denitrification and yields $N_2$ instead of $N_2O$ as the final product[47].

**HS addition enhanced EET and complete denitrification processes in sediment**

To confirm whether HS enhance denitrification performance and reduce $N_2O$ emissions by increasing EET capacity, we supplemented sediment samples with additional HS and monitored MPD, denitrification rate, $N_2O$ accumulated level, and functional gene expression

level (Fig. 4). Compared to the blank group (no additional HS), the HS-added group (HS_C) showed significantly elevated MPD ($p = 0.0025$) (Fig. 4a). Further analysis using $^{15}N$ isotope labeling confirmed that denitrification ($71.1 \pm 31.1\%$) was the dominant pathway for $N_2$ production, with HS addition enhanced denitrification rates by 2.6 times ($p = 0.0028$, Fig. 4b). These results align with prior studies showing that electron shuttles like HA accelerate electron transfer and boost denitrification performance in wastewater treatment systems[48–51]. Importantly, after 24 h incubation, HS addition reduced accumulated $N_2O$ concentration by 78.3% compared to the Blank group ($p = 0.0156$). Similar effects have been observed with FA, which suppressed $N_2O$ generation by 41.3% – 98.8% in *Paracoccus denitrificans*-mediated denitrification[52]. Together, these findings demonstrate that natural electron shuttles such as HS significantly enhance in situ denitrification in lake sediments.

Bacterial biomass exhibited no significant difference during the 24-h incubation (Supplementary Fig. S10). To characterize active microbial processes, we therefore further analyzed the effects of HS addition on functional gene expression (Fig. 4d–g). At the initial time point (0 h), relative expression levels of these functional genes showed no significantly differences (Supplementary Table S8). By 24 h, however, the ratios for all functional gene expression levels (HS_C/Blank) were notably higher than 1 (ranging from 4.6- to 396.1-fold), indicating that HS addition broadly enhanced gene expression associated with denitrification, $N_2O$ reduction, and EET (Fig. 4d–e). Notably, the high ratio of $N_2O$-reducing genes (*nosZI*+*nosZII*) to denitrifying genes (*nirS*+*nirK*), quantified as *nos/nir* (Fig. 4d), highlighted the preferential upregulation of $N_2O$ reduction pathways. These findings align with prior studies showing that FA upregulates denitrification genes, particularly *nosZ*, in the model denitrifier *Paracoccus denitrificans*[52]. Our results demonstrate that HS-enhanced EET preferentially improves $N_2O$ reduction activity, thereby accelerating denitrification rates while reducing $N_2O$ production.

Notably, across the tested HS concentration range (0-230 mg-HS $g^{-1}$ sediment), the enhanced ratios (HS_C/Blank) of denitrification rate decreased exponentially with increasing native HS concentrations ($R^2 = 0.6899$) (Fig. 4c). This trend aligns with the observation that L-MPD samples exhibited a 34.1-fold increase in MPD after HS addition, significantly higher than the 2.0-fold increase in all samples (Fig. 4a).

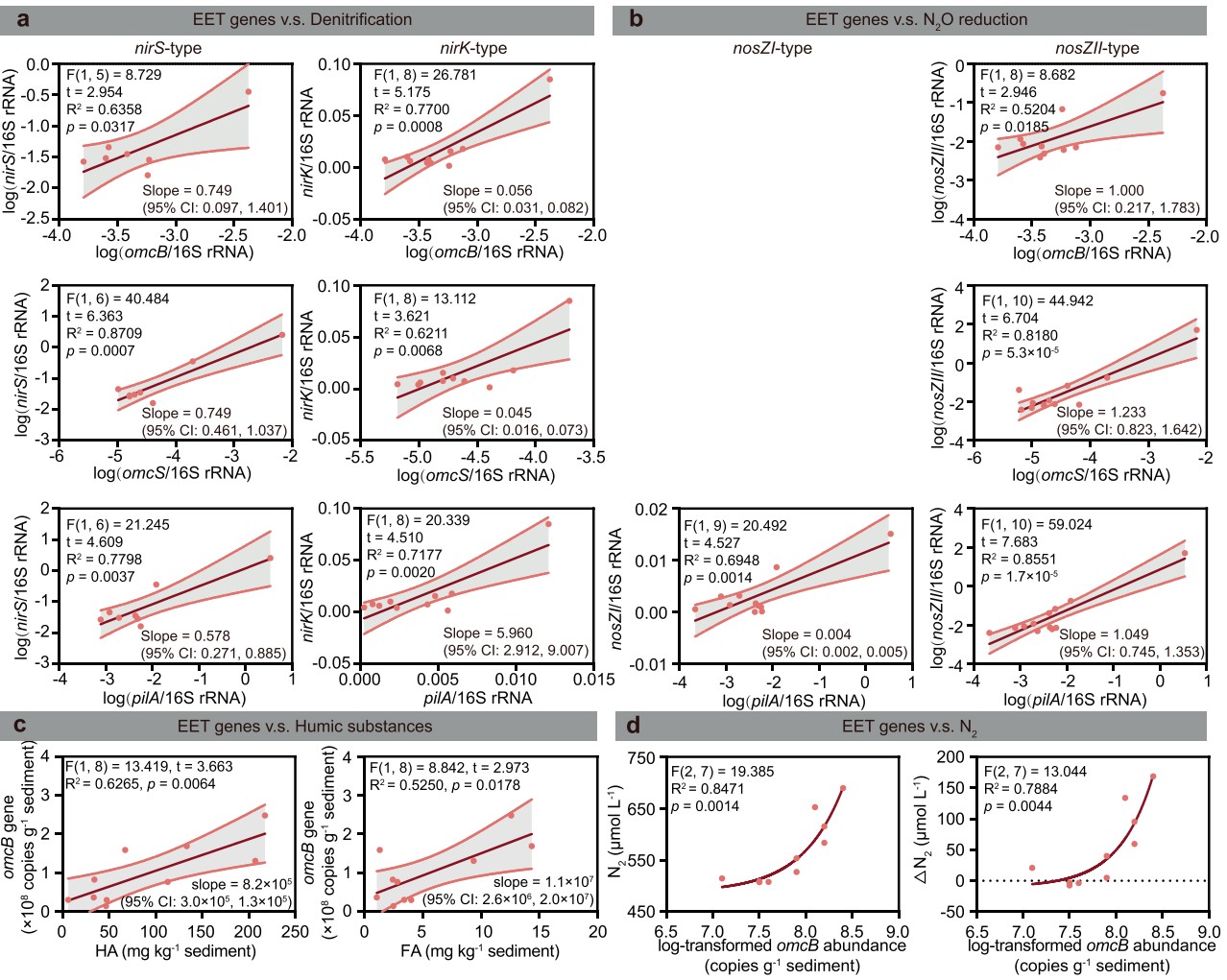

**Fig. 3 | The relationship between the extracellular electron transfer (EET) process and the nitrogen removal process. a** Two-sided simple linear regression between standardized EET genes and denitrifying genes. **b** Two-sided simple linear regression between standardized EET genes and $N_2O$ reducing genes. **c** Two-sided simple linear regression between humic substances and EET gene *omcB*. **d** Exponential correlation between the EET functional gene and dissolved $N_2$ characteristics (y = a*exp(b*x)+c). For parameter b, the one-sided t-test results are:

$t(7) = 2.105$, $p = 0.0366$ for the $N_2$ model; and $t(7) = 1.945$, $p = 0.0464$ for the $\Delta N_2$ model. All functional genes were standardized by 16S rRNA gene abundance. Dots represent individual sampling sites for $n = 12$ biological replicates. The dark-red solid lines denote the linear regression fits (**a–c**). The light gray shading with red borders indicates the 95% confidence intervals (CI) around the regression lines. Statistical significance was determined at $p < 0.05$.

Consistently, while HS addition broadly upregulated functional genes (Fig. 4d, e; and Supplementary Table S9), significant overexpression of *nirK*, *nosZI*, *nosZII*, and all EET genes occurred exclusively in L-MPD samples (Fig. 4f, g). These results demonstrate that the efficacy of exogenous electron shuttle amendment attenuates significantly in sediments with high native HS concentrations (> 50 mg-HS g⁻¹). Therefore, adding electron shuttles in low-HS environments can achieve optimal cost-effectiveness for nitrogen pollution remediation and $N_2O$ mitigation.

## Microbial community composition variation with MPD

The microbial community composition was analyzed using both 16S rRNA gene high-throughput sequencing and metagenomic sequencing (Fig. 5). Principal component analysis revealed distinct variations in microbial community composition that corresponded to MPD and HS concentrations (Fig. 5a). The dominant phyla identified were Actinobacteriota, Halobacterota, Chlorobiota, and Bacteroidota (Fig. 5b). Notably, Crenarchaeota (archaea) were predominated in low-MPD sediments, while

Bacteroidota was more abundant in high-MPD sediments (Fig. 5b). Organisms capable of using reduced HS as electron donors for nitrate reduction are commonly found in aquatic environments, with particularly high abundance in lake sediments[15]. In the present study, electroactive methanogens such as *Methanosaeta*, *Methanosarcina*, and *Methanolinea*[53,54], along with the denitrifying bacterium *Flavobacterium*[45], were dominant in high-MPD sediments (Fig. 5c). Additionally, at the species level, dominant microbes (> 0.2%) were compared across MPD using metagenomic sequencing (Fig. 5d). In high-MPD sediments, there was an increased abundance of electroactive iron-reducing bacterium *Ferribacterium limneticum*, hydrogenotrophic denitrifier *Quatrionicoccus australiensis*, and heterotrophic denitrifier *Acinetobacter sp. CS-2*[55,56]. Laboratory studies have shown that artificial electron shuttles enhance methanogenesis, iron reduction, and denitrification[53–55]. Consequently, we used metagenomic binning to further investigate the mechanisms by which natural electron shuttles mediate EET processes in lake sediments, specifically supporting denitrification.

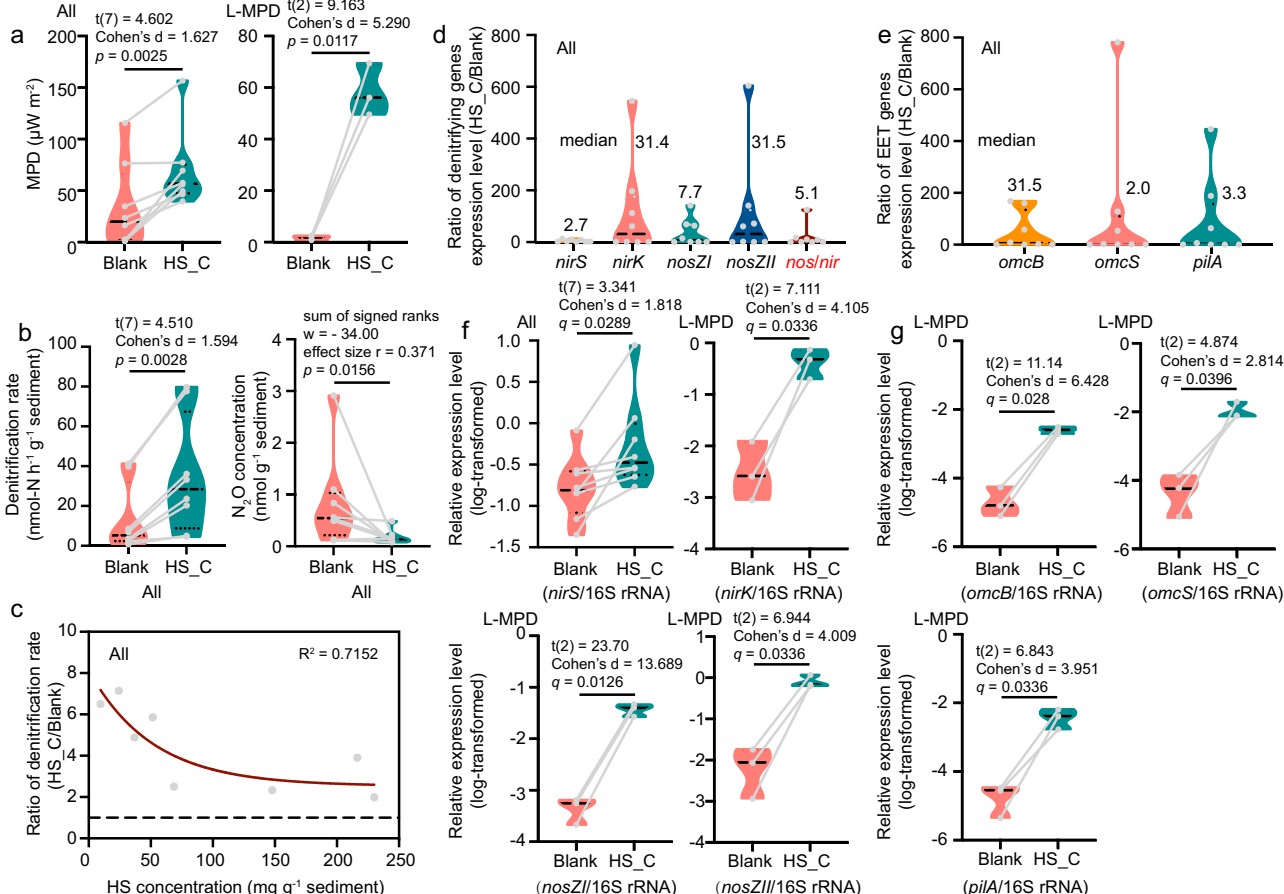

**Fig. 4 | Effects of humic substance (HS) addition on extracellular electron transfer (EET)-mediated denitrification. a** Impact of HS addition on maximum power density (MPD) across all groups and the low-MPD (L-MPD) group (two-tailed paired *t*-test). **b** Influence of HS addition on denitrification rates (two-tailed paired *t*-test) and N₂O accumulation levels (two-tailed Wilcoxon matched-pairs signed rank test). **c** Nonlinear regression analysis (one-phase decay exponential model, GraphPad Prism v9.0.0) demonstrating the quantitative relationship between HS concentration and denitrification enhancement ratio (HS-C/Blank). **d** The ratio of relative expression levels of denitrifying genes, N₂O reduction genes, and *nos/nir*, i.e., (*nosZI*+*nosZII*)/(*nirS*+*nirK*), in HS_C groups compared to Blank groups (HS_C/Blank). **e** The ratio of relative expression levels of EET genes (*omcB*, *omcS*, *pilA*) in HS_C groups compared to Blank groups (HS_C/Blank). **f** Log-transformed relative expression levels of denitrifying genes (*nirS* and *nirK*) and N₂O reduction genes (*nosZI*, and *nosZII*). **g** Log-transformed relative expression levels of EET genes

(*omcB*, *omcS*, and *pilA*). All gene expression levels were normalized against 16S rRNA levels. Statistical comparisons were performed using paired *t*-tests (two-tailed), with Benjamini-Hochberg false discovery rate (FDR) correction for multiple comparisons (7 total tests across f and g) to control FDR at 0.05. FDR-adjusted *p* values (*q* values) are labeled in the figure. Additional statistical information is provided in Supplementary Table S13. Relative expression levels of *nirS* were compared across all samples, while *nirK*, *nosZI*, *nosZII*, *omcB*, *omcS*, and *pilA* were analyzed specifically in the L-MPD subgroup. All groups contain 8 biological replicates, while the L-MPD group contains 3 biological replicates. Sediments without added HS are labeled as the Blank group, while sediments with added commercial HS are designated as the HS_C group. In violin plots, horizontal bars indicate medians, while dotted lines represent 25th and 75th percentiles. Statistical significance was determined at *p* < 0.05.

## Mechanisms of EET in facilitating denitrification in lakes

This study identified 41 medium- (completion ≥50%, contamination <10%) and high-quality (completion >90%, contamination 5%) metagenome-assembled genomes (MAGs) (https://doi.org/10.6084/m9.figshare.28013063), some of which exhibited complete EET capacity (Fig. 6). Our findings demonstrate that EET bacteria transfer electrons either directly or indirectly via endogenous electron shuttles (i.e., flavins) to HS (Fig. 7). Specifically, the proteins FmnA and FmnB are responsible for secreting endogenous flavin shuttles, while DmkA and DmkB synthesize demethylmenaquinone (DMK) derivatives, a critical component in the EET chain[57]. Type II NADH dehydrogenase (Ndh2) transfers electrons from NAD to DMK, which subsequently transfer these electrons to free flavin shuttles via EetA and EetB proteins. These flavins then transfer the electrons to extracellular electron shuttle (e.g., HS)[57]. In this study, MAG LXG.52 was found to possess a complete set of genes for flavin-based EET[57] (Fig. 6), whereas MAG DX.27 lacked the gene required for FmnA synthesis. Consequently, LXG.52 likely transfers electrons indirectly

through flavins, while DX.27 transfers electrons directly to HS. HA can stimulate the release of endogenous electron shuttles, such as riboflavin and flavin mononucleotide[58]. These compounds have been shown to enhance electron transfer and promote EET-mediated denitrification[58,59]. Interestingly, neither LXG.52 nor DX.27 participated directly in the denitrification process (Fig. 6). Further analysis revealed that MAG LXG.52 (family *Anaerovoracaceae*) provided electrons via a heterotrophic glycolysis pathway (Supplementary Fig. S11). In contrast, MAG DX.27 (order *Thermodesulfovibrionales*) utilized an autotrophic dissimilatory sulfur reduction pathway (Supplementary Fig. S12). Previous studies have shown that electric currents can link oxygen reduction at the marine sediment surface with sulfide oxidation in deeper layers[9]. Our findings expand this understanding by suggesting that electron shuttling processes also couple NO₃⁻ reduction in sediments with both sulfide and organic matter oxidation. In summary, both heterotrophic and chemo-lithoautotrophic bacteria are capable of transferring electrons to extracellular electron shuttles.

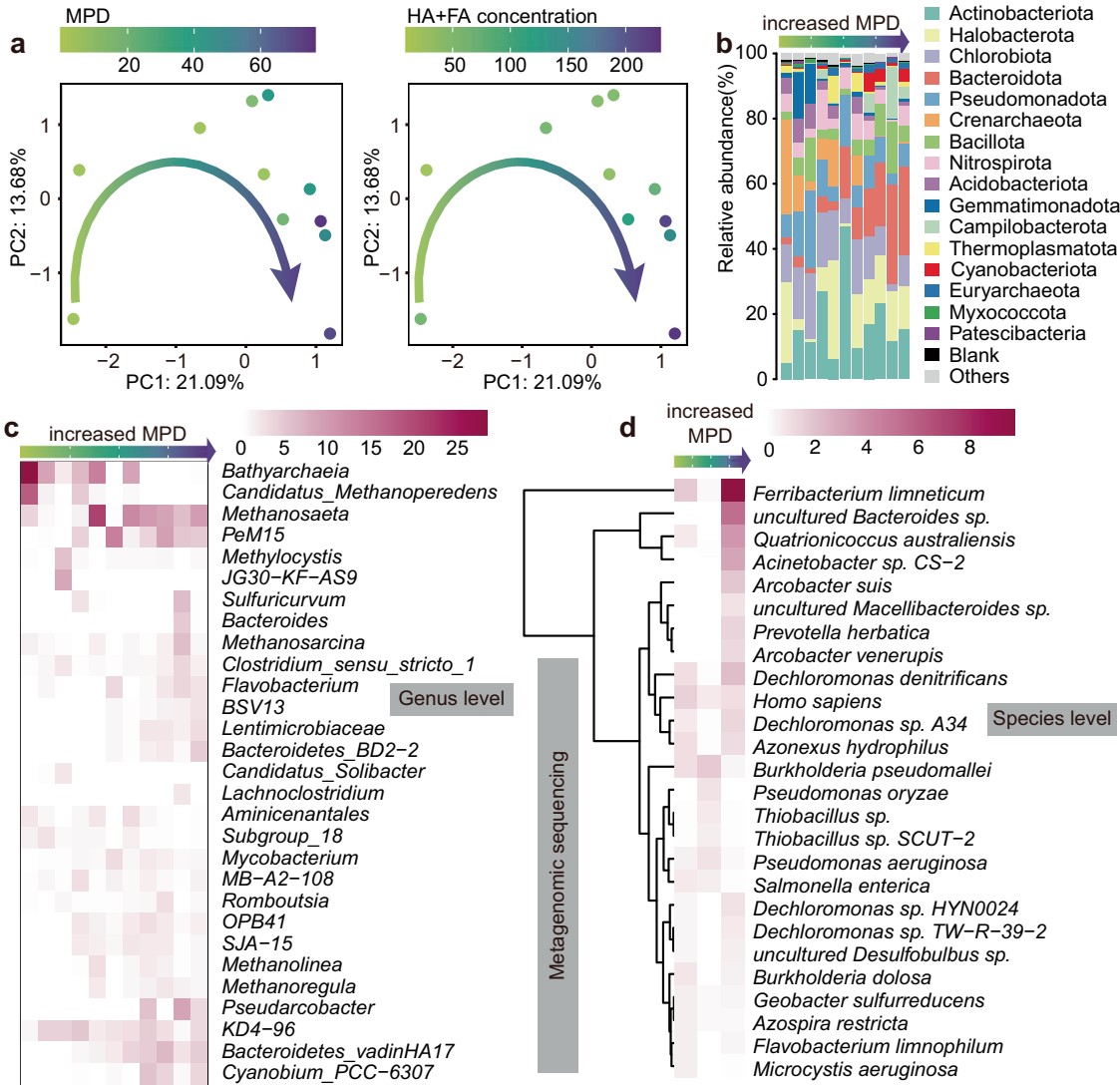

**Fig. 5 | Variation of microbial community structure in estuary sediment along with increased MPD. a** Principal component analysis of the microbial community at genus level. The dots represent microbial community of different sampling sites, and were colored based on maximum power density (MPD) and electron shuttles concentration, respectively. **b–d** The variation of dominant microbial community structure along with MPD at phylum (> 1% in at least one sample), genus (> 2% in at least one sample), and species (> 0.5% in at least one sample) levels. The microbial community structure at phylum and genus levels were analyzed using 16S high-throughput sequencing ($n = 11$ biological replicates). The microbial community structure at species level was analyzed using metagenomic sequencing (DX_1, LX_3, and LX_1; $n = 3$ biological replicates). Abbreviations: *HA* humic acid; *FA* fulvic acid.

Nineteen of the forty-one MAGs were associated with denitrification (Fig. 6). However, each MAG only contained genes mediating one or two reduction steps (Fig. 6)[60]. Consequently, nitrite likely becomes the terminal denitrification product in specific MAGs and the denitrifier *Quatrionicoccus australiensis* due to the absence of nitrite reduction genes[56] (Figs. 5–6). Supporting this, sediment nitrite concentration showed a significant positive correlation with both MPD and humic substance concentration (Supplementary Fig. S13). Furthermore, HS addition significantly increased nitrite accumulation in L-MPD sediments (Supplementary Fig. S13d). Consequently, the EET-coupled denitrification process might enhance the potential risk of toxic nitrite accumulation in sediment. This risk should be carefully considered during ecological monitoring and application of EET for enhancing environmental bioremediation.

Notably, none of these denitrification-associated MAGs possessed fully developed EET capabilities; however, some demonstrated the ability to uptake extracellular electrons. For example, MAGs DX.24 (family *Dissulfurispiraceae*) and DX.76 (family *Burkholderiaceae*) can

receive extracellular electrons via a heme-containing periplasmic membrane-bound cytochrome (HPMC, mediated by DFE_0449 and DFE_0461). These electrons are subsequently transferred to a periplasmic soluble cytochrome (PSC, mediated by DFE_0448, DFE_0462, and DFE_0465) or a complex stabilizing cytochrome (CSC, mediated by DFE_0451 and DFE_0463) (Figs. 6; 7)[61]. This electron transfer process enhanced both aerobic (*napA* and *napB*) and anaerobic (*narG* and *narH*) $NO_3^-$ reduction pathways. Similarly, LXG.32, which contained *octR* for $NO_2^-$ reduction, also possessed HPMC, PSC, and CSC synthesis capabilities. Additionally, through HPMC and PSC, LXG.39 facilitated *norB*-mediated NO reduction, while LXG.62 and LXG.70 facilitated *nosZ*-mediated $N_2O$ reduction. Notably, MAGs LXG.62 and LXG.70 were classified as *nosZII*-type $N_2O$-reducing bacteria (Supplementary Fig. S14). Compared to *nosZI*-type denitrifiers, *nosZII*-type denitrifiers can utilize lower concentrations of $N_2O$[46], whice may explain the lower $N_2O$ emissions observed at high-MPD sampling sites. Altogether, EET processes significantly contribute to completing the denitrification process, leading to high denitrification rates and reduced $N_2O$ emissions.

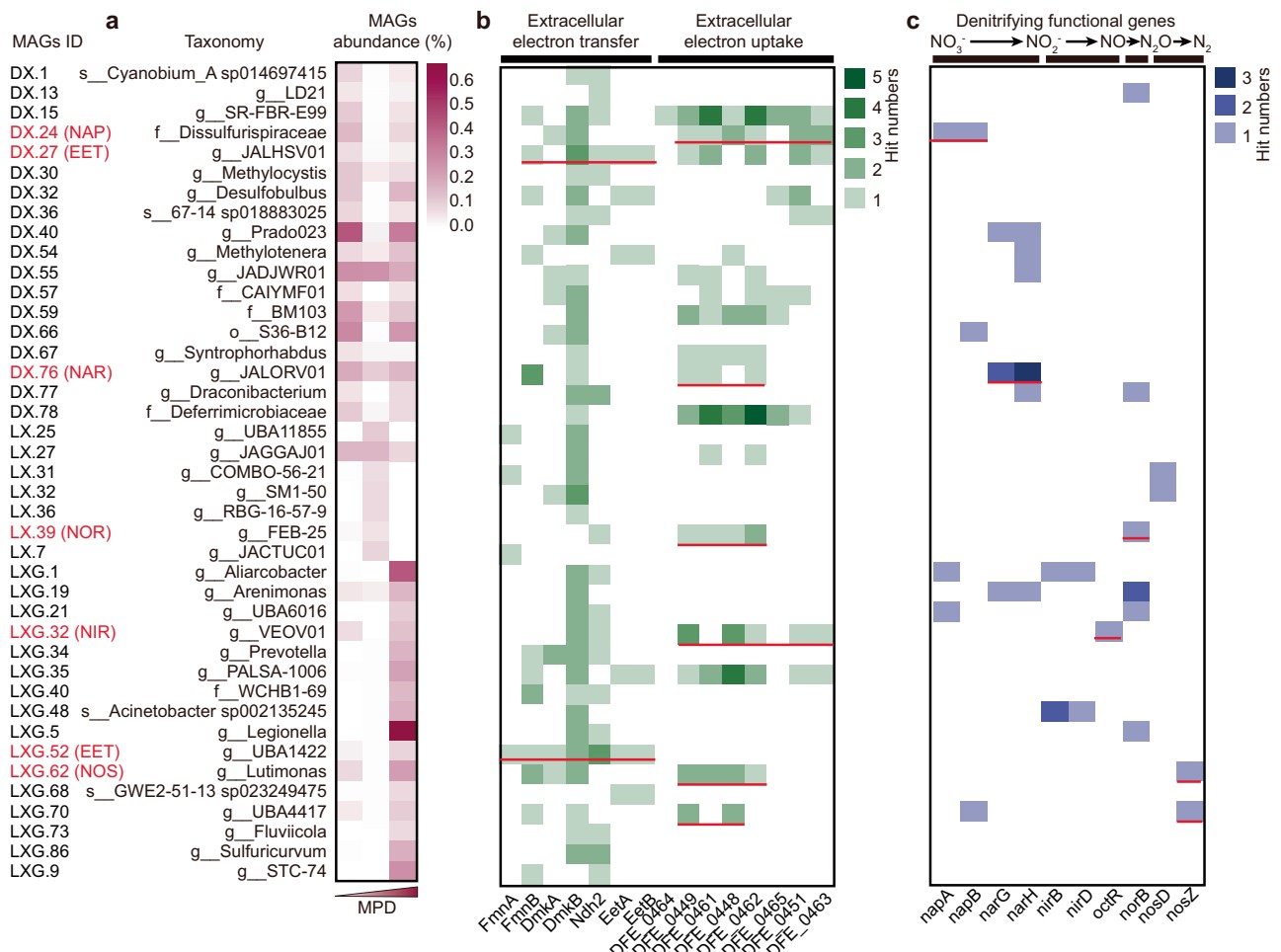

**Fig. 6 | Taxonomy, relative abundance, and functional genes of metagenome-assembled genomes (MAGs) recovered from lake sediments. a** Taxonomy and relative abundance of MAGs in representative low-, medium-, and high-MPD samples (DX_1, LX_3, and LX_1). Relative abundance refers to the proportion of MAGs among all sequencing reads. **b** The presence and hit numbers of extracellular electron transfer and extracellular electron uptake genes in recovered MAGs. MPD,

maximum power density; DFE_0464, outer membrane cytochrome; DFE_0449 and DFE_0461, heme-containing periplasmic membrane-bound cytochrome, DFE_0448, DFE_0462, and DFE_0465, periplasmic soluble cytochrome; DFE_0451 and DFE_0463, complex stabilizing cytochrome. **c** The presence and hit numbers of denitrifying functional genes in recovered MAGs. $n = 3$ biological replicates.

In summary, we propose a conceptual model illustrating how HS-derived electron shuttling modulates nitrogen biogeochemical cycling in lakes. The essence of redox reactions lies in electron transfer. However, the contribution of electron shuttling-mediated external electron supply to sedimentary denitrification and N$_2$O reduction has been largely overlooked. Our findings demonstrate that HS-mediated EET, primarily regulated by *omcB*, significantly enhances biological nitrogen removal in lakes, contributing to eutrophication mitigation. Furthermore, this electron shuttling process promotes N$_2$O reduction (mainly regulated by *nosZII*), mitigating anthropogenically induced global warming. Given the widespread presence of HS electron shuttles in natural environments, such as sediments, soils, and dissolved organic matter in permafrost, understanding the contribution of extracellular and intracellular electrons to earth's biogeochemical cycles on a global scale, as well as their response to global changes, requires further investigation. Since the nitrogen removal enhancement effect of HS amendments diminishes with increasing native HS concentrations in sediment, environmental management should prioritize applying electron shuttle amendments in polluted aquatic systems with low native HS levels to maximize cost-effectiveness. Nevertheless, the specific application methods and efficacy of these approaches remain to be elucidated.

## Methods

### Study sites and samples collection

This study was conducted in the Taihu region, China's third-largest freshwater lake, which spans an area of 2400 km$^2$ (Fig. 1). The basin experiences a subtropical monsoon climate, with an average annual temperature ranging from 16.0–18.0 °C and annual precipitation between 1100 and 1150 mm[45]. River water flows into Lake Taihu from the northwest and exits to the east, with an estimated annual outflow of 7.5 billion m$^3$ [62].

To examine sediments with varying MPD gradients, twelve sampling sites were selected across four influent rivers influenced by urban and agricultural land use (Fig. 1). These sites spanned both submerged and non-submerged vegetation zones (E120°0′31″-E120°24′19″, N31°26′34″-N31°33′68″) (Fig. 1a; Supplementary Table S10). Land use patterns in the Taihu Basin were classified based on data from the Globeland30 website and processed using ArcGIS (version 10.7)[45]. The region has long practiced a summer rice-winter wheat double-crop rotation system[63]. Field investigations and sample collections were conducted in late April 2024, coinciding with the wheat maturity stage. During this phase, farmers typically avoid large-scale nitrogen fertilization due to diminished crop root absorption capacity and risks of delayed maturity. While rainfall is a primary driver of nitrogen loss

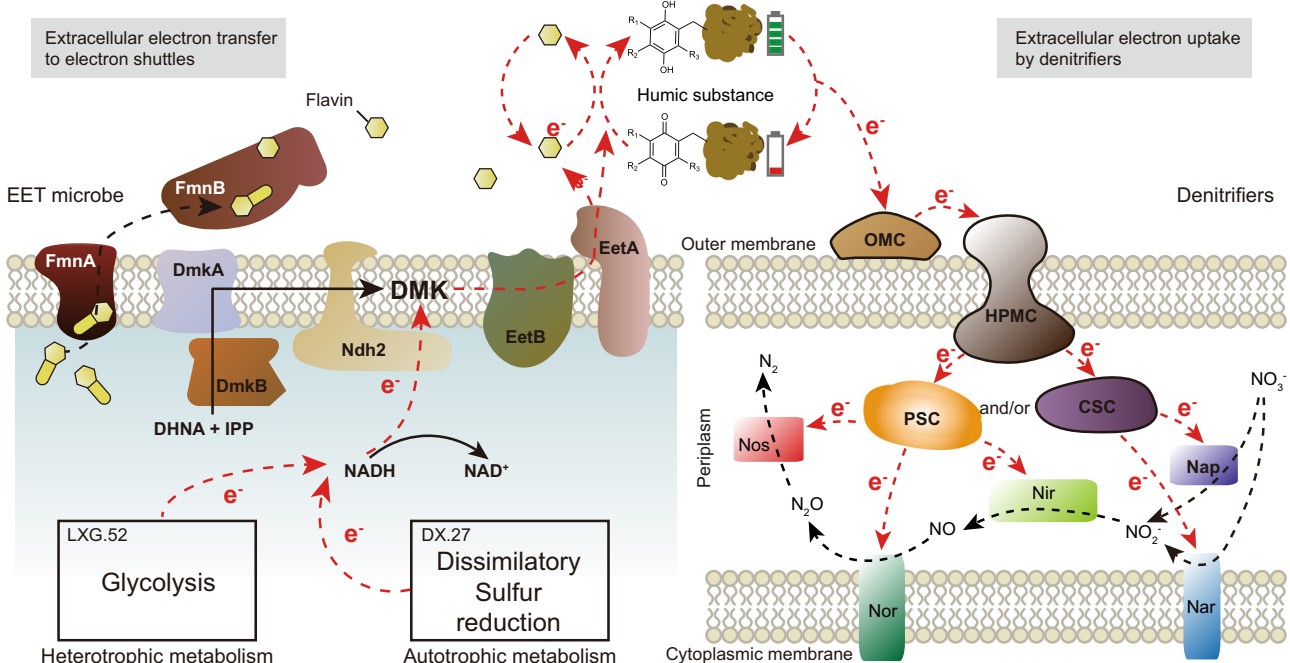

**Fig. 7 | A model illustrating the molecular mechanisms underlying the humic substances-dediated coupled processes of extracellular electron transfer (EET) and uptake by denitrifiers.** Electrons generated by EET microbes through glycolysis or dissimilatory sulfur reduction are transferred stepwise to extracellular electron shuttles (i.e., humic substances). These electrons are then further transferred via a series of cytochromes to denitrification enzymes. DHNA, 1,4-dihydroxy-

2-naphthoyl-CoA; *DMK* demethylmenaquinone; *IPP* isopentenyl pyrophosphate; *OMC* outer membrane cytochrome; *HPMC* heme-containing periplasmic membrane-bound cytochrome; *PSC* periplasmic soluble cytochrome; *CSC* complex stabilizing cytochrome; *Nap* periplasmic nitrate reductase; *Nar* membrane-bound nitrate reductase; *Nir* nitrite reductase; *Nor* nitric oxide reductase; *Nos* nitrous oxide reductase.

from farmland, April is a dry season with minimal precipitation in the region[63,64]. Sampling was conducted on sunny days to avoid short-term drastic changes in nitrogen nutrient concentrations, with |$Z$-score | < 3 confirming no statistical outliers (Supplementary Table S2).

At each site, triplicate water samples were collected at an average sampling depth of ~30 cm below the surface using a 5 L water sampler (Wuhan Xuan Ming Yu Environmental Technology Development Co., Ltd., Wuhan, China). A single surface sediment (0–10 cm depth) was collected per site using a PBS-211 grab sampler (Wuhan Petersen Technology Co., Ltd., Wuhan, China). All samples were stored at 4 °C and transported to the laboratory within 72 h for immediate water quality analysis.

### Electron transfer capacity of fresh sediments
The electron transfer capacity of in situ sediments were evaluated by measuring the MPD of fresh sediments. This was determined using a current density-power density plot generated through MFC equipment[23] (Supplementary Fig. S2). Detailed descriptions of the experimental setup are provided in the Supplementary Methods S1. These include the pre-treatment of the proton exchange membrane, preparation of fresh sediment and buffer solution, and the method used to evaluate the MPD.

### Component and functional groups of extracted HS
Dissolved HS, including HA and FA, were extracted from fresh sediment samples, with 40 g taken from each sampling site[65]. After extraction, the HA and FA samples were freeze-dried, weighed, and normalized based on the dry weight of the sediment (Supplementary Table S1). DX_1, LX_3, and LX_1 were randomly selected as representative samples from L-MPD, M-MPD, and H-MPD, respectively (Supplementary Table S1). The components of the extracted representative HS were first analyzed using EEM fluorescence spectroscopy. Following this, the surface functional groups of the HA and FA powders

was characterized using XPS (AXIS-ULTRA DLD-600W, Japan). Each sample was scanned under a monochromatic Al K$\alpha$ X-ray source. The binding energy calibration was performed using the C 1$s$ peak of the aliphatic carbons at 284.8 eV. High-resolution XPS analysis of the C 1$s$ spectrum identified the predominant carbon-based functional groups in HA and FA, including C-C (284.8 eV), C-OH (286.3 eV), C = O (288.1 eV), and -COOH (289.0 eV). Data processing and peak area integration were conducted with Avantage software (v5.9931, Thermo Fisher).

### Dissolved $N_2$ and $N_2O$, excess dissolved $N_2$ and $N_2O$, and $N_2O$ flux estimation
Dissolved $N_2$ was directly measured using the $N_2$:Ar method with a membrane inlet mass spectrometer (MIMS) system (HPR-40, Hiden Analytical Co.)[33]. Further details of the method are provided in the Supplementary Methods S2. The MIMS system provides rapid, high-precision measurements on small samples without preprocessing, and it has been widely used to study in situ denitrification potential in wetlands, rivers, estuaries, and oceans[33,34].

The concentration of dissolved $N_2O$ was estimated using a modified headspace equilibrium method[66]. For dissolved $N_2O$ analysis, water samples were collected 30 cm below the surface water using 200 mL plastic syringes fitted with a three-way stopcock. More details about water sample collection are provided in Supplementary Methods S3. In each syringe, a 40 mL of the water sample was replaced with an equal volume (40 mL) of ambient air. Ambient air samples for atmosphere $N_2O$ ($C_{air}$) measurement were collected in triplicate at each sampling site. Each syringe was shaken for 5 min to equilibrate the dissolved $N_2O$ between the water and the headspace. The $N_2O$ in the headspace was then gently injected into a 12 mL pre-evacuated vial (Labco, UK) and analyzed within 72 h using a gas chromatograph (7890B Agilent Technologies, Santa Clara, California, USA)[33]. The original concentration of $N_2O$ in water ($C_w$) before equilibrium was

calculated using the headspace balancing method as follows:

$$Cw = C_g \times (k_0 \times R \times T + V_g/V_1) - C_{air} \times V_g/V_1 \quad (1)$$

where $C_g$ represents the $N_2O$ concentration in the headspace after equilibrium ($\mu mol\ L^{-1}$); $C_{air}$ is the site specific ambient air $N_2O$ concentration; $V_g$ and $V_1$ denote the gas volume (40 mL) and water volume (160 mL) in the balance vial, respectively; $R$ is the ideal gas constant ($0.082\ L\ atm\ mol^{-1}\ K^{-1}$); $T$ is the absolute temperature during the balancing procedure ($K$); and $K_0$ ($mol\ L^{-1}\ atm^{-1}$) is the solubility coefficient[67].

Excess dissolved $N_2$ ($\Delta N_2$) and $N_2O$ ($\Delta N_2O$) concentrations were calculated by subtracting the equilibrium concentrations from the measured dissolved concentrations:

$$\Delta N_2 = N_{2(water)} - N_{2(eq)} \quad (2)$$

$$\Delta N_2O = N_2O_{(water)} - N_2O_{(eq)} \quad (3)$$

where $N_{2(water)}$ and $N_2O_{(water)}$ denote the measured concentrations of dissolved $N_2$ and $N_2O$ in water, while $N_{2(eq)}$ and $N_2O_{(eq)}$ represent equilibrium concentrations with the atmosphere[68,69].

Finally, the $N_2O$ flux was estimated using the diffusion method[45]:

$$F = k \times \Delta N_2O \quad (4)$$

where $F$ ($\mu mol\ m^{-2}\ d^{-1}$) is the diffusive $N_2O$ flux from water to the atmosphere; $k$ ($m\ d^{-1}$) is the gas transfer velocity[67].

### Physicochemical parameters analysis
For the water samples, in situ measurements of temperature, dissolved oxygen concentration, pH, oxidation-reduction potential, and electrical conductivity were conducted at a depth of 30 cm using a multiparameter water quality meter (Q31d, HACH, USA). Additionally, total suspended solids, chlorophyll-*a*, total phosphorus, total organic carbon, nitrate nitrogen, nitrite nitrogen, total ammonia nitrogen, and total nitrogen were analyzed using standard methods[70].

### Effects of external HS addition on electron transfer capacity, denitrification rate, $N_2O$ accumulation level, and related functional genes expression
To examine the impact of external HS addition on electron transfer capacity, we designed a paired experiment with two groups: a blank group (no added HS) and a HS_C group (added commercial HS). Commercial HS (H811077, Macklin, Shanghai Macklin Biochemical Technology Co., Ltd., China) was added to the HS_C group at a rate of 20 mg-HS $g^{-1}$ sediment. Eight biological replicates were used, comprising three L-MPD samples (WX_2, WX_3, WY_1) and all M-MPD and H-MPD samples (Supplementary Table S1).

To measure electron transfer capacity, the anode chamber (working volume: 120 mL) of the MFC was inoculated with 15 g of homogenized sediment. The measurement protocol followed the methods described above. To investigate the effects of HS addition on denitrification rates, $N_2O$ accumulation concentration, and functional gene expression, a paired experimental design was implemented subsequently (Supplementary Fig. S15). For each treatment, ~2 g of homogenized sediment and 11 mL 0.17 mM $^{15}NO_3^-$ solution were placed in 12 mL Exetainer vial (039 W, Labco, UK) and incubated in the dark at 25 °C. The $^{15}NO_3^-$ solution was purified with helium (99.999% purity) for 45 min to removal dissolved $N_2$ and $N_2O$ before addition. Denitrification and anammox rates ($n = 8$ biological replicates) were determined using a $^{15}N$-tracer assay. Specifically, at each treatment and time point (0, 6, 12, 24 h), 100 $\mu L$ of saturated $ZnCl_2$ solution was added to terminate the incubation process, with three replicates sacrificed for each

time point. The $^{15}N$-labeled $N_2$ ($^{29}N_2$ and $^{30}N_2$) was analyzed using membrane inlet mass spectrometry (MIMS 200, Bay Instruments, USA). The production rates of $^{29}N_2$ and $^{30}N_2$ were calculated based on the slope of linear regression (Supplementary Fig. S16–S17). The detailed calculation methods for $N_2$ production rates via denitrification and anammox are provided in Supplementary methods S4[71]. The supernatant was also collected for measurement of accumulated nitrite concentration. Additionally, sediments in the Exetainer vials were immediately stored at −20 °C until DNA extraction using the DNeasy PowerSoil Kit (Qiagen, Hilden, Germany)[45]. To quantify total bacterial biomass variation, real-time qPCR was performed at 0 h and 24 h. The primers used for real-time qPCR are listed in Supplementary Table S11, and further details are provided in Supplementary Methods S5.

To analyze $N_2O$ accumulation levels during denitrification ($n = 8$ biological replicates), 5 mL of overlying water was sampled from each vial using a 20 mL sterile syringe equipped with a three-way valve (Supplementary Fig. S15). The syringe was then flushed with 10 mL of helium gas and shaken for 5 min to equilibrate the dissolved $N_2O$ between the liquid (5 mL) and headspace (10 mL) phases. Dissolved $N_2O$ concentrations were determined using the headspace equilibrium method. Concurrently, sediment samples collected from the vials were immediately frozen at −80 °C. Within 72 h, total bacterial RNA was extracted using the OMEGA Soil RNA Kit (R6825-01, Omega Bio-Tek, Norcross, GA, USA). The extracted RNA was reverse-transcribed into cDNA using the PrimeScript™ RT reagent Kit (RR047A, Takara Bio, Shiga, Japan). Real-time qPCR was subsequently performed using the Hieff® qPCR SYBR Green Master Mix (11201ES08, Yeasen, Shanghai, China) to quantify gene abundances of denitrification (*nirS* and *nirK*), $N_2O$ reduction (*nosZI* and *nosZII*), and EET functional genes (*omcB*, *omcS*, and *pilA*). The 16S rRNA gene served as an internal reference gene for normalization. Primers for real-times qPCR are listed in Supplementary Table S11, and detailed protocols can be found in Supplementary Methods S5.

### Microbial community and function analysis
DNA extraction was performed on sediment samples (~0.3 g wet weight per sample) from 12 sampling sites using the DNeasy PowerSoil Kit (47014, Qiagen, Hilden, Germany)[45]. Blank extraction was also conducted alongside the samples to monitor potential contamination during the extraction process. Details for the PCR protocol and library construction are provided in Supplementary Methods S6. The community composition was then examined through Illumina Miseq sequencing. Detailed procedures for Illumina Miseq sequencing, quality control assembly, amplicon sequence variants (ASVs) clustering, and taxonomy annotation are provided in Supplementary Methods S6. The primer set 341 F (5′-CCTACGGGNGGCWGCAG-3′) and 806 R (5′-GACTACHVGGGTATCTAATCC-3′) were used to amplify the V3-V4 regions of the 16S rRNA gene. Both the rarefaction curve and high Good's coverage values (> 0.995) indicated that the sequencing depth was sufficient for this study (Supplementary Fig. S18, Supplementary Table S12). Real-time qPCR was employed to quantify the total bacteria, EET functional genes (*omcB*, *omcS*, and *pilA*), as well as $N_2O$-producing (*nirS* and *nirK*) and reducing (*nosZI* and *nosZII*) functional genes. The primers utilized for real-time qPCR are listed in Supplementary Table S11. See more details in Supplementary Methods S5.

### Assembly and binning of metagenomes
Metagenomic sequencing was performed on representative samples DX1, LX3, and LX1, which correspond to L-MPD, M-MPD, and H-MPD, respectively (Supplementary Table S1). The concentration and integrity of DNA in these samples were assessed using a Qubit fluorometer (Thermo Fisher Scientific, Waltham, MA, USA) with the Qubit dsDNA HS Assay Kit (Q32854, Thermo Fisher Scientific, Waltham, MA, USA), and by agarose gel electrophoresis. DNA libraries were then

constructed using the MGIEasy Fast DNA Library Prep Kit V2.0 (1000017572, MGI, Shenzhen, China) according to the manufacturer's instructions, with an input DNA amount of 200 ng. Finally, the metagenomic sequencing was carried out using the DNBSEQ-T7 platform (MGI, Shenzhen, China) to generate 2×150 bp paired-end reads. Low-quality reads were filtered out using Fastp v0.23.4[72]. High-quality reads were then assembled using MEGAHIT v1.2.9[73] with the following parameters: "--min-contig-len 2500 --k-min 21 --k-max 141". Contigs longer than 2500 bp were selected and binned using MetaBAT2 v2.12.13[74], based on nucleotide frequency and sequencing depth. The resulting MAGs were dereplicated with dRep v3.4.3 (dereplicate -com 50 -con 10)[75]. Finally, non-redundant MAGs were assessed for quality using the "lineage_wf"command in CheckM v1.2.2[76].

### Taxonomic classification, metabolic annotation, and abundance calculation of MAGs

Taxonomic classification of the reads was performed using Kraken v2.1.3[77] and Bracken v2.9[78]. The MAGs, which had undergone dRep processing, were then subjected to taxonomic annotation using the "classify_wf" command in GTDB-Tk v2.4.0 with the Genome Taxonomy Database (Release r220)[79]. Open reading frames (ORFs) predictions for the MAGs were carried out using Prodigal v2.6.3[80], with the options "-p meta" for contigs and "-p single" for MAGs. To annotate and construct metabolic pathways in the MAGs, the METABOLIC tool was used[81]. Specifically, the Hidden Markov Model searches (hmmsearch) in HMMER v3.3.2[82] was employed to match the predicted ORFs with an E-value cut-off of $< 1e^{-5}$. The Pfam[83], TIGRfam[84], and KEGG[85] databases were used for this process. Coverm v0.6.1 (https://github.com/wwood/CoverM) was used to calculate the abundance of non-redundant MAGs based on genome mode with default parameters.

### Statistical analysis

The mean values of three technical replicates at each of the 12 sampling sites were used for statistical analysis ($n = 12$ biological replicates). We quantified key environmental factors influencing in situ denitrification potential (dissolved $N_2$ and $\Delta N_2$ concentration) through sequential statistical analyses. Spearman correlation identified factors significantly associated with denitrification potential, followed by simple linear regression. Variables meeting residual assumptions (homoscedasticity, normality) after transformation were used in stepwise multiple linear regression (bidirectional Akaike Information Criterion optimization). The optimal model was validated by screening multicollinearity (variance inflation factor < 5), re-testing residuals, and assessing predictor significance. Relative influence of factors was determined using standardized coefficients, nested model comparisons (ANOVA), and visualization (MASS, lmtest, car, and ggplot2 packages in R software, version 4.0.4). Detailed code analytical procedures are provided in Supplementary Methods S7. Linear and exponential regression analyses were performed using GraphPad Prism (version 9.0.0). To test the effects of humic substance addition on MPD, denitrification rates, $N_2O$ accumulation levels, microbial biomass, and relative expression levels of functional genes, we conducted paired analyses ($n = 8$ biological replicates). First, the Shapiro-Wilk test was used to examine the normality of the residuals of the paired samples. If the data met the normality assumptions, a paired *t*-test was applied; otherwise, the Wilcoxon matched-pairs signed rank test was used (GraphPad Prism) (Supplementary Table S13). For the analysis of microbial community based on 16S rRNA high-throughput sequencing, the microeco package within the R software environment (version 4.0.4) was employed. Initially, "mitochondria" and "chloroplasts" were removed from the dataset. All samples were then resampled to equalize the sequence number at the minimum value observed among the samples. This resampling procedure was performed to facilitate a more precise evaluation of α-diversity indexes. Spearman correlation analysis, principal component analysis (standardized covariance matrix), and heatmap were carried out using the Hmisc, corrplot, vegan, ggplot2, and pheatmap packages in R software environment (version 4.0.4).

### Reporting summary

Further information on research design is available in the Nature Portfolio Reporting Summary linked to this article.

## Data availability

The following databases were used in this study: Kraken2DB, GTDB r220, NCBI, Pfam, KEGG, TIGRfam, Silva v138.1. The 16S rRNA gene high-throughput sequencing data generated in this study have been deposited in the NCBI database under accession code SRP550967. The metagenomic sequencing data generated in this study have been deposited in the NCBI database under accession code SRP551189. The supplementary methods, figures and tables generated in this study are provided in the Supplementary Information. The processed MAGs (https://doi.org/10.6084/m9.figshare.28013063) and representative *nosZ* sequences (https://doi.org/10.6084/m9.figshare.28041077) data are available at figshare. The raw figures data generated in this study have been deposited in the Github (https://doi.org/10.5281/zenodo.16416644)[86] and figshare (https://doi.org/10.6084/m9.figshare.28050437).

## Code availability

The codes used in this study are available in the GitHub (https://doi.org/10.5281/zenodo.16416644)[86] and figshare (https://doi.org/10.6084/m9.figshare.28050473). Figures were created using GraphPad Prism 9.0.0, R software, and Adobe Illustrator CC 2015.

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

## Acknowledgements

This work was supported by the National Natural Science Foundation of China (Grant No. 42222709, K.S.; 42307505, M.D.; 42107277, L.L.). National key research and development program of China (Grant no. 2023YFC3205800, K.S.), Hubei Provincial Natural Science Foundation of China (2022CFA109, K.S.), GDAS' Project of Science and Technology Development (2020GDASYL–20200101002; 2023GDASZH-2023010103, K.S.), Project of southern marine science and engineering Guangdong laboratory (Guangzhou) (BYQ20231202, K.S.).

## Author contributions

Y.X., Y.W., M.D., S.Z., Y.H., and S.Y. conducted the field experiments; Y.X., M.D., and K.S. supervised mass spectrometry, X-ray photoelectron spectroscopy, water chemistry analysis, bioelectrochemical, and qPCR analysis; Y.W. processed metagenomic data. M.D. processed microbial community and function analysis; M.D., L.L., K.S., and F.W. identified the research questions. M.D., K.S., and F.W. designed the study approach, analyzed the data, and wrote the manuscript.

## Competing interests

The authors declare no competing interests.
