## [Peer Review file · Nature Communications]

Electron shuttling promotes denitrification and mitigates nitrous oxide emissions in lakes

Corresponding Author: Professor Kang Song

Version 0:

Reviewer comments:

Reviewer #1

(Remarks to the Author)

This paper demonstrates that electron transfer mediated by humic acid can facilitate electron transfer from sediments to promote denitrification in the overlying water, thereby reducing the risk of eutrophication and promoting the production of N₂ instead of the greenhouse gas N₂O. This finding is significant for understanding the microbial mechanisms underlying electron shuttling-derived denitrification processes in lakes and adds a new dimension to the understanding of biogeochemical cycles. However, the article is not yet at a level suitable for publication in Nature Communications. The following are the main comments, and the manuscript may be considered for publication after major revision:

The title includes "mitigates eutrophication," but the content of the study primarily focuses on promoting denitrification, without directly addressing eutrophication. Eutrophication is not only related to nitrogen but also involves other factors such as phosphorus and dissolved oxygen concentrations. Therefore, the title should be revised to better reflect the actual content of the study.

While humic acid promotes denitrification through electron transfer, the products of denitrification are not always limited to N₂. The connection between enhanced denitrification and reduced N₂O production needs to be further substantiated. The authors should provide stronger evidence and more detailed discussion of this relationship.

The statement that an increase in hydroxyl and carbonyl functional groups promotes electron transfer capacity (EET capacity) seems somewhat tenuous. While many substances contain these functional groups, not all of them facilitate electron transfer. The authors should provide further justification for this claim or consider rephrasing it.

The manuscript could place greater emphasis on the new experimental findings obtained by the authors. Hypothetical conclusions and speculations should not be included in the abstract and conclusion sections. The abstract and conclusion should focus on the results and their direct implications, while speculative content should be placed in the discussion section.

(Remarks on code availability)

Reviewer #2

(Remarks to the Author)

Electron shuttling mitigates eutrophication and nitrous oxide emissions in lakes

The authors of the study "Electron shuttling mitigates eutrophication and nitrous oxide emissions in lakes" analyzed samples from 12 sites of a freshwater lake in China in terms of the use of external electron shuttles to support denitrification. They demonstrate that the potential for full denitrification increases with the amount of humic substances and MPD and that this is caused by a strongly significant correlation of denitrification genes and EET related genes. Based on MAGs they further indicate that this coupling might take place between different bacterial/archaeal strains.

The overall research question is interesting and relevant and the authors nicely combined molecular and in situ

measurements with ¹⁵N incubation measurements. There is certainly potential in the data and study and the manuscript was generally easy to read and the figures easy to understand. I very much appreciate the provision of the method related sketches, but very often the description of sampling and measurements is not very detailed and it is hard to repeat this study based on the description given (see comments below).

However, the manuscript has some major limitations:

1. The conclusions of the study solely relies on correlative analysis of gene abundance/presence and gas emission and HS/MPD data. There is no proof of principal that or which microbes were really active. The ¹⁵N incubation experiment would have been an excellent target to measure active microbial populations to better understand the mechanism behind the observation.
2. The authors measured several things which were not included in discussion or results at all, for example, the Annamox measurement (where I have more concerns outlined below) and nutrient, oxygen data etc. obtained from water and sediment. Is it possible that another factor than HS/MPD drives gas emissions like nitrate availability or oxygen concentration?
3. Despite the relevance of N₂O emissions, I feel that potential nitrite accumulation is similarly important but was not considered in the study, although it seems to be measured for some instances.
4. For me it is not clear whether the positive correlation comes from general growth or enrichment of certain groups or increased activity. How was the overall increase of bacterial biomass compared to the targeted genes?
5. In the results and discussion part the comparison to existing data is too superficial, to which extent are the data transferable, are your values comparable to other or not. What is the true new and novel piece of data compared to other studies as the link between HS and denitrification was already shown previously.

Specific comments:

Abstract:

It is generally not clear how you performed the study, which methods did you use, where did you the samples derive from (lake sampling or lab experiment, or both?)

L25: Electron donors you mean.

L29: Write "in situ" in italics

L30: HS is not explained before.

Introduction:

L48ff: Beside the production of N₂O there is also a certain risk of the accumulation of toxic nitrite during denitrification, which is similarly important and dangerous.

L55: What do you mean with "cable" bacteria? Is that a common term, if so add reference or explain better?

L61: It is not clear what you mean with "electron acceptors" as it stand it sounds like a protein/enzyme but then you refer to bacteria later in the sentence. Please be more precise and add the type of acceptor you refer to e.g. electron accepting bacteria. Then you refer to HS in the next sentence. This passage needs clarification.

L86: Is "electroactive" the correct term? Maybe better "involved in EET".

L86: As it stand now, it sounds as if the process is done by two independent bacteria in cooperation, one doing the nitrate reduction and the other the HS oxidation. Do I get this correctly? In the manuscript you cited before (Coates et al) it seemed to be done by one microorganism. Please clarify.

Methods:

L251/Table S1: I do not really get your classification of high, medium low HS concentrations. For example, WX1 (Low HS) has higher values than DX2 (medium)

L253: As I understood, samples were taken in influent areas, thus I assume water movement was quite substantial.

Sampling took place ones in 2024 for sediments and water samples. I wonder, how representative the samples can be if taken only ones. Could the measured values be a snapshot of a recent event happening around the river area like harvest or fertilization etc.

L254: Did you also sample 3 sediment samples per site? If not I wonder how replication looks like and assume that t-test are not a suitable statistical method.

Section 3.3.: I don't really get the design of this measurements, there are several conflicting statements in this section. I think a scheme would be helpful. It is also not clear if that was done with sediments from all 12 locations, and how the proportion of target sediments and WX1 sediment was.

L272/267: These two lines contradict each other ones you said HS-C group received 300mg ones 40 mg of commercial HS.

L269/273: Ones you said 15g WX1 and ones 2g. I don't get the difference.

L273: Which tracer did you use? If you want to distinguish denitrification and annamox I assume you need to different ones, meaning also to separate incubations or?

L276: Was the denitrification activity stable over time? If not, can you exclude microbial growth in HS treated vials, which then explain the increase in denitrification activity?

L279: If you used ¹⁵N-nitrate, I don't understand how you can calculate annamox, which uses ammonia instead of nitrate.

L303/304: The Cair N₂O concentrations might differ. Please specify which values you used for correction and how or if you measured site specific Cair N₂O concentrations (highly recommended).

L328: Why did you measure physicochemical values in 30cm, when you otherwise sampled from 10 cm below water surface. I assume that there is a gradient of nutrients across the water column.

L345: Please also provide information about qPCR standard source, qPCR performance, quality check of amplicon size etc.

Section 3.7 Please provide more details about the PCR protocol and library preparation kit. Also I miss important quality controls like blank extraction and negative PCR controls processed along the sediment samples. This is especially important for low biomass samples.

Section 3.8. How did you prepare the metagenomic library, which kits, which input, how sequenced (also with MiSeq)?

L374: Add the database you used.

L375: Couldn't find the metagenomic data in NCBI, please upload and release before publication and at least provide reviewer link meanwhile.

Section 3.10: Regression analysis are not mentioned as well as PCA (which distance matrix did you use). I am also not sure how exactly the replication of the study worked. Did you samples three times per site, then site needs to be random factor or did you have only 1 sample per site, then a t-test or similar is not suitable at all.

Results/Discussion:

L98: Are these values comparable to other studies?

Fig. 2: Panel b,c are showing redundant content as panel a. I suggest to put into supplement. Panel d, e: I still don't understand if that was done for all 12 sites or only for 1. It would be very interesting to see the relationship between the treatments and inherent HS concentrations, does that weaken in high HS sites?

Fig. 3: What are representative samples, which ones did you exactly use, and how did you decide?

L172: Throughout the whole text please write 16S rRNA gene.

Fig. 5c: Where do the chloroplasts come from, does it make sense to keep them in the dataset?

L185: Especially for *Quatrionococcus* it was shown before that nitrite is a likely endproduct of denitrification. How did nitrite concentration correlate to your data in general? Did you measure it in the in situ and incubation samples? It would be quite important to see whether there is an accumulation of nitrite across your MPD/HS gradient. Also your MAGs indicate that there might be a potential risk of nitrite accumulation (DX76).

L191: Please give threshold values for medium and high quality MAGs.

L215: I don't think you can say that sequential denitrification is the general mode. This strongly depends on the strain, environmental conditions and can be even different for closely related species. To proof this you need more physiological testing of isolates of your system, or at least some transcription analysis over a timer series.

L231: I am not sure if you can talk about efficiency, that would require some kind of normalization to be able to compare across the different sites, e.g. N₂ production per mol nitrate or so.

(Remarks on code availability)

I partly reviewed to code, which is not too helpful. The read.me did not contain a list of packages to be installed. Regarding the 16S code there is even a difference between what the code states and what the manuscript states for the truncation command.

--p-trunc-len-f 0 \ (in code)

--p-trunc-len-f 240 (in manuscript)

It would be helpful to see the error plots etc on which the truncation decision was made. As it stand now it can't be judged if that was correctly done.

For the community to learn from it, it would very helpful to add an explanation what each single step does, especially for the metagenomic code.

Version 1:

Reviewer comments:

Reviewer #1

(Remarks to the Author)

The authors have carefully addressed the comments raised by the reviewers and the manuscript can be accepted now for publication.

(Remarks on code availability)

Reviewer #2

(Remarks to the Author)

This was a revised manuscript analysing denitrification in sediments after humic substance addition serving as electron shuttle. The study combines field and lab experiments as well as 15N,transcription and metagenomic analyses. The additional measurements done during the revision strongly improved the manuscript by clearly identifying denitrifier activity as the underlying mechanism and also unraveling the potential risk of nitrite accumulation under specific circumstances. Both conclusions strongly improved the novelty of the manuscript. I am fully satisfied with the responses and changes made to the manuscript. The additional methodological information in teh supplement is now also sufficient.

(Remarks on code availability)

I just checked the read me file, and find it much better organized then before and much mure helpful in general.

Response to the reviewers' comments

We are grateful to the reviewers for their careful reading of the manuscript, constructive criticisms, positive comments and encouraging remarks, all of which have significantly enhanced the quality of our resubmitted paper. Below, we have addressed each reviewer's comments (noted in *italic*) with our responses (in blue):

REVIEWER COMMENTS

Reviewer #1 (Remarks to the Author):

[Comment 1-1] This paper demonstrates that electron transfer mediated by humic acid can facilitate electron transfer from sediments to promote denitrification in the overlying water, thereby reducing the risk of eutrophication and promoting the production of N₂ instead of the greenhouse gas N₂O. This finding is significant for understanding the microbial mechanisms underlying electron shuttling-derived denitrification processes in lakes and adds a new dimension to the understanding of biogeochemical cycles. However, the article is not yet at a level suitable for publication in Nature Communications. The following are the main comments, and the manuscript may be considered for publication after major revision:

[Response] Thank you very much for your positive comments on our manuscript. Your comments below have been extremely helpful in guiding us through a thorough revision. We truly appreciate this professional review, which has significantly enhanced the quality of our paper. Detailed modifications are provided in our responses to the specific comments below.

[Comment 1-2] 1. The title includes "mitigates eutrophication," but the content of the study primarily focuses on promoting denitrification, without directly addressing eutrophication. Eutrophication is not only related to nitrogen but also involves other factors such as phosphorus and dissolved oxygen concentrations. Therefore, the title should be revised to better reflect the actual content of the study.

[Response] We agree that the original title may have overemphasized the direct impact on eutrophication. To address this concern, we have revised the title.

[Line 1] “Electron shuttling promotes denitrification and mitigates nitrous oxide emissions in lakes”.

[Comment 1-3] 2. While humic acid promotes denitrification through electron transfer, the products of denitrification are not always limited to N₂. The connection between enhanced denitrification and reduced N₂O production needs to be further substantiated. The authors should provide stronger evidence and more detailed discussion of this relationship.

[Response] Thank you for your insightful comment! To address this, we have re-conducted the ¹⁵N isotope experiment. Our measurements of both denitrification rates and accumulated N₂O levels demonstrate that humic substance (HS) enhance denitrification rates while simultaneously reducing N₂O production. Furthermore, by analyzing the expression levels of denitrifying genes (*nirS* and *nirK*) and N₂O reducing genes (*nosZI* and *nosZII*), we provide additional mechanistic evidence supporting these findings.

[Lines 193-228] “2.5 HS addition enhanced EET and complete denitrification processes in sediment

To confirm whether HS enhance denitrification performance and reduce N₂O emissions by increasing EET capacity, we supplemented sediment samples with additional HS and monitored MPD, denitrification rate, N₂O accumulated level, and functional gene expression level (Fig. 4). Compared to the blank group (no additional HS), the HS-added group (HS_C) showed significantly elevated MPD ($p = 0.0025$) (Fig. 4a). Further analysis using ¹⁵N isotope labeling confirmed that denitrification (71.1±31.1%) was the dominant pathway for N₂ production, with HS addition enhanced denitrification rates by 2.6 times ($p = 0.0028$, Fig. 4b). These results align with prior studies showing that electron shuttles like HA accelerate electron transfer and boost denitrification performance in wastewater treatment systems^{48, 49, 50, 51}. Importantly, after 24 h incubation, HS addition reduced accumulated N₂O concentration by 78.3% compared to the Blank group ($p = 0.0156$). Similar effects have been observed with FA, which suppressed N₂O generation by 41.3%~98.8% in *Paracoccus denitrificans*-mediated denitrification⁵². Together, these findings demonstrate that natural electron shuttles such as HS significantly enhance *in situ* denitrification in lake sediments.

Bacterial biomass exhibited no significant difference during the 24-h incubation (Supplementary Fig. S10). To characterize active microbial processes, we therefore further analyzed the effects of HS addition on functional gene expression (Fig. 4d-g). At the initial time point (0 h), relative expression levels of these functional genes showed no significant differences (Supplementary Table S8). By 24 h, however, the ratios for all functional gene expression levels (HS_C/Blank) were notably higher than 1 (ranging from 4.6- to 396.1-fold), indicating that HS addition broadly enhanced gene expression associated with denitrification, N₂O reduction, and EET (Fig. 4d-e). Notably, the high ratio of N₂O-reducing genes (*nosZI+nosZII*) to denitrifying genes (*nirS+nirK*), quantified as *nos/nir* (Fig. 4d), highlighted the preferential upregulation of N₂O reduction pathways. These findings align with prior studies showing that FA upregulates denitrification genes, particularly *nosZ*, in the model denitrifier *Paracoccus denitrificans*⁵². Our results demonstrate that HS-enhanced EET preferentially improves N₂O reduction activity, thereby accelerating denitrification rates while reducing N₂O production.

Notably, across the tested HS concentration range (0-230 mg-HS g⁻¹ sediment), the enhanced ratios (HS_C/Blank) of denitrification rate decreased exponentially with increasing native HS concentrations ($R^2 = 0.6899$) (Fig. 4c). This trend aligns with the observation that L-MPD samples exhibited a 34.1-fold increase in MPD after HS addition, significantly higher than the 2.0-fold increase in all samples (Fig. 4a). Consistently, while HS addition broadly upregulated functional genes (Fig. 4d-e; Supplementary Table S9), significant overexpression of *nirK*, *nosZI*, *nosZII*, and all EET genes occurred exclusively in L-MPD samples (Fig. 4f-g). These results demonstrate that the efficacy of exogenous electron shuttle amendment attenuates significantly in sediments with high native HS concentrations (> 50 mg-HS g⁻¹). Therefore, adding electron shuttles in low-HS environments can achieve optimal cost-effectiveness for nitrogen pollution remediation and N₂O mitigation.”

[Lines 392-433] “3.6 Effects of external HS addition on electron transfer capacity, denitrification rate, N₂O accumulation level, and related functional genes expression

To examine the impact of external HS addition on electron transfer capacity, we designed a paired experiment with two groups: a blank group (no added HS) and a HS_C group (added commercial HS). Commercial HS (Macklin, Shanghai Macklin Biochemical Technology Co., Ltd., China) was added to the HS_C group at a rate of 20 mg-HS g⁻¹ sediment. Eight biological replicates were used, comprising three L-MPD samples (WX_2, WX_3, WY_1) and all M-MPD and H-MPD samples (Supplementary Table S1).

To measure electron transfer capacity, the anode chamber (working volume: 120 mL) of the MFC

was inoculated with 15 g of homogenized sediment. The measurement protocol followed the methods described above. To investigate the effects of HS addition on denitrification rates, N₂O accumulation concentration, and functional gene expression, a paired experimental design was implemented subsequently (Supplementary Fig. S15). For each treatment, approximately 2 g of homogenized sediment and 11 mL 0.17 mM ¹⁵NO₃⁻ solution were placed in 12 mL Exetainer vial (039W, Labco, UK) and incubated in the dark at 25°C. The ¹⁵NO₃⁻ solution was purified with helium (99.999% purity) for 45 minutes to removal dissolved N₂ and N₂O before addition. Denitrification and anammox rates (*n* = 8 biological replicates) were determined using a ¹⁵N-tracer assay. Specifically, at each treatment and time point (0, 6, 12, 24h), 100 µL of saturated ZnCl₂ solution was added to terminate the incubation process, with three replicates sacrificed for each time point. The ¹⁵N-labeled N₂ (²⁹N₂ and ³⁰N₂) was analyzed using membrane inlet mass spectrometry (MIMS 200, Bay Instruments, USA). The production rates of ²⁹N₂ and ³⁰N₂ were calculated based on the slope of linear regression (Supplementary Fig. S16-S17). The N₂ production rates via denitrification and anammox were calculated based on previously reported ⁷¹ (See calculation methods in Supplementary methods S4). The supernatant was also collected for measurement of accumulated nitrite concentration. Additionally, sediments in the Exetainer vials were immediately stored at -20 °C until DNA extraction using the DNeasy PowerSoil Kit (Qiagen, Hilden, Germany) ⁴⁵. To quantify total bacterial biomass variation, real-time qPCR was performed at 0 h and 24 h. The primers used for real-time qPCR are listed in Supplementary Table S11, and further details are provided in Supplementary Methods S5.

To analyze N₂O accumulation levels during denitrification (*n* = 8 biological replicates), 5 mL of overlying water was sampled from each vial using a 20 mL sterile syringe equipped with a three-way valve (Supplementary Fig. S15). The syringe was then flushed with 10 mL of helium gas and shaken for 5 minutes to equilibrate the dissolved N₂O between the liquid (5 mL) and headspace (10 mL) phases. Dissolved N₂O concentrations were determined using the headspace equilibrium method. Concurrently, sediment samples collected from the vials were immediately frozen at -80 °C. Within 72 h, total bacterial RNA was extracted using the OMEGA Bacterial RNA Kit (Omega Bio-Tek, Norcross, GA, USA). The extracted RNA was reverse-transcribed into cDNA using the PrimeScript™ RT reagent Kit (Takara Bio, Shiga, Japan). Real-time qPCR was subsequently performed using the Hieff® qPCR SYBR Green Master Mix (Yeasen, Shanghai, China) to quantify gene abundances of denitrification (*nirS* and *nirK*), N₂O reduction (*nosZI* and *nosZII*), and EET functional genes (*omcB*, *omcS*, and *pilA*). The 16S rRNA gene served as an internal reference gene for normalization. Primers for real-times qPCR are listed in Supplementary Table S11, and detailed protocols can be found in Supplementary Methods S5.”

[Lines 487-492] “To test the effects of humic substance addition on MPD, denitrification rates, N₂O accumulation levels, microbial biomass, and relative expression levels of functional genes, we conducted paired analyses (*n* = 8 biological replicates). First, the Shapiro-Wilk test was used to examine the normality of the residuals of the paired samples. If the data met the normality assumptions, a paired *t*-test was applied; otherwise, the Wilcoxon matched-pairs signed rank test was used (GraphPad Prism) (Supplementary Table S13).”

Fig. 4. Effects of humic substance (HS) addition on extracellular electron transfer (EET)-mediated denitrification. **a** Impact of HS addition on maximum power density (MPD) across all groups and the low-MPD (L-MPD) group. **b** Influence of HS addition on denitrification rates and N₂O accumulation levels. **c** Nonlinear regression analysis (one-phase decay exponential model, GraphPad Prism v9.0.0) demonstrating the quantitative relationship between HS concentration and denitrification enhancement ratio (HS-C/Blank). **d** The ratio of relative expression levels of denitrifying genes, N₂O reduction genes, and *nos/nir*, i.e., (*nosZI*+*nosZII*)/(*nirS*+*nirK*), in HS_C groups compared to Blank groups (HS_C/Blank). **e** The ratio of relative expression levels of EET genes (*omcB*, *omcS*, *pilA*) in HS_C groups compared to Blank groups (HS_C/Blank). **f** Log-transformed relative expression levels of denitrifying genes (*nirS* and *nirK*) and N₂O reduction genes (*nosZI*, and *nosZII*). **g** Log-transformed relative expression levels of EET genes (*omcB*, *omcS*, and *pilA*). All gene expression levels were normalized against 16S rRNA levels. Additional statistical information is provided in Supplementary Table S13. Relative expression levels of *nirS* were compared across all samples, while *nirK*, *nosZI*, *nosZII*, *omcB*, *omcS*, and *pilA* were analyzed specifically in the L-MPD subgroup. In violin plots, horizontal bars indicate medians, while dotted lines represent 25th and 75th percentiles.

Supplementary Fig. S15 | Workflow for investigating the effects of humic substance addition on denitrification rates, N₂O accumulation concentration, bacterial biomass variation, and active microbial processes. The denitrification rate was determined using ¹⁵N isotope tracing method, while N₂O accumulation concentration was measured via the headspace equilibrium method. qPCR was conducted to quantify total bacterial biomass variation during the 24-hour incubation experiment. Reverse transcription quantitative PCR (RT-qPCR) was employed to assess the relative expression levels of three functional gene categories: (1) denitrification genes (*nirS*, *nirK*), (2) N₂O reduction genes (*nosZI*, *nosZII*), and (3) extracellular electron transfer (EET, including *omcB*, *omcS*, and *pilA*) genes. All functional gene expression levels were normalized against the 16S rRNA gene. $n = 8$ biological replicates, including three L-MPD samples (WX_2, WX_3, and WY_1) and all M-MPD and H-MPD samples (Supplementary Table S1).

Supplementary Figure S16 | Linear regression analysis of dissolved $^{30}\text{N}_2$ concentrations variation ($n = 8$ biological replicates). For LX3, only three time points (0-, 6-, and 12-h) were setup due to insufficient sediment samples. To maximize data retention while ensuring a suitable fitting R^2 ($R^2 > 0.8$), all available data points were included in the regression analysis, excluding the 24-h's data in WX_3, LX_4, and LX_1. Each data point and error bar represents the mean and standard deviation, respectively ($n = 3$ technical replicates).

Supplementary Figure S17 | Linear regression analysis of dissolved $^{29}\text{N}_2$ concentrations variation ($n = 8$ biological replicates). For LX3, only three time points (0-, 6-, and 12-h) were setup due to insufficient sediment samples. To maximize data retention while ensuring a suitable fitting R^2 ($R^2 > 0.86$ except for WY_1 ($R^2 = 0.6253$ for Blank group)), all available data points were included in the regression analysis, excluding the 24-h's data in LX_4, LX_1 and LX_2.

Supplementary Table S8 | Relative expression levels of denitrifying genes and N_2O reduction genes, and extracellular electron transfer (EET) genes at 0 h.

Samples	Processes	Genes	Blank	HS_C	
All	Denitrifying genes	nirS ($\times 10^{-2}$)	2.5 (1.0)	3.9 (1.8)	$p = 0.1656^a$
		nirK ($\times 10^{-3}$)	5.6 (3.0)	1.4 (0.3)	$p = 0.2500^b$
	N_2O -reducing genes	nosZI ($\times 10^{-4}$)	4.6 (1.0)	6.3 (1.9)	$p = 0.2736^a$
		nosZII ($\times 10^{-3}$)	9.6 (5.4)	1.2 (0.4)	$p = 0.1484^b$
	EET genes	omcB ($\times 10^{-5}$)	2.8 (1.2)	0.6 (0.3)	$p = 0.1177^a$
		omcS ($\times 10^{-5}$)	3.9 (2.2)	1.1 (0.4)	$p = 0.3828^b$
pilA ($\times 10^{-5}$)		9.7 (5.3)	3.7 (1.3)	$p = 0.7422^b$	
L-MPD	Denitrifying genes	nirS ($\times 10^{-4}$)	3.5 (1.8)	6.4 (2.3)	$p = 0.2500^b$
		nirK ($\times 10^{-3}$)	1.6 (0.1)	2.0 (0.6)	$p = 0.7500^b$
	N_2O -reducing genes	nosZI ($\times 10^{-4}$)	2.1 (0.5)	4.1 (1.6)	$p = 0.2836^a$
		nosZII ($\times 10^{-3}$)	2.4 (0.8)	1.9 (0.6)	$p = 0.7436^a$
	EET genes	omcB ($\times 10^{-5}$)	0.9 (0.4)	1.1 (0.7)	$p = 0.7814^a$
		omcS ($\times 10^{-5}$)	1.5 (0.8)	2.1 (0.7)	$p = 0.4476^a$
		pilA ($\times 10^{-5}$)	3.1 (1.2)	6.9 (2.4)	$p = 0.2073^a$

Values were presented as mean (standard error of mean). All functional gene expression levels were

normalized to the 16S rRNA gene. $n = 8$ biological replicates for All samples and $n = 3$ biological replicates for L-MPD samples. The Shapiro-Wilk test was used to examine the normality of the residuals of the paired samples. If the data met the normality assumptions, a paired t -test was applied; otherwise, the Wilcoxon matched-pairs signed rank test was used (GraphPad Prism, version 9.0.0). ^a Paired t -test was used; ^b Wilcoxon matched-pairs signed rank test was used.

Supplementary Table S9 | Relative expression levels of denitrifying genes and N₂O reduction genes, and extracellular electron transfer (EET) genes at 24 h.

Processes	Genes	Blank	HS_C	HS_C/Blank
Denitrifying genes	nirS	0.2685 (0.1043)	1.4880 (1.0480)	4.6 ± 1.6
	nirK	0.0383 (0.0243)	0.1991 (0.0986)	98.9 ± 63.0
N ₂ O-reducing genes	nosZI	0.0035 (0.0020)	0.0164 (0.0069)	33.2 ± 15.8
	nosZII	0.0670 (0.0416)	0.3388 (0.1555)	106.4 ± 68.8
EET genes	omcB	0.0002 (0.0001)	0.0012 (0.0005)	74.2 ± 32.2
	omcS	0.0006 (0.0004)	0.0057 (0.0025)	396.1 ± 274.4
	pilA	0.0001 (0.0001)	0.0019 (0.0009)	201.8 ± 115.3

Values were presented as mean (standard error of mean) ($n = 8$ biological replicates). All functional gene expression levels were normalized to the 16S rRNA gene.

Supplementary Table S13. Statistical analysis information.

		Normality of Paired t -test Residuals				Wilcoxon matched-pairs signed rank test
		Shapiro-Wilk (p value)	t	df	p value (two-tailed)	p value (two-tailed)
MPD	All	0.4848	0.9046	7	0.3957	-
	L-MPD	0.5246	9.163	2	0.0117	-
D_{total}	All	0.6130	4.510	7	0.0028	-
C_{N_2O}	All	0.0105	-	-	-	0.0156
nirS /16S rRNA	All	0.0770	3.341	7	0.0124	-
nirK /16S rRNA	All	0.2548	0.6325	7	0.5472	-
	L-MPD	0.2025	7.111	2	0.0192	-
nosZI /16S rRNA	All	0.1428	0.8431	7	0.4270	-
	L-MPD	0.0675	23.70	2	0.0018	-
nosZII /16S rRNA	All	0.1865	0.5225	7	0.6174	-
	L-MPD	0.0893	6.944	2	0.0201	-
omcB /16S rRNA	All	0.0634	0.9950	7	0.3529	-
	L-MPD	0.5600	11.14	2	0.0080	-
omcS /16S rRNA	All	0.4857	0.7625	7	0.4707	-
	L-MPD	0.4291	4.874	2	0.0396	-
pilA /16S rRNA	All	0.2729	0.9055	7	0.3953	-
	L-MPD	0.8322	6.843	2	0.0207	-

MPD, maximum power density; D_{total} , Denitrification rate; C_{N_2O} , Dissolved N₂O concentration; All, All groups ($n = 8$) included three biological replicates in Low-MPD (L-MPD) group, three biological

replicates in medium-MPD and two biological replicates in high-MPD groups. L-MPD, low-MPD group (WX_2, WX_3, and WY_1). Normality of residuals between paired samples (blank and HS-added) was assessed using the Shapiro-Wilk test. For metrics with normal residuals ($p > 0.05$), a paired t -test was used, while the Wilcoxon matched-pairs signed rank test was applied for non-normal residuals ($p < 0.05$). All analyses were performed using GraphPad Prism (Version 9.0.0).

[Comment 1-4] 3. *The statement that an increase in hydroxyl and carbonyl functional groups promotes electron transfer capacity (EET capacity) seems somewhat tenuous. While many substances contain these functional groups, not all of them facilitate electron transfer. The authors should provide further justification for this claim or consider rephrasing it.*

[Response] We agree with your comments and have revised the manuscript accordingly. Specifically, we removed the statement in the Abstract regarding “an increase in hydroxyl and carbonyl functional groups promotes electron transfer capacity (EET capacity)”.

[Lines 152-153] “Overall, the increased concentrations of HA and FA, combined with the rise in electron exchange-related functional groups, could be beneficial for enhancing MPD in sediments.”

[Comment 1-5] 4. *The manuscript could place greater emphasis on the new experimental findings obtained by the authors. Hypothetical conclusions and speculations should not be included in the abstract and conclusion sections. The abstract and conclusion should focus on the results and their direct implications, while speculative content should be placed in the discussion section.*

[Response] Thank you for your insightful comments. We fully agree that the abstract and conclusion should focus on the core results and their direct implications. In response, we will revise the abstract and conclusion sections accordingly, removing any hypothetical conclusions and speculations to ensure that these parts of the manuscript are concise and factual.

[Lines 24-36] “Eutrophication is an emerging global issue that is becoming increasingly severe due to the rising nutrient inputs and insufficient availability of electron donors for nitrogen removal. In sediments where redox conditions fluctuate dramatically, extracellular electron transfer (EET) critically supports microbial metabolism. However, the biogeochemical significance of EET-coupled denitrification and its EET mechanisms remains unclear. Here, through field investigations and laboratory ^{15}N isotope experiments, we reveal that humic substance (HS)-mediated electron shuttling significantly enhances denitrification primarily by stimulating bacterial outer membrane c -type cytochrome. Specifically, EET mitigates the emission of greenhouse gas nitrous oxide by enriching *nosZII*-type reducers. Notably, the efficacy of exogenous HS amendment attenuates in sediment with high native HS concentration. Metagenomic binning further revealed multiple cytochromes forming a complete EET-coupled denitrification electron transport chain. Our findings shed light on the microbial mechanisms underlying electron shuttling-driven denitrification in lakes, adding a new dimension to the understanding of biogeochemical cycles.”

[Lines 300-304] “Our findings demonstrate that HS-mediated EET, primarily regulated by *omcB*, significantly enhances biological nitrogen removal in lakes, contributing to eutrophication mitigation. Furthermore, this electron shuttling process promotes N_2O reduction (mainly regulated by *nosZII*), mitigating anthropogenically induced global warming.”

[Lines 307-311] “Since the nitrogen removal enhancement effect of HS amendments diminishes with increasing native HS concentrations in sediment, environmental management should prioritize applying electron shuttle amendments in polluted aquatic systems with low native HS levels to maximize cost-effectiveness. Nevertheless, the specific application methods and efficacy of these approaches remain to be elucidated.”

Reviewer #2 (Remarks to the Author):

Electron shuttling mitigates eutrophication and nitrous oxide emissions in lakes

[Comment 2-1] *The authors of the study “Electron shuttling mitigates eutrophication and nitrous oxide emissions in lakes” analyzed samples from 12 sites of a freshwater lake in China in terms of the use of external electron shuttles to support denitrification. They demonstrate that the potential for full denitrification increases with the amount of humic substances and MPD and that this is caused by a strongly significant correlation of denitrification genes and EET related genes. Based on MAGs they further indicate that this coupling might take place between different bacterial/archaeal strains. The overall research question is interesting and relevant and the authors nicely combined molecular and in situ measurements with ¹⁵N incubation measurements. There is certainly potential in the data and study and the manuscript was generally easy to read and the figures easy to understand. I very much appreciate the provision of the method related sketches, but very often the description of sampling and measurements is not very detailed and it is hard to repeat this study based on the description given (see comments below).*

[Response] We sincerely appreciate your positive and constructive comments on our manuscript. We are glad that the research question and the integration of different methodologies were found interesting and relevant. We have carefully reviewed the comments and have made detailed revisions to address the concerns regarding the description of sampling and measurements. Specifically, we have further elaborated on the sampling and measurement processes to provide more clarity and detail. Additionally, we have included more method-related sketches to better illustrate the procedures. We believe these revisions will enhance the reproducibility of our study. Thank you again for your valuable feedback.

However, the manuscript has some major limitations:

[Comment 2-2] *1. The conclusions of the study solely relies on correlative analysis of gene abundance/presence and gas emission and HS/MPD data. There is no proof of principal that or which microbes were really active. The ¹⁵N incubation experiment would have been an excellent target to measure active microbial populations to better understand the mechanism behind the observation.*

[Response] We agree with your comment and have re-conducted the ¹⁵N incubation experiment. Using a paired experimental design, we compared denitrification rates and N₂O accumulation levels between the blank group (no HS addition) and the HS_C groups (with commercial HS addition). To measure the active microbial populations, we analyzed the expression levels of relevant functional genes via reverse transcription qPCR (RT-qPCR) to clarify how HS addition influences denitrification, N₂O reduction, and extracellular electron transfer (EET) processes. These supplementary analyses have

provided stronger mechanistic insights into the observed phenomena by linking microbial activity measurements with molecular-level functional assessments.

[Lines 168-170] “Furthermore, the addition of HS in sediment (HS_C group) significantly increased the expression level ratio (HS_C/Blank) of the *omcB* gene by 31.5 times, notably higher than the increases observed for *omcS* (2.0 times) and *pilA* (3.3 times) genes (Fig. 4e).”

[Lines 184-186] “When comparing sediment gene expression between HS_C (HS-added) and Blank (no HS addition), the relative expression ratio (HS_C/Blank) of the *nosZII* gene (31.5-fold) was 4.1-fold higher than that of the *nosZI* gene (7.7-fold) (Fig. 4d).”

[Lines 188-190] “The heightened abundance and expression of *nosZII* gene are particularly effective at mitigating N₂O emissions, as *nosZII*-type reducers outcompete *nosZI*-type reducers in utilizing low N₂O concentrations⁴⁶. These findings suggested that EET can lower N₂O emissions by enhancing N₂O reduction, specifically via *nosZII*-type N₂O-reducers (Fig. 1d; Fig. 3b).”

[Lines 208-219] “Bacterial biomass exhibited no significant difference during the 24-h incubation (Supplementary Fig. S10). To characterize active microbial processes, we therefore further analyzed the effects of HS addition on functional gene expression (Fig. 4d-g). At the initial time point (0 h), relative expression levels of these functional genes showed no significant differences (Supplementary Table S8). By 24 h, however, the ratios for all functional gene expression levels (HS_C/Blank) were notably higher than 1 (ranging from 4.6- to 396.1-fold), indicating that HS addition broadly enhanced gene expression associated with denitrification, N₂O reduction, and EET (Fig. 4d-e). Notably, the high ratio of N₂O-reducing genes (*nosZI*+*nosZII*) to denitrifying genes (*nirS*+*nirK*), quantified as *nos/nir* (Fig. 4d), highlighted the preferential upregulation of N₂O reduction pathways. These findings align with prior studies showing that FA upregulates denitrification genes, particularly *nosZ*, in the model denitrifier *Paracoccus denitrificans*⁵². Our results demonstrate that HS-enhanced EET preferentially improves N₂O reduction activity, thereby accelerating denitrification rates while reducing N₂O production.”

[Lines 392-433] “3.6 Effects of external HS addition on electron transfer capacity, denitrification rate, N₂O accumulation level, and related functional genes expression

To examine the impact of external HS addition on electron transfer capacity, we designed a paired experiment with two groups: a blank group (no added HS) and a HS_C group (added commercial HS). Commercial HS (Macklin, Shanghai Macklin Biochemical Technology Co., Ltd., China) was added to the HS_C group at a rate of 20 mg-HS g⁻¹ sediment. Eight biological replicates were used, comprising three L-MPD samples (WX_2, WX_3, WY_1) and all M-MPD and H-MPD samples (Supplementary Table S1).

To measure electron transfer capacity, the anode chamber (working volume: 120 mL) of the MFC was inoculated with 15 g of homogenized sediment. The measurement protocol followed the methods described above. To investigate the effects of HS addition on denitrification rates, N₂O accumulation concentration, and functional gene expression, a paired experimental design was implemented subsequently (Supplementary Fig. S15). For each treatment, approximately 2 g of homogenized sediment and 11 mL 0.17 mM ¹⁵NO₃⁻ solution were placed in 12 mL Exetainer vial (039W, Labco, UK) and incubated in the dark at 25°C. The ¹⁵NO₃⁻ solution was purified with helium (99.999% purity) for 45 minutes to removal dissolved N₂ and N₂O before addition. Denitrification and anammox rates (*n* = 8 biological replicates) were determined using a ¹⁵N-tracer assay. Specifically, at each treatment and time point (0, 6, 12, 24h), 100 μL of saturated ZnCl₂ solution was added to terminate the incubation process, with three replicates sacrificed for each time point. The ¹⁵N-labeled N₂ (²⁹N₂ and ³⁰N₂) was

analyzed using membrane inlet mass spectrometry (MIMS 200, Bay Instruments, USA). The production rates of $^{29}\text{N}_2$ and $^{30}\text{N}_2$ were calculated based on the slope of linear regression (Supplementary Fig. S16-S17). The N_2 production rates via denitrification and anammox were calculated based on previously reported ⁷¹ (See calculation methods in Supplementary methods S4). The supernatant was also collected for measurement of accumulated nitrite concentration. Additionally, sediments in the Exetainer vials were immediately stored at $-20\text{ }^\circ\text{C}$ until DNA extraction using the DNeasy PowerSoil Kit (Qiagen, Hilden, Germany) ⁴⁵. To quantify total bacterial biomass variation, real-time qPCR was performed at 0 h and 24 h. The primers used for real-time qPCR are listed in Supplementary Table S11, and further details are provided in Supplementary Methods S5. To analyze N_2O accumulation levels during denitrification ($n = 8$ biological replicates), 5 mL of overlying water was sampled from each vial using a 20 mL sterile syringe equipped with a three-way valve (Supplementary Fig. S15). The syringe was then flushed with 10 mL of helium gas and shaken for 5 minutes to equilibrate the dissolved N_2O between the liquid (5 mL) and headspace (10 mL) phases. Dissolved N_2O concentrations were determined using the headspace equilibrium method. Concurrently, sediment samples collected from the vials were immediately frozen at $-80\text{ }^\circ\text{C}$. Within 72 h, total bacterial RNA was extracted using the OMEGA Bacterial RNA Kit (Omega Bio-Tek, Norcross, GA, USA). The extracted RNA was reverse-transcribed into cDNA using the PrimeScriptTM RT reagent Kit (Takara Bio, Shiga, Japan). Real-time qPCR was subsequently performed using the Hieff[®] qPCR SYBR Green Master Mix (Yeasen, Shanghai, China) to quantify gene abundances of denitrification (*nirS* and *nirK*), N_2O reduction (*nosZI* and *nosZII*), and EET functional genes (*omcB*, *omcS*, and *pilA*). The 16S rRNA gene served as an internal reference gene for normalization. Primers for real-times qPCR are listed in Supplementary Table S11, and detailed protocols can be found in Supplementary Methods S5.”

Fig. 4. Effects of humic substance (HS) addition on extracellular electron transfer (EET)-mediated denitrification. **a** Impact of HS addition on maximum power density (MPD) across all groups and the low-MPD (L-MPD) group. **b** Influence of HS addition on denitrification rates and N₂O accumulation levels. **c** Nonlinear regression analysis (one-phase decay exponential model, GraphPad Prism v9.0.0) demonstrating the quantitative relationship between HS concentration and denitrification enhancement ratio (HS-C/Blank). **d** The ratio of relative expression levels of denitrifying genes, N₂O reduction genes, and *nos/nir*, i.e., (*nosZI*+*nosZII*)/(*nirS*+*nirK*), in HS_C groups compared to Blank groups (HS_C/Blank). **e** The ratio of relative expression levels of EET genes (*omcB*, *omcS*, *pilA*) in HS_C groups compared to Blank groups (HS_C/Blank). **f** Log-transformed relative expression levels of denitrifying genes (*nirS* and *nirK*) and N₂O reduction genes (*nosZI*, and *nosZII*). **g** Log-transformed relative expression levels of EET genes (*omcB*, *omcS*, and *pilA*). All gene expression levels were normalized against 16S rRNA levels. Additional statistical information is provided in Supplementary Table S13. Relative expression levels of *nirS* were compared across all samples, while *nirK*, *nosZI*, *nosZII*, *omcB*, *omcS*, and *pilA* were analyzed specifically in the L-MPD subgroup. In violin plots, horizontal bars indicate medians, while dotted lines represent 25th and 75th percentiles.

[Comment 2-3] 2. The authors measured several things which were not included in discussion or results at all, for example, the Annamox measurement (where I have more concerns outlined below)

and nutrient, oxygen data etc. obtained from water and sediment. Is it possible that another factor than HS/MPD drives gas emissions like nitrate availability or oxygen concentration?

[Response] The denitrification and anammox rates were measured to confirm that denitrification is the dominant N₂ production process in lake sediments [Lines 199-200]. The methods for calculating denitrification and anammox rates using ¹⁵N isotope techniques were supplemented in Supplementary Method S4. We have added sequential statistical analyses to identify key drivers of denitrification. Specifically, the relative influence of these factors was determined using standardized coefficients in a stepwise multiple linear regression model. Additionally, including predictors other than MPD did not significantly improve the model's goodness-of-fit for dissolved N₂ or ΔN₂ concentrations ($p > 0.05$). The results confirmed MPD as the key environmental factor driving N₂ emissions.

[Lines 113-128] “The physicochemical characteristics of overlying water and sediment are shown in Supplementary Table S2. Inorganic nitrogen concentrations in the water column aligned with prior Lake Taihu studies, while sediment inorganic nitrogen concentrations and dissolved N₂ concentration exceeded reported ranges ^{30, 31, 32, 33}. The relationships between environmental factors and *in situ* denitrification performance are shown in Supplementary Fig. S4. Spearman correlation analysis revealed that water-column dissolved oxygen and pH, as well as sediment NO₃⁻-N, NH₄⁺-N, MPD, and HA, exhibited significant positive correlations with dissolved N₂ or ΔN₂ concentrations ($p < 0.05$) (Fig. 1d; Supplementary Fig. S4). Linear regression analysis further confirmed that MPD exhibited the strongest explanatory power for variations in dissolved N₂ and ΔN₂ concentrations ($R^2 \geq 0.7926$) (Supplementary Fig. S5). Furthermore, stepwise multiple linear regression analysis identified MPD as the only significant predictor for both dissolved N₂ ($p < 0.05$) and ΔN₂ ($p < 0.01$) concentrations (Supplementary Table S3-S4). The standardized coefficient of MPD was significantly higher than those of other environmental factors (Fig. 1c). Additionally, including predictors other than MPD did not significantly improve the model's goodness-of-fit for dissolved N₂ or ΔN₂ concentrations ($p \geq 0.05$) (Supplementary Table S5-S6). Hence, MPD was the key environmental factor driving N₂ emissions.”

[Lines 476-486] “We quantified key environmental factors influencing *in situ* denitrification potential (dissolved N₂ and ΔN₂ concentration) through sequential statistical analyses. Spearman correlation identified factors significantly associated with denitrification potential, followed by simple linear regression. Variables meeting residual assumptions (homoscedasticity, normality) after transformation were used in stepwise multiple linear regression (bidirectional Akaike Information Criterion optimization). The optimal model was validated by screening multicollinearity (variance inflation factor < 5), re-testing residuals, and assessing predictor significance. Relative influence of factors was determined using standardized coefficients, nested model comparisons (ANOVA), and visualization (‘MASS’, ‘lmtest’, ‘car’, and ‘ggplot2’ packages in R software, version 4.0.4). Detailed code analytical procedures are provided in Supplementary Methods S7. Linear and exponential regression analyses were performed using GraphPad Prism (version 9.0.0).”

Fig. 1 | Sampling sites, characteristics of electron shuttles, and *in situ* nitrogen removal performance. **a** Distribution of sampling sites (red stars, $n = 12$ biological replicates). The pie charts indicate the concentrations and proportions of dissolved electron shuttles, i.e., humic acid (HA) and fulvic acid (FA). The size of each pie chart is proportional to the concentration of humic substances (HS), while the color represents the excess N_2 (ΔN_2) concentration at the respective sites. **b** Linear regression analyses depicting the relationships between the concentrations of HS (HA + FA) and the electron transfer capacity, represented by maximum power density (MPD). Light gray shading with red borders indicates 95% confidence intervals. **c** Standardized coefficients of environmental factors derived from multiple linear regression model predicting dissolved N_2 concentration and ΔN_2 concentration. **d** Spearman correlations between the MPD of fresh sediments and the *in situ* emission characteristics of N_2 and N_2O . Blue and red circles denote positive and negative correlations, respectively, with color saturation representing correlation strength. $n = 12$ biological replicates. * and ** indicate p value < 0.05 and < 0.01 , respectively.

Supplementary Fig. S4 | Spearman correlation between the physicochemical characteristics and the *in situ* emission characteristics of nitrogen gas (N_2). In the correlation matrices, red and blue

dots correspond to the negative and positive correlations, respectively. $n = 12$ biological replicates. Light-colored dots represent low correlations, while dark-colored dots correspond to higher correlations. *, **, and *** represent significant level at $p < 0.05$, $p < 0.01$, and $p < 0.001$, respectively. Abbreviations: NO_3^- -N, nitrate nitrogen; NO_2^- -N, nitrite nitrogen; NH_4^+ -N, ammonia nitrogen; TN, total nitrogen; TP, total phosphorus; Chl-*a*, chlorophyll *a*; SPS, suspended sediment; TOC, total organic carbon; DO, dissolved oxygen; EC, electrical conductivity; ORP, Oxidation-reduction potential; HA, humic acid; FA, fulvic acid.

Supplementary Fig. S5 | Linear regression analyses of key environmental factors with N_2 gas emission characteristics. **a** Linear regression analyses of $\log(\text{NO}_3^-$ -N_S+1), $\log(\text{NH}_4^+$ -N_S), maximum power density (MPD), and humic acid (HA) concentrations with dissolved N_2 concentration. **b** Linear regression analyses of $10^{(-\text{pH})}$ in water, $\log(\text{NO}_3^-$ -N_S+1), $\log(\text{NH}_4^+$ -N_S), MPD, and HA concentrations with ΔN_2 concentration. $n = 12$ biological replicates. Light gray shading bordered in red represents the 95% confidence intervals for these regressions. The key physicochemical factors were identified according to the significant level ($p < 0.05$) in Spearman correlation analysis (Supplementary Fig. S3). Only linear regression with a p -value < 0.05 were shown.

Supplementary Table S3 | Coefficients of the stepwise multiple linear regression model for dissolved N_2 concentration.

Variable	Estimate	t value	Pr(> t)
MPD	1.16	2.745	0.0227*
$\log(\text{NO}_3^-$ _S+1)	49.54	2.183	0.0569

Adjusted $R^2 = 0.8343$, $p < 0.0002$. Homogeneity of variance (studentized Breusch-Pagan (BP) test): BP = 2.9972, $p = 0.2234$. Normality of residuals (Shapiro-Wilk normality test): W = 0.9422, $p = 0.5264$. * $p < 0.05$.

Supplementary Table S4 | Coefficients of the stepwise multiple linear regression model for ΔN_2 concentration.

Variable	Estimate	t value	Pr(> t)
MPD	1.66	4.546	0.0039**
DO	10.28	2.090	0.0816
$\log(\text{NO}_3^-$ _S+1)	49.34	2.364	0.0560
$10^{(-\text{pH})}$	2.17×10^9	1.199	0.2756
$\log(\text{NH}_4^+)$	-10.81	-1.060	0.3299

Adjusted $R^2 = 0.8978$, $p < 0.0011$. Homogeneity of variance (studentized Breusch-Pagan (BP) test): $BP = 4.7903$, $p = 0.4420$. Normality of residuals (Shapiro-Wilk normality test): $W = 0.9032$, $p = 0.1746$. $**p < 0.01$.

Supplementary Table S5 | Comparison of prediction model goodness-of-fit for dissolved N_2 concentration using ANOVA.

Model	Res.Df	RSS	Df	Sum of Sq	F	Pr(>F)
1	10	2.2811				
2	9	1.4914	1	0.7897	4.7655	0.0569

Model 1: $\text{scale}(\text{dissolved } N_2) \sim \text{scale}(\text{MPD})$; Model 2: $\text{scale}(\text{dissolved } N_2) \sim \text{scale}(\text{MPD}) + \text{scale}(\log(\text{NO}_3^- \text{ }_S+1))$. Environmental factors and ΔN_2 concentrations were standardized (mean = 0, standard deviation = 1) using the scale function in R software. The ANOVA comparison indicated no significant improvement in goodness-of-fit for Model 2 versus Model 1 ($\text{Pr}(>F) > 0.05$). Res.Df, Residual degrees of freedom; RSS, Residual sum of squares; Df, Degrees of freedom; Sum of Sq, Sum of squares; $\text{Pr}(>F)$, Probability of F-statistic.

Supplementary Table S6 | Comparison of prediction model goodness-of-fit for ΔN_2 concentration using ANOVA.

Model	Res.Df	RSS	Df	Sum of Sq	F	Pr(>F)
1	10	1.9643				
2	6	0.6134	4	1.3509	3.3035	0.0933

Model 1: $\text{scale}(\Delta N_2) \sim \text{scale}(\text{MPD})$; Model 2: $\text{scale}(\Delta N_2) \sim \text{scale}(\text{MPD}) + \text{scale}(\text{DO}) + \text{scale}(\log(\text{NO}_3^- \text{ }_S+1)) + \text{scale}(10^{-(\text{pH})}) + \text{scale}(\log(\text{NH}_4^+))$. Environmental factors and ΔN_2 concentrations were standardized (mean = 0, standard deviation = 1) using the scale function in R software. The ANOVA comparison indicated no significant improvement in goodness-of-fit for Model 2 versus Model 1 ($\text{Pr}(>F) > 0.05$). Res.Df, Residual degrees of freedom; RSS, Residual sum of squares; Df, Degrees of freedom; Sum of Sq, Sum of squares; $\text{Pr}(>F)$, Probability of F-statistic.

[Comment 2-4] 3. Despite the relevance of N_2O emissions, I feel that potential nitrite accumulation is similarly important but was not considered in the study, although it seems to be measured for some instances.

[Response] We have supplemented the content related to toxic nitrite accumulation based on your suggestions.

[Lines 46-48] “However, incomplete denitrification also contributes significantly to the accumulation of toxic nitrite and the emission of nitrous oxide (N_2O), a potent greenhouse gas with a global warming potential approximately 310 times that of carbon dioxide^{5, 6, 7}.”

[Lines 271-280] “However, each MAG only contained genes mediating one or two reduction steps (Fig. 6)⁶⁰. Consequently, nitrite likely becomes the terminal denitrification product in specific MAGs and the denitrifier *Quatrionococcus australiensis* due to the absence of nitrite reduction genes⁵⁶ (Fig. 5-6). Supporting this, sediment nitrite concentration showed a significant positive correlation with both MPD and humic substance concentration (Supplementary Fig. S13). Furthermore, HS addition significantly increased nitrite accumulation in L-MPD sediments (Supplementary Fig. S13d). Consequently, the EET-coupled denitrification process might enhance the potential risk of toxic nitrite

accumulation in sediment. This risk should be carefully considered during ecological monitoring and application of EET for enhancing risk environmental bioremediation.”

Supplementary Fig. S13 | Effects of maximum power density (MPD) and humic substances on nitrite concentration in sediment. **a** Linear regression between sediment nitrite concentration and MPD. **b** Linear regression between sediment nitrite concentration and HA+FA concentration. **c** Linear regression between sediment nitrite concentration and HA concentration. **d** Temporal variation of nitrite concentration in supernatant during 24-h incubation for Blank (no humic substances addition) and HS_C (commercial humic substances added) groups. * $p < 0.05$. $n = 3$ biological replicates for L-MPD samples (WX_2, WX_3, WY_1).

[Comment 2-5] 4. For me it is not clear whether the positive correlation comes from general growth or enrichment of certain groups or increased activity. How was the overall increase of bacterial biomass compared to the targeted genes?

[Response] We appreciate this insightful inquiry. To distinguish whether the observed functional gene correlations arose from (a) overall community growth, (b) specific population enrichment, or (c) enhanced metabolic activity, we quantified 16S rRNA gene abundance as a proxy for total bacterial biomass. Linear regression revealed strong positive correlations ($R^2 > 0.5$, $p < 0.05$) between total bacterial abundance and abundances of *nirS*, *omcB*, and *omcS* genes (Supplementary Fig. S8), indicating that general community growth primarily drives the relationships between *nirS* and EET genes. To isolate specific inter-gene associations independent of biomass variation, we normalized functional gene abundances to 16S rRNA levels. This normalization maintained significant correlations between EET genes (*omcB* and *omcS*) and both denitrification genes ($R^2 \geq 0.6211$, $p < 0.05$) and N_2O reduction genes ($R^2 \geq 0.5204$, $p < 0.05$), demonstrating interactions beyond co-growth effects. Functionally validating these relationships, our ^{15}N isotope experiment confirmed humic substance (HS) enhance microbial activity. Quantitative assessments of denitrification rates, N_2O

accumulation levels, and gene expression responses to HS addition are detailed in our response to Comment [2-2].

[Lines 154-164] “The relationships between EET functional genes and denitrifying functional genes are shown in Fig. 3 and Supplementary Fig. S7. In this study, the absolute abundance of all EET functional genes exhibited positive correlations with denitrifying functional genes (i.e., *nirS* and *nirK*), particularly *omcB* ($R^2 \geq 0.6663$, $p < 0.05$) and *omcS* ($R^2 \geq 0.6818$, $p \leq 0.01$) (Supplementary Fig. S7). However, the absolute abundances of *nirS*, *omcB*, and *omcS* genes showed significant positive correlations with total bacterial abundance, characterized by high goodness-of-fit ($R^2 \geq 0.5304$, $p < 0.05$) (Supplementary Fig. S8). This suggests that the significant positive correlation between *nirS* and EET genes (*omcB* and *omcS*) (Fig. 3a) might stem from overall bacterial community growth in sediment. Hence, the linear correlation between 16S rRNA gene-standardized functional genes were further performed (Fig. 3). The results indicated all standardized EET genes were significantly positively correlated with denitrifying genes (Fig. 3a).”

[Lines 207-208] “Bacterial biomass exhibited no significant difference during the 24-h incubation (Supplementary Fig. S10).”

[Lines 449-452] “Real-time qPCR was employed to quantify the total bacteria, EET functional genes (*omcB*, *omcS*, and *pilA*), as well as N_2O -producing (*nirS* and *nirK*) and reducing (*nosZI* and *nosZII*) functional genes. The primers utilized for real-time qPCR are listed in Supplementary Table S11. See more details in Supplementary Methods S5.”

Fig. 3 | The relationship between the extracellular electron transfer (EET) process and the nitrogen removal process. a Linear regression between standardized EET genes and denitrifying genes. **b** Linear regression between standardized EET genes and N₂O reducing genes. **c** Linear regression between humic substances and EET gene *omcB*. **d** Exponential correlation between the EET functional gene and dissolved N₂ characteristics. All functional genes were standardized by 16S rRNA gene abundance. *n* = 12 biological replicates. The light gray shading with red borders indicates the 95% confidence intervals around the regression lines.

Supplementary Fig. S7 | Linear regressions between the absolute abundances of denitrification genes, N₂O reduction gene, and extracellular electron transfer genes. a Linear regression between EET genes (*omcB*, *omcS*, and *pilA*) and denitrifying genes (*nirS* and *nirK*). **b** Linear regression between EET genes and N₂O reducing gene (*nosZI*). Functional genes were log-transformed or sqrt-transformed to meet residual assumptions (homoscedasticity, normality). *n* = 12 biological replicates. The light gray shading with red borders indicates the 95% confidence intervals around the regression lines.

Supplementary Figure S8 | Linear regression between denitrification (*nirS* and *nirK*), N₂O reduction (*nosZI* and *nosZII*), extracellular electron transfer (*omcB*, *omcS*, and *pilA*) functional genes and 16S rRNA gene. *n* = 12 biological replicates. Light gray shading with red borders indicates 95% confidence intervals.

Supplementary Fig. S10 | Bacterial biomass variation during the 24 h incubation period in blank and HS_C group, respectively. $n = 8$ biological replicates. Blank group, no humic substance addition. HS_C group, commercial humic substance addition group.

Supplementary Table S11 | Primers used for Real-time PCR (qPCR) amplification in this study.

Target genes	Primers	Sequence (5'-3')	Gene size (bp)	Annealing temperature (°C)	Supplementary References
omcB	8912	CCCACTTCGACAACACTATTCG	212	58	10
	8908-2	GGTCAGCAGGCCACCGG			
omcS	omcS-F	CCATCAAGAACAGCGGTTCC	133	60	11
	omcS-R	TGGTTGGCGAAGGCATAGG			
pilA	pilA-F	ATCGGTATTCTCGCTGCAAT	116	60	11
	pilA-R	AATGCGGACTCAAGAGCAGT			
nirS	cd3aF	G TSAACG TSAAGGARACSGG	426	57	12
	R3cd	GASTTCGGRTGSGTCTTGA			
nirK	F1aCu	ATCATGGTSCTGCCGCG	473	55	13
	R3Cu	GCCTCGATCAGRTTGTGGTT			
nosZI	nosZ2F	CGCRACGGCAASAAGGTSMSSGT	276	60	14
	nosZ2R	CAKRTGCAKSGCRTGGCAGAA			
nosZII	nosZ-II-F	CTIGGICCIYTKCAYAC	690	54	15
	nosZ-II-R	GCIGARCARAAITCBGTRC			
Bacterial 16S rRNA	16SrRNA-F	CGGTGAATACGTTTCYCGG	142	55	16
	16SrRNA-R	GGHTACCTTGTTACGACTT			

[Comment 2-6] 5. In the results and discussion part the comparison to existing data is too superficial, to which extent are the data transferable, are your values comparable to other or not. What is the true new and novel piece of data compared to other studies as the link between HS and denitrification was already shown previously.

[Response] To address data comparability with existing studies, we supplemented detailed comparisons of key parameters, including humic substance (HS) concentration, maximum power density (MPD) range, physicochemical properties, and dissolved N₂ and ΔN₂ concentrations. These analyses confirm that our environmental variables either align with established ranges in prior research or expand the coverage of critical factors, demonstrating broader representativeness of our findings. Crucially, while previous work established links between HS and denitrification, to the best of our knowledge, no studies have investigated EET-coupled denitrification mechanisms in natural lake sediments. To strengthen the foundation for these novel insights, we benchmarked our sediment-based results against laboratory data from pure cultures.

Furthermore, we have revised the original text to emphasize the contributions of this study to the following four significant advances in understanding EET-coupled denitrification:

- (1) We demonstrate that cooperative interaction between HS and EET bacteria, specifically mediated by *omcB*-associated outer membrane c-type cytochromes, act as the primary environmental driver of lake sediment denitrification. Crucially, the resulting EET capacity (measured as MPD) serves as the key determining factor [Lines 164-179].
- (2) While heightened denitrification elevates N₂O release risks, our results confirm that EET upregulates both the abundance of *nosZII*-type N₂O reducers and their functional gene expression. This mechanism effectively reduces net N₂O emissions [Lines 180-192; 292-296].
- (3) Exogenous electron shuttle amendments lose effectiveness in sediments where native HS concentrations exceed 50 mg g⁻¹. This finding provides actionable guidance for optimizing electron shuttle applications in aquatic ecosystem remediation. [Lines 219-228]
- (4) Using metagenome-assembled genomes, we reconstructed complete electron transfer pathways from EET bacteria to denitrifiers and identified the key enzymes involved. [Section 2.7]

[Lines 93-98] “By establishing sampling sites across different land-use areas and submerged plant zones, we successfully obtained a wide range of HS concentrations (9.5~230.1 mg g⁻¹ sediment) (Fig.1b; Supplementary Table S1). This range encompasses the previously reported HS concentrations in shallow, eutrophic lakes (14.5 ~ 78.5 mg g⁻¹ sediment)²¹. HA was the predominant form of HS, comprising 92.8±9.4% of the total dissolved HS (Fig. 1a), which was consist with the HS composition in other lakes²¹.”

[Lines 101-107] “The MPD values spanned a wide range (0.05 ~ 76.65 μW m⁻²), comparable to those measured in aquaculture pond sediment MFC (52.2 ~ 99.5 μW m⁻²)²⁴ and plant MFC (0.28 ~ 37.0 μW m⁻²)^{25,26}. However, these values were significantly lower than those reported for lake Taihu sediment MFC (3150 ~ 5700 μW m⁻²)^{27,28,29}, which were attribute to the formation of mature biofilm on anode surfaces after 15 to 60 days of cultivation in prior studies. Thus, the MPD measured in the present study (within 1 h after sediment addition) better reflects *in situ* characteristics.”

[Lines 113-116] “The physicochemical characteristics of overlying water and sediment are shown in Supplementary Table S2. Inorganic nitrogen concentrations in the water column aligned with prior Lake Taihu studies, while sediment inorganic nitrogen concentrations and dissolved N₂ concentration exceeded reported ranges^{30,31,32,33}.”

[Lines 200-202] “These results align with prior studies showing that electron shuttles like HA accelerate electron transfer and boost denitrification performance in wastewater treatment systems^{48,49,50,51}.”

[Lines 202-205] “Importantly, after 24 h incubation, HS addition reduced accumulated N₂O concentration by 78.3% compared to the Blank group ($p = 0.0156$). Similar effects have been observed with FA, which suppressed N₂O generation by 41.3%~98.8% in *Paracoccus denitrificans*-mediated denitrification⁵².”

[Lines 215-217] “These findings align with prior studies showing that FA upregulates denitrification genes, particularly *nosZ*, in the model denitrifier *Paracoccus denitrificans*⁵².”

[Lines 272-277] “Consequently, nitrite likely becomes the terminal denitrification product in specific MAGs and the denitrifier *Quatrionicoccus australiensis* due to the absence of nitrite reduction genes⁵⁶ (Fig. 5-6). Supporting this, sediment nitrite concentration showed a significant positive correlation with both MPD and humic substance concentration (Supplementary Fig. S13). Furthermore, HS addition significantly increased nitrite accumulation in L-MPD sediments (Supplementary Fig. S13d).”

Specific comments:

Abstract:

[Comment 2-7] *It is generally not clear how you performed the study, which methods did you use, where did you the samples derive from (lake sampling or lab experiment, or both?)*

[Response] Thank you for your valuable comments. We have revised the abstract to clearly state that both lake sampling and indoor ¹⁵N isotope experiments were conducted.

[Lines 28-30] “Here, through field investigations and laboratory ¹⁵N isotope experiments, we reveal that humic substance (HS)-mediated electron shuttling significantly enhances denitrification primarily by stimulating bacterial outer membrane *c*-type cytochrome.”

[Comment 2-8] L25: *Electron donors you mean.*

[Response] Changed “electrons” to “electron donors”.

[Lines 24-25] “Eutrophication is an emerging global issue that is becoming increasingly severe due to the rising nutrient inputs but insufficient availability of electron donors for nitrogen removal.”

[Comment 2-9] L29: *Write “in situ” in italics*

[Response] The term “*in situ*” has been italicized throughout the text.

[Comment 2-10] L30: *HS is not explained before.*

[Response] The full name of HS, “humic substance,” is provided for the first time.

[Lines 28-30] “Here, through field investigations and laboratory ¹⁵N isotope experiments, we reveal that humic substance (HS)-mediated electron shuttling significantly enhances denitrification primarily by stimulating bacterial outer membrane *c*-type cytochrome.”

Introduction:

[Comment 2-11] L48ff: *Beside the production of N₂O there is also a certain risk of the accumulation of toxic nitrite during denitrification, which is similarly important and dangerous.*

[Response] The risk of the accumulation of toxic nitrite during denitrification have been added.

[Lines 46-48] “However, incomplete denitrification also contributes significantly to the accumulation of toxic nitrite and the emission of nitrous oxide (N₂O), a potent greenhouse gas with a global warming potential approximately 310 times that of carbon dioxide^{5, 6, 7.}”

[Lines 272-281] “However, each MAG only contained genes mediating one or two reduction steps (Fig. 6)⁶⁰. Consequently, nitrite likely becomes the terminal denitrification product in specific MAGs and the denitrifier *Quatrionicoccus australiensis* due to the absence of nitrite reduction genes⁵⁶ (Fig. 5-6). Supporting this, sediment nitrite concentration showed a significant positive correlation with both MPD and humic substance concentration (Supplementary Fig. S13). Furthermore, HS addition significantly increased nitrite accumulation in L-MPD sediments (Supplementary Fig. S13d). Consequently, the EET-coupled denitrification process might enhance the potential risk of toxic nitrite accumulation in sediment. This risk should be carefully considered during ecological monitoring and application of EET for enhancing environmental bioremediation.”

[Comment 2-12] L55: *What do you mean with “cable” bacteria? Is that a common term, if so add reference or explain better?*

[Response] We have clarified the term "cable bacteria" and provide additional context and references.

[Lines 54-56] “Similarly, in sediments, cable bacteria, which are filamentous microorganisms capable of conducting electricity over long distances, enable electron transfer across distances of 1-2 cm in the suboxic zone^{9, 10.}”

[Comment 2-13] L61: *It is not clear what you mean with “electron acceptors” as it stand it sounds like a protein/enzyme but then you refer to bacteria later in the sentence. Please be more precise and add the type of acceptor you refer to e.g. electron accepting bacteria. Then you refer to HS in the next sentence. This passage needs clarification.*

[Response] We appreciate your insightful comment regarding the clarity of the term “electron acceptors” in our manuscript. In this section, we aim to describe the extracellular electron transfer (EET) process mediated by electron shuttles. Specifically, we introduce humic substances (HS) as the primary electron acceptors in sediments. To address the ambiguity noted in your comment, we have removed the phrase “, such as extracellular electron-uptake denitrifiers”. This revision ensures focus remains on HS as the key electron acceptors discussed in this context.

[Lines 59-63] “In EET mediated by electron shuttles, electrons produced through intracellular metabolic processes are transported to extracellular electron acceptors via redox-active electron shuttles¹¹. Among these electron shuttles, humic substances (HS) stand out as the most abundant natural electron shuttles, forming 50%-80% of natural organic matter in water, soil, and sediments^{12, 13, 14.}”

[Comment 2-14] L86: *Is “elctroactive” the correct term? Maybe better “involved in EET”.*

[Response] Thank you for your attention to detail regarding the terminology used in our manuscript. We agree that the term “electroactive bacteria” might introduce ambiguity, especially when discussing

specific pathways related to extracellular electron transfer (EET). To address this concern and ensure consistency throughout the manuscript, we have replaced “electroactive bacteria” with “EET bacteria”. [Lines 86-88] “Additionally, metagenomic binning revealed detailed electron transfer pathways from EET bacteria to HS and subsequently to denitrifiers, highlighting the extracellular electron uptake pathways utilized by denitrifiers.”

[Comment 2-15] L86: As it stand now, it sounds as if the process is done by two independent bacteria in cooperation, one doing the nitrate reduction and the other the HS oxidation. Do I get this correctly? In the manuscript you cited before (Coates et al) it seemed to be done by one microorganism. Please clarify.

[Response] In our study, we are indeed highlighting the complete electron shuttles mediated extracellular electron transfer (EET) process. Specifically, the EET bacteria donate electrons to oxidized humic substance (HS), reducing them to their reduced state. Subsequently, these reduced HS serve as electron donors for denitrifiers to carry out nitrate reduction, while the reduced HS are oxidized. It appears that the confusion arose because the process described by Coates et al. only encompasses the second step, where a single microorganism performs both nitrate reduction and HS oxidation. In contrast, our study involves two linked processes mediated by different microorganisms and different redox states of HS. To clarify this distinction and avoid any further ambiguity, we have revised the original sentence as follows:

[Lines 86-88] “Additionally, metagenomic binning revealed detailed electron transfer pathways from EET bacteria to HS and subsequently to denitrifiers, highlighting the extracellular electron uptake pathways utilized by denitrifiers.”

Methods:

[Comment 2-16] L251/Table S1: I do not really get your classification of high, medium low HS concentrations. For example, WX1 (Low HS) has higher values than DX2 (medium)

[Response] Thank you for your comment. We apologize for the confusion caused by the description in manuscript and the table header in Supplementary Table S1. Our classification of samples into low, medium and high levels was not based on humic substances (HS) concentration, but rather on the measured electron transfer capacity, i.e., maximum power density (MPD). The grouping in Supplementary Table S1 was primarily based on the distinct clustering of samples according to MPD and ΔN_2 concentrations (Supplementary Table S7). The MPD is determined by both extracellular electron transfer (EET) activity and electron shuttles (e.g., HS) concentration. Although HS concentrations generally correlate positively with MPD (Fig. 1), there are instances where samples with low MPD have higher HS concentrations than those with medium MPD. This is due to the combined influence of EET activity and HS on MPD. We have revised the text and Supplementary Table S1; S7 to clarify this distinction and avoid further misunderstandings. Thank you for bringing this to our attention.

[Lines 137-141] “Given that MPD significantly affects *in situ* denitrification potential (Fig. 1c), the samples were categorized into three representative groups to analyze the functional group characteristics of HS under varying MPD conditions (L-MPD: $4.1 \pm 6.1 \mu W m^{-2}$, M-MPD: $44.0 \pm 3.5 \mu W m^{-2}$, and H-MPD: $73.3 \pm 3.6 \mu W m^{-2}$) (Supplementary Table S7).”

[Lines 314-317] “To examine sediments with varying MPD gradients, twelve sampling sites were selected across four influent rivers influenced by urban and agricultural land use (Fig. 1). These sites spanned both submerged and non-submerged vegetation zones (E120°0’31”-E120°24’19”, N31°26’34”-N31°33’68”) (Fig. 1a; Supplementary Table S10).”

Supplementary Table S1 | Maximum power density (MPD) levels and dissolved humic substance concentrations of sediment in lake Taihu.

MPD Levels	Sampling sites	MPD ($\mu\text{W m}^{-2}$)	HA concentration (mg g^{-1} sediment)	FA concentration (mg g^{-1} sediment)
L-MPD	WX_1	0.06±0.01	46.87	2.47
	WX_2	0.09±0.03	6.12	3.41
	WX_3	0.29±0.02	47.52	4.01
	WY_1	0.56±0.01	24.10	0.21
	WY_2	1.12±0.02	23.46	0.86
	WY_3	15.63±0.40	113.07	2.82
	DX_1	10.70±0.49	33.74	1.05
M-MPD	DX_2	41.69±0.63	34.43	2.43
	LX_3	48.44±1.12	133.52	14.36
	LX_4	41.97±2.21	67.44	1.30
H-MPD	LX_1	70.40±2.77	217.53	12.59
	LX_2	76.13±0.53	207.11	9.37

Abbreviations: L-MPD, Low-MPD; M-MPD, medium-MPD; H-MPD, high-MPD; HA, humic acid; FA, fluvic acid; WX, Wuxi river; WY, Wangyu river; DX, Daixi river; LX, liangxi river.

Supplementary Table S7 | In situ denitrification potential across sampling sites categorized by maximum power density (MPD).

Groups	MPD ($\mu\text{W m}^{-2}$)	Dissolved N_2 concentration ($\mu\text{mol L}^{-1}$)	ΔN_2 concentration ($\mu\text{mol L}^{-1}$)
L-MPD	4.1±6.1	524.0±22.3	5.9±16.2
M-MPD	44.0±3.5	584.2±30.9	65.0±28.1
H-MPD	73.3±3.6	671.1±26.3	151.3±24.6

The detailed sample names for each group are provided in Supplementary Table S1. Abbreviations: L-MPD, Low-MPD; M-MPD, medium-MPD; H-MPD, high-MPD; ΔN_2 , excess dissolved N_2 .

[Comment 2-17] L253: As I understood, samples were taken in influent areas, thus I assume water movement was quite substantial. Sampling took place ones in 2024 for sediments and water samples. I wonder, how representative the samples can be if taken only ones. Could the measured values be a snapshot of a recent event happening around the river area like harvest or fertilization etc.

[Response] Thank you for your insightful comment regarding the representativeness of our sampling strategy. We acknowledge concerns that single-time sampling might be influenced by transient events like fertilization or harvesting. To address these concerns systematically: First, our late-April sampling was timed to coincide with the wheat maturity stage (late April-May) in Taihu Lake’s summer rice and winter wheat double-crop rotation system (Qiao et al., 2012). During this phase, diminished root absorption capacity and risks of delayed maturity lead farmers to avoid large-scale nitrogen

fertilization. Critically, April's dry season inherently minimizes rainfall-driven nitrogen loss (Qiao et al., 2012; Wang et al., 2019), an effect further reinforced by our sunny-day sampling protocol. Second, spatial diversity was ensured through twelve sampling sites across four influent rivers, covering submerged/non-submerged vegetation zones and diverse land uses (urban/agricultural). This design captures hydrological variability, supported by $|Z\text{-scores}| < 3$ for nitrogen nutrients (indicating no statistical outliers). In total, our sampling confirming temporal representativeness despite single-time sampling.

[Lines 319-326] “The region has long practiced a summer rice-winter wheat double-crop rotation system⁶³. Field investigations and sample collections were conducted in late April 2024, coinciding with the wheat maturity stage. During this phase, farmers typically avoid large-scale nitrogen fertilization due to diminished crop root absorption capacity and risks of delayed maturity. While rainfall is a primary driver of nitrogen loss from farmland, April is a dry season with minimal precipitation in the region^{63, 64}. Sampling was conducted on sunny days to avoid short-term drastic changes in nitrogen nutrient concentrations, with $|Z\text{-score}| < 3$ confirming no statistical outliers (Supplementary Table S2).”

[Comment 2-18] L254: *Did you also sample 3 sediment samples per site? If not I wonder how replication looks like and assume that t-test are not a suitable statistical method.*

[Response] Your insightful comment raises a valid concern regarding the data analysis approach. Specifically, only one sediment sample was collected per site, which initially prompted us to reevaluate our analytical strategy. To address this, we reanalyzed the data using mean values instead of all technical replicates. Consequently, *t*-test for field measurements are no longer applied and Supplementary Fig. S3 was revised to Supplementary Table S7. Furthermore, we implemented a paired experimental design for the ¹⁵N isotope experiment. First, we assessed whether data residuals followed a normal distribution; subsequent analyses were then conducted using paired *t*-tests or the Wilcoxon matched-pairs signed rank test, depending on the distribution results.

[Lines 327-332] “At each site, triplicate water samples were collected at an average sampling depth of approximately 30 cm below the surface using a 5 L water sampler (Wuhan Xuan Ming Yu Environmental Technology Development Co., Ltd., Wuhan, China). A single surface sediment (0-10 cm depth) was collected per site using a PBS-211 grab sampler (Wuhan Petersen Technology Co., Ltd., Wuhan, China). All samples were stored at 4 °C and transported to the laboratory within 72 h for immediate water quality analysis.”

[Lines 475-491] “The mean values of three technical replicates at each of the 12 sampling sites were used for statistical analysis ($n = 12$ biological replicates). We quantified key environmental factors influencing *in situ* denitrification potential (dissolved N₂ and ΔN_2 concentration) through sequential statistical analyses. Spearman correlation identified factors significantly associated with denitrification potential, followed by simple linear regression. Variables meeting residual assumptions (homoscedasticity, normality) after transformation were used in stepwise multiple linear regression (bidirectional Akaike Information Criterion optimization). The optimal model was validated by screening multicollinearity (variance inflation factor < 5), re-testing residuals, and assessing predictor significance. Relative influence of factors was determined using standardized coefficients, nested model comparisons (ANOVA), and visualization (*‘MASS’*, *‘lmtest’*, *‘car’*, and *‘ggplot2’* packages in

R software, version 4.0.4). Detailed code analytical procedures are provided in Supplementary Methods S7. Linear and exponential regression analyses were performed using GraphPad Prism (version 9.0.0). To test the effects of humic substance addition on MPD, denitrification rates, N₂O accumulation levels, microbial biomass, and relative expression levels of functional genes, we conducted paired analyses ($n = 8$ biological replicates). First, the Shapiro-Wilk test was used to examine the normality of the residuals of the paired samples. If the data met the normality assumptions, a paired t -test was applied; otherwise, the Wilcoxon matched-pairs signed rank test was used (GraphPad Prism) (Supplementary Table S13).”

Supplementary Table S7 | In situ denitrification potential across sampling sites categorized by maximum power density (MPD).

Groups	MPD ($\mu\text{W m}^{-2}$)	Dissolved N ₂ concentration ($\mu\text{mol L}^{-1}$)	ΔN_2 concentration ($\mu\text{mol L}^{-1}$)
L-MPD	4.1±6.1	524.0±22.3	5.9±16.2
M-MPD	44.0±3.5	584.2±30.9	65.0±28.1
H-MPD	73.3±3.6	671.1±26.3	151.3±24.6

The detailed sample names for each group are provided in Supplementary Table S1. Abbreviations: L-MPD, Low-MPD; M-MPD, medium-MPD; H-MPD, high-MPD; ΔN_2 , excess dissolved N₂.

Supplementary Table S13 | Statistical analysis information.

		Normality of Paired t -test				Wilcoxon matched-
		Residuals				pairs signed rank test
		Shapiro-Wilk	t	df	p value	p value
		(p value)			(two-tailed)	(two-tailed)
MPD	All	0.4848	0.9046	7	0.3957	-
	L-MPD	0.5246	9.163	2	0.0117	-
D_{total}	All	0.6130	4.510	7	0.0028	-
$C_{\text{N}_2\text{O}}$	All	0.0105	-	-	-	0.0156
nirS /16S	All	0.0770	3.341	7	0.0124	-
rRNA						
nirK /16S	All	0.2548	0.6325	7	0.5472	-
rRNA	L-MPD	0.2025	7.111	2	0.0192	-
nosZI /16S	All	0.1428	0.8431	7	0.4270	-
rRNA	L-MPD	0.0675	23.70	2	0.0018	-
nosZII /16S	All	0.1865	0.5225	7	0.6174	-
rRNA	L-MPD	0.0893	6.944	2	0.0201	-
omcB /16S	All	0.0634	0.9950	7	0.3529	-
rRAN	L-MPD	0.5600	11.14	2	0.0080	-
omcS /16S	All	0.4857	0.7625	7	0.4707	-
rRNA	L-MPD	0.4291	4.874	2	0.0396	-
pilA /16S	All	0.2729	0.9055	7	0.3953	-
rRNA	L-MPD	0.8322	6.843	2	0.0207	-

MPD, maximum power density; D_{total} , Denitrification rate; $C_{\text{N}_2\text{O}}$, Dissolved N₂O concentration; All, All groups ($n = 8$) included three biological replicates in Low-MPD (L-MPD) group, three biological replicates in medium-MPD and two biological replicates in high-MPD groups. L-MPD, low-MPD

group (WX_2, WX_3, and WY_1). Normality of residuals between paired samples (blank and HS-added) was assessed using the Shapiro-Wilk test. For metrics with normal residuals ($p > 0.05$), a paired t -test was used, while the Wilcoxon matched-pairs signed rank test was applied for non-normal residuals ($p < 0.05$). All analyses were performed using GraphPad Prism (Version 9.0.0).

[Comment 2-19] Section 3.3.: I don't really get the design of this measurements, there are several conflicting statements in this section. I think a scheme would be helpful. It is also not clear if that was done with sediments from all 12 locations, and how the proportion of target sediments and WX1 sediment was.

[Response] We have re-conducted the ^{15}N isotope experiment and provided a detailed workflow scheme. Eight biological replicates were used, comprising three L-MPD samples (WX_2, WX_3, WY_1) and all M-MPD and H-MPD samples (Supplementary Table S1). Crucially, humic substances (HS) were added at a consistent ratio of 20 mg-HS g⁻¹ sediment in both the MPD measurement experiment and the \$^{15}\text{N}\$ isotope experiment. However, because the reactor volumes differed between these two experiments, the absolute mass of sediment added to each reactor also varied accordingly. Furthermore, since the revised experiment measured accumulated N_2O levels and functional gene expression, we have moved this detailed description from Section 3.3 to Section 3.6 to enhance clarity and avoid ambiguity.

[Lines 391-405] “3.6 Effects of external HS addition on electron transfer capacity, denitrification rate, N_2O accumulation level, and related functional genes expression

To examine the impact of external HS addition on electron transfer capacity, we designed a paired experiment with two groups: a blank group (no added HS) and a HS_C group (added commercial HS). Commercial HS (Macklin, Shanghai Macklin Biochemical Technology Co., Ltd., China) was added to the HS_C group at a rate of 20 mg-HS g⁻¹ sediment. Eight biological replicates were used, comprising three L-MPD samples (WX_2, WX_3, WY_1) and all M-MPD and H-MPD samples (Supplementary Table S1).

To measure electron transfer capacity, the anode chamber (working volume: 120 mL) of the MFC was inoculated with 15 g of homogenized sediment. The measurement protocol followed the methods described above. To investigate the effects of HS addition on denitrification rates, N_2O accumulation concentration, and functional gene expression, a paired experimental design was implemented subsequently (Supplementary Fig. S15). For each treatment, approximately 2 g of homogenized sediment and 11 mL 0.17 mM \$^{15}\text{NO}_3^-\$ solution were placed in 12 mL Exetainer vial (039W, Labco, UK) and incubated in the dark at 25°C. ...”

Supplementary Fig. S15 | Workflow for investigating the effects of humic substance addition on denitrification rates, N_2O accumulation concentration, bacterial biomass variation, and active microbial processes. The denitrification rate was determined using ^{15}N isotope tracing method, while N_2O accumulation concentration was measured via the headspace equilibrium method. qPCR was conducted to quantify total bacterial biomass variation during the 24-hour incubation experiment. Reverse transcription quantitative PCR (RT-qPCR) was employed to assess the relative expression levels of three functional gene categories: (1) denitrification genes (*nirS*, *nirK*), (2) N_2O reduction genes (*nosZI*, *nosZII*), and (3) extracellular electron transfer (EET, including *omcB*, *omcS*, and *pilA*) genes. All functional gene expression levels were normalized against the 16S rRNA gene. $n = 8$ biological replicates, including three L-MPD samples (WX_2, WX_3, and WY_1) and all M-MPD and H-MPD samples (Supplementary Table S1).

[Comment 2-20] L272/267: These two lines contradict each other ones you said HS-C group received 300mg ones 40 mg of commercial HS.

[Response] Thank you for highlighting the potential confusion regarding the different amounts of commercial humic substances (HS) used in our experiments. As clarified in our response to [Comment 2-19], HS was added at a consistent ratio of 20 mg g⁻¹ sediment in both the MPD measurement experiment and the \$^{15}N\$ isotope experiment. Nevertheless, because the reactor volumes differed between these two experiments, the absolute mass of sediment—and consequently the absolute amount

of HS added—varied accordingly. To eliminate ambiguity, we have revised the text to uniformly describe the HS addition as “20 mg-HS g⁻¹ sediment” throughout, omitting reactor-specific absolute values.

[Lines 395-396] “Commercial HS (Macklin, Shanghai Macklin Biochemical Technology Co., Ltd., China) was added to the HS_C group at a rate of 20 mg-HS g⁻¹ sediment.”

[Lines 399-400] “To measure electron transfer capacity, the anode chamber (working volume: 120 mL) of the MFC was inoculated with 15 g of homogenized sediment.”

[Lines 403-405] “For each treatment, approximately 2 g of homogenized sediment and 11 mL 0.17 mM ¹⁵NO₃⁻ solution were placed in 12 mL Exetainer vial (039W, Labco, UK) and incubated in the dark at 25°C.”

[Comment 2-21] L269/273: *Ones you said 15g WX1 and ones 2g. I don't get the difference.*

[Response] As clarified in our response to [Comment 2-19] and [Comment 2-20], HS was added at a consistent ratio of 20 mg g⁻¹ sediment in both the MPD measurement experiment and the ¹⁵N isotope experiment. Nevertheless, because the reactor volumes differed between these two experiments, the absolute mass of sediment—and consequently the absolute amount of HS added—varied accordingly. We have rewritten this part of the experimental process to avoid confusion.

[Lines 400-401] “To measure electron transfer capacity, the anode chamber (working volume: 120 mL) of the MFC was inoculated with 15 g of homogenized sediment.”

[Lines 404-406] “For each treatment, approximately 2 g of homogenized sediment and 11 mL 0.17 mM ¹⁵NO₃⁻ solution were placed in 12 mL Exetainer vial (039W, Labco, UK) and incubated in the dark at 25°C.”

[Comment 2-22] L273: *Which tracer did you use? If you want to distinguish denitrification and annamox I assume you need to different ones, meaning also to separate incubations or?*

[Response] We exclusively used the ¹⁵NO₃⁻ tracer to distinguish denitrification and annamox rates. The detailed calculation method, adapted and refined from Thamdrup & Dalsgaard (2002), is provided in Supplementary Methods S4.

“S4. Calculation of denitrification and annamox rates

The denitrification rate (D_{total}) and annamox rate (A_{total}) were determined using the ¹⁵N-NO₃⁻ incubation method previously described with some modification¹. Specifically, the primary nitrogen species in the ¹⁵N-labeled incubation bottles were ¹⁴N-NH₄⁺, ¹⁴N-NO_x⁻, and ¹⁵N-NO_x⁻. The initial molar fraction (mol/mol) of ¹⁵N in NO_x⁻ was calculated using Equation (2):

$$F_N = n(^{15}\text{N-NO}_x^-) / [n(^{15}\text{N-NO}_x^-) + n(^{14}\text{N-NO}_x^-)] \quad (4)$$

For denitrification (D), assuming that NO_x⁻ is randomly utilized by denitrifiers, the isotopic composition of N₂ produced during denitrification is as follows:

Therefore, the production rates of ²⁸N₂ (D₂₈), ²⁹N₂ (D₂₉), and ³⁰N₂ (D₃₀) from denitrification are:

$$D_{28} = D_{\text{total}} \times (1 - F_N) \times (1 - F_N) \quad (9)$$

$$D_{29} = D_{\text{total}} \times (1-F_N) \times F_N + D_{\text{total}} \times F_N \times (1-F_N) \quad (10)$$

$$D_{30} = D_{\text{total}} \times F_N \times F_N \quad (11)$$

For anammox (A), the isotopic composition of N_2 produced is:

Thus, the production rates of $^{28}\text{N}_2$ (A_{28}) and $^{29}\text{N}_2$ (A_{29}) from anammox are:

$$A_{28} = A_{\text{total}} \times (1 - F_N) \quad (14)$$

$$A_{29} = A_{\text{total}} \times F_N \quad (15)$$

Based on the above derivation, only denitrification can produce $^{30}\text{N}_2$. The denitrification rate (D_{total}) can be derived from Equation (9) as:

$$D_{\text{total}} = D_{30}/(F_N \times F_N) \quad (16)$$

$$D_{30} = ^{30}\text{N}_2 \quad (17)$$

Therefore,

$$D_{\text{total}} = ^{30}\text{N}_2/(F_N \times F_N) \quad (18)$$

Further, the anammox rate (A_{total}) can be derived from Equation (13) as:

$$A_{\text{total}} = A_{29}/F_N \quad (19)$$

$$A_{29} = ^{29}\text{N}_2 - D_{29} \quad (20)$$

From Equation (8), we can derive:

$$D_{29} = D_{\text{total}} \times 2 \times (1-F_N) \times F_N \quad (21)$$

Combining Equations (15) - (18), we obtain:

$$A_{\text{total}} = \{^{29}\text{N}_2 - [^{30}\text{N}_2 \times 2 \times (1-F_N)]/F_N\}/F_N \quad (22)$$

Finally, based on $^{30}\text{N}_2$ and $^{29}\text{N}_2$ production rates (Supplementary Fig. S16-S17) and F_N in the culturing vials, the denitrification rate and anammox rate can be calculated using Equations (16) and (20), respectively.”

Supplementary References

1. Thamdrup B, Dalsgaard T. Production of N_2 through anaerobic ammonium oxidation coupled to nitrate reduction in marine sediments. *Appl. Environ. Microb.* **68**, 1312-1318 (2002).

[Comment 2-23] L276: Was the denitrification activity stable over time? If not, can you exclude microbial growth in HS treated vials, which then explain the increase in denitrification activity?

[Response] Overall, linear regression analysis of eight biological replicates demonstrated a strong fit ($R^2 > 0.8$) (Supplementary Fig. S16). Additionally, bacterial biomass, as determined by 16S rRNA gene abundance, exhibited no significant difference during the 24-h incubation (Supplementary Fig. S10). To further investigate the functional dynamics, RNA was extracted from samples taken at 0 h and 24 h, and reverse-transcription qPCR was performed to quantify the relative expression levels of denitrification-related genes (*nirS* and *nirK*) and N_2O reductase genes (*nosZI* and *nosZII*), with the 16S rRNA gene serving as an internal reference for normalization. Results showed that the addition of HS significantly enhanced the relative expression of these functional genes, particularly in L-MPD samples (Fig. 4). Combined with denitrification activity measurements, these findings suggest that HS effectively promotes denitrification activity.

[Lines 193-228] “2.5 HS addition enhanced EET and complete denitrification processes in sediment

To confirm whether HS enhance denitrification performance and reduce N₂O emissions by increasing EET capacity, we supplemented sediment samples with additional HS and monitored MPD, denitrification rate, N₂O accumulated level, and functional gene expression level (Fig. 4). Compared to the blank group (no additional HS), the HS-added group (HS_C) showed significantly elevated MPD ($p = 0.0025$) (Fig. 4a). Further analysis using ¹⁵N isotope labeling confirmed that denitrification (71.1±31.1%) was the dominant pathway for N₂ production, with HS addition enhanced denitrification rates by 2.6 times ($p = 0.0028$, Fig. 4b). These results align with prior studies showing that electron shuttles like HA accelerate electron transfer and boost denitrification performance in wastewater treatment systems^{48, 49, 50, 51}. Importantly, after 24 h incubation, HS addition reduced accumulated N₂O concentration by 78.3% compared to the Blank group ($p = 0.0156$). Similar effects have been observed with FA, which suppressed N₂O generation by 41.3%~98.8% in *Paracoccus denitrificans*-mediated denitrification⁵². Together, these findings demonstrate that natural electron shuttles such as HS significantly enhance *in situ* denitrification in lake sediments.

Bacterial biomass exhibited no significant difference during the 24-h incubation (Supplementary Fig. S10). To characterize active microbial processes, we therefore further analyzed the effects of HS addition on functional gene expression (Fig. 4d-g). At the initial time point (0 h), relative expression levels of these functional genes showed no significant differences (Supplementary Table S8). By 24 h, however, the ratios for all functional gene expression levels (HS_C/Blank) were notably higher than 1 (ranging from 4.6- to 396.1-fold), indicating that HS addition broadly enhanced gene expression associated with denitrification, N₂O reduction, and EET (Fig. 4d-e). Notably, the high ratio of N₂O-reducing genes (*nosZI+nosZII*) to denitrifying genes (*nirS+nirK*), quantified as *nos/nir* (Fig. 4d), highlighted the preferential upregulation of N₂O reduction pathways. These findings align with prior studies showing that FA upregulates denitrification genes, particularly *nosZ*, in the model denitrifier *Paracoccus denitrificans*⁵². Our results demonstrate that HS-enhanced EET preferentially improves N₂O reduction activity, thereby accelerating denitrification rates while reducing N₂O production.”

[Lines 411-413] “The production rates of ²⁹N₂ and ³⁰N₂ were calculated based on the slope of linear regression (Supplementary Fig. S16-S17).”

Fig. 4. Effects of humic substance (HS) addition on extracellular electron transfer (EET)-mediated denitrification. **a** Impact of HS addition on maximum power density (MPD) across all groups and the low-MPD (L-MPD) group. **b** Influence of HS addition on denitrification rates and N₂O accumulation levels. **c** Nonlinear regression analysis (one-phase decay exponential model, GraphPad Prism v9.0.0) demonstrating the quantitative relationship between HS concentration and denitrification enhancement ratio (HS-C/Blank). **d** The ratio of relative expression levels of denitrifying genes, N₂O reduction genes, and *nos/nir*, i.e., (*nosZI*+*nosZII*)/(*nirS*+*nirK*), in HS_C groups compared to Blank groups (HS_C/Blank). **e** The ratio of relative expression levels of EET genes (*omcB*, *omcS*, *pilA*) in HS_C groups compared to Blank groups (HS_C/Blank). **f** Log-transformed relative expression levels of denitrifying genes (*nirS* and *nirK*) and N₂O reduction genes (*nosZI*, and *nosZII*). **g** Log-transformed relative expression levels of EET genes (*omcB*, *omcS*, and *pilA*). All gene expression levels were normalized against 16S rRNA levels. Additional statistical information is provided in Supplementary Table S13. Relative expression levels of *nirS* were compared across all samples, while *nirK*, *nosZI*, *nosZII*, *omcB*, *omcS*, and *pilA* were analyzed specifically in the L-MPD subgroup. In violin plots, horizontal bars indicate medians, while dotted lines represent 25th and 75th percentiles.

Supplementary Fig. S10 | Bacterial biomass variation during the 24 h incubation period in blank and HS_C group, respectively. $n = 8$ biological replicates. Blank group, no humic substance addition. HS_C group, commercial humic substance addition group.

Supplementary Figure S16 | Linear regression analysis of dissolved $^{30}\text{N}_2$ concentrations variation ($n = 8$ biological replicates). For LX3, only three time points (0-, 6-, and 12-h) were setup due to insufficient sediment samples. To maximize data retention while ensuring a suitable fitting R^2 ($R^2 > 0.8$), all available data points were included in the regression analysis, excluding the 24-h's data in WX_3, LX_4, and LX_1. Each data point and error bar represents the mean and standard deviation, respectively ($n = 3$ technical replicates).

[Comment 2-24] L279: *If you used 15N-nitrate, I don't understand how you can calculate annamox, which uses ammonia instead of nitrate.*

[Response] This study followed the ¹⁵N-nitrate method described by Thamdrup & Dalsgaard (2002). Given that nitrite (NO₂⁻) in strictly anoxic sediments primarily originates from nitrate (NO₃⁻), the study assumed that N₂ produced through anaerobic ammonium oxidation (anammox) consists of one nitrogen atom from NO_x⁻ and one from NH₄⁺. The detailed calculation process is provided in Supplementary Methods S4.

[Lines 411-414] “The production rates of ²⁹N₂ and ³⁰N₂ were calculated based on the slope of linear regression (Supplementary Fig. S16-S17). The N₂ production rates via denitrification and anammox were calculated based on previously reported ⁷¹ (See calculation methods in Supplementary methods S4).”

“S4. Calculation of denitrification and anammox rates

The denitrification rate (D_{total}) and anammox rate (A_{total}) were determined using the ¹⁵N-NO₃⁻ incubation method previously described with some modification ¹. Specifically, the primary nitrogen species in the ¹⁵N-labeled incubation bottles were ¹⁴N-NH₄⁺, ¹⁴N-NO_x⁻, and ¹⁵N-NO_x⁻. The initial molar fraction (mol/mol) of ¹⁵N in NO_x⁻ was calculated using Equation (2):

$$F_N = n(^{15}\text{N-NO}_x^-) / [n(^{15}\text{N-NO}_x^-) + n(^{14}\text{N-NO}_x^-)] \quad (4)$$

For denitrification (D), assuming that NO_x⁻ is randomly utilized by denitrifiers, the isotopic composition of N₂ produced during denitrification is as follows:

Therefore, the production rates of ²⁸N₂ (D_{28}), ²⁹N₂ (D_{29}), and ³⁰N₂ (D_{30}) from denitrification are:

$$D_{28} = D_{\text{total}} \times (1 - F_N) \times (1 - F_N) \quad (9)$$

$$D_{29} = D_{\text{total}} \times (1 - F_N) \times F_N + D_{\text{total}} \times F_N \times (1 - F_N) \quad (10)$$

$$D_{30} = D_{\text{total}} \times F_N \times F_N \quad (11)$$

For anammox (A), the isotopic composition of N₂ produced is:

Thus, the production rates of ²⁸N₂ (A_{28}) and ²⁹N₂ (A_{29}) from anammox are:

$$A_{28} = A_{\text{total}} \times (1 - F_N) \quad (14)$$

$$A_{29} = A_{\text{total}} \times F_N \quad (15)$$

Based on the above derivation, only denitrification can produce ³⁰N₂. The denitrification rate (D_{total}) can be derived from Equation (9) as:

$$D_{\text{total}} = D_{30} / (F_N \times F_N) \quad (16)$$

$$D_{30} = ^{30}\text{N}_2 \quad (17)$$

Therefore,

$$D_{\text{total}} = ^{30}\text{N}_2 / (F_N \times F_N) \quad (18)$$

Further, the anammox rate (A_{total}) can be derived from Equation (13) as:

$$A_{\text{total}} = A_{29} / F_N \quad (19)$$

$$A_{29} = ^{29}\text{N}_2 - D_{29} \quad (20)$$

From Equation (8), we can derive:

$$D_{29} = D_{\text{total}} \times 2 \times (1-F_N) \times F_N \quad (21)$$

Combining Equations (15) - (18), we obtain:

$$A_{\text{total}} = \{^{29}\text{N}_2 - [^{30}\text{N}_2 \times 2 \times (1-F_N)]/F_N\}/F_N \quad (22)$$

Finally, based on $^{30}\text{N}_2$ and $^{29}\text{N}_2$ production rates (Supplementary Fig. S16-S17) and F_N in the culturing vials, the denitrification rate and anammox rate can be calculated using Equations (16) and (20), respectively.”

Supplementary References

1. Thamdrup B, Dalsgaard T. Production of N_2 through anaerobic ammonium oxidation coupled to nitrate reduction in marine sediments. *Appl. Environ. Microb.* **68**, 1312-1318 (2002).

[Comment 2-25] L303/304: *The Cair N2O concentrations might differ. Please specify which values you used for correction and how or if you measured site specific Cair N2O concentrations (highly recommended).*

[Response] As mentioned in the manuscript **[Lines 362-363]**, “Ambient air samples for atmosphere N_2O (C_{air}) measurement were collected in triplicate at each sampling site.”. We have measured site-specific C_{air} N_2O concentrations. To further clarity, we have revised the annotation of C_{air} in Formula (1) to specify the values used for correction.

[Lines 370-371] “...; C_{air} is the site specific ambient air N_2O concentration;”

[Comment 2-26] L328: *Why did you measure physicochemical values in 30cm , when you otherwise sampled from 10 cm below water surface. I assume that there is a gradient of nutrients across the water column.*

[Response] The 5L water sampler used in our study has a height of 33 cm and was submerged 10 cm below the water surface, resulting in an average sampling depth of approximately 30 cm. To avoid ambiguity, we have revised the description of water sample collection and provided more detailed information about the sampling equipment.

[Lines 326-328] “At each site, triplicate water samples were collected at an average sampling depth of approximately 30 cm below the surface using a 5 L water sampler (Wuhan Xuan Ming Yu Environmental Technology Development Co., Ltd., Wuhan, China).”

[Comment 2-27] L345: *Please also provide information about qPCR standard source, qPCR performance, quality check of amplicon size etc.*

[Response] Thank you for your suggestion. We have supplemented the information regarding the qPCR standard source, qPCR performance (including amplification efficiency and R^2 value), and quality check of amplicon size via agarose gel electrophoresis in Supplementary Methods S5 and Supplementary Fig. S19.

[Lines 417-418] “The primers used for real-time qPCR are listed in Supplementary Table S11, and further details are provided in Supplementary Methods S5.”

[Lines 431-433] “Primers for real-times qPCR are listed in Supplementary Table S11, and detailed protocols can be found in Supplementary Methods S5.”

“S5. Functional genes Real-time qPCR

The qPCR standards for *nirS*, *nirK*, *nosZI*, *nosZII*, and 16S rRNA were prepared using environmental DNA extracted from Taihu Lake sediment⁸. Specifically, PCR amplification was performed with premix TaqTM (Ex TaqTM Version 2.0, Takara, Bio, Shiga, Japan) with primers listed in Supplementary Table S11. The target DNA was purified using the AxyPrep DNA Gel Extraction Kit (AP-GX-50, Axygen Biosciences, CA, USA) according to the manufacturer’s instructions. The purified DNA fragments were cloned into the pMDTM 18-T Vector (Takara Bio, Shiga, Japan) and transformed into *E. coli* JM109 competent cells (Takara Bio, Shiga, Japan). Positive clones were selected on LB plates containing Amp antibiotic, and plasmid DNA was extracted and verified by sequencing.

For extracellular electron transfer (EET) genes (*omcB*, *omcS*, and *pilA*), the full-length gene sequences were obtained from the NCBI database by searching for the whole-genome sequences of model bacteria involved in EET. The primers provided in the references (Supplementary Table S11) and their complementary base pairing were used to obtain the target gene sequences, retaining 10 nucleotides upstream and downstream of the target gene sequences from the whole-genome sequences. The target genes were synthesized and cloned into plasmid bacteria using pUC118 vector (Takara Bio, Shiga, Japan) and TOP10 competent cells (Thermo Fisher Scientific, MA, USA) by Wuhan Icongene Gene Technology Co., Ltd. The specific sequences are as follows:

omcB 212 bp

>dbj|AP028967.1|:208725-209026 *Geobacter* sp. 60473 DNA, complete genome

ATCGCCACGACCCACTTCGACA ACTATTCGACCGGTCCCCAGGCCGGTGCCGGCGCTGG
CGGCACCAACGCCAAGGTTGAAGGCTACGTCCTCCGCGGTACCGGCGCCAACCCCTGCT
TCGACTGCCACGGCCACGAGGGCGAAGACCAATACCCGTCCGGGTCGTGATGCCACGATC
CACACTGACTGGGCCAAGTCCGCCACGCCGGTGGCCTGCTGACCGCCAAGTACA

omcS 133 bp

>gb|CP072789.1|:2747491-2747792 *Geobacter sulfurreducens* strain PL chromosome, complete genome

GGCCGGTACCTGGTTGGCGAAGGCATAGGAGCCGCTCAGGGACTTGGGCTGGTAACCGG
CACCACCAAGGATACGGTATGCACCAACGGCGCCCCAGGCAGTCGGATCGGCGCTGGTG
CTGTAGGAACCGCTGTTCTTGATGGGGAGACCGGT

pilA 116 bp

>dbj|AP028967.1|:1031228-1031378 *Geobacter* sp. 60473 DNA, complete genome

CGTTGCGATCATCGGTATTCTCGCTGCAATTGCGATTCCGCAGTTCTCGGCCGTATCGTGTC
AAGGCGTACAACAGCGCGGCGTCAAGCGACTTGAGAAACCTGAAGACTGCTCTTGAGT
CCGCATTGCTGATGAT

The concentration of the plasmid DNA was measured using a NanoDrop spectrophotometer (Thermo Fisher Scientific, DE, USA), and copy numbers were calculated based on plasmid size. Standard curves were generated by creating a series of concentration gradients through stepwise dilution of the plasmid DNA.

The qPCR reaction was performed in a 20- μ L mixture comprising 10.0 μ L SYBR Premix Ex Taq (Takara Bio, Shiga, Japan), 0.5 μ M of each primer, and 1 μ L of DNA template. qPCR assays were conducted in triplicate using a CFX96 Optical Real-Time Detection System (Bio-Rad, Laboratories

Inc., Hercules, CA, USA). Annealing temperatures followed those previously described (Supplementary Table S11). PCR conditions were: initial denaturation at 94 °C for 5 min; followed by 40 cycles of denaturation at 94 °C for 30 s, annealing at a specific temperature for 30 s (except for *nosZ I* at 15 s and *nosZ II* at 60 s), extension at 72 °C for 30 s, and final extension at 72 °C for 5 min. All standard curves exhibited an R^2 value greater than 0.98. The amplification efficiency ranged from 90.2% to 110.9%. Melt curve analysis was performed for each gene's amplification products to ensure specificity. Furthermore, the size of each amplification product, including those from the blank extractions and negative controls, was verified by agarose gel electrophoresis to ensure consistency with the expected size (Supplementary Fig. S19).”

Supplementary Fig. S19 | Agarose gel electrophoresis analysis of qPCR amplicons. Lanes 1, 2, and 3 represent the negative control (Sterile water used in place of nucleic acid sample), the blank extraction (no sediment added during nucleic acid extraction), and the experimental sample, respectively.

[Comment 2-28] Section 3.7 Please provide more details about the PCR protocol and library preparation kit. Also I miss important quality controls like blank extraction and negative PCR controls processed along the sediment samples. This is especially important for low biomass samples.

[Response] We have now provided detailed information on the PCR protocol and library preparation kit used for 16S rRNA high-throughput sequencing (Supplementary Methods S6). We have also conducted quality controls, i.e., blank extraction and negative PCR control, with relevant descriptions added to the sections on 16S rRNA high-throughput sequencing, qPCR quantification, and RT-qPCR quantification. The size of each amplification product, including those from the blank extractions and negative controls, is provided in Supplementary Fig. S19.

[Lines 435-437] “Blank extraction was also conducted alongside the samples to monitor potential contamination during the extraction process. Details for the PCR protocol and library construction are provided in Supplementary Methods S6.”

“S6. Library construction, Illumina Miseq sequencing, quality control assembly, amplicon sequence variants (ASVs) clustering, and taxonomy annotation

After genomic DNA extraction from the samples, the conserved region of DNA was amplified using specific primers (341F/806R) with barcodes. To ensure the accuracy and reliability of the results, blank extraction and negative PCR controls were processed alongside the sediment samples to monitor potential contamination during the extraction and amplification steps. The PCR reaction was set up in a total volume of 50 μ L, containing 15 μ L of Phusion[®] High-Fidelity PCR Master Mix (New England Biolabs, Ipswich, MA, USA), 0.2 μ M primers, 10 ng of genomic DNA template, and nuclease-free water to make up the volume. The PCR was performed under the following conditions: initial denaturation at 98 $^{\circ}$ C for 1 min, followed by 30 cycles of denaturation at 98 $^{\circ}$ C for 10 s, annealing at 50 $^{\circ}$ C for 30 s, and extension at 72 $^{\circ}$ C for 30 s, with a final extension at 72 $^{\circ}$ C for 5 min. The resulting

PCR products were gel-purified and quantified using a QuantiFluor™ fluorometer. The purified amplicons were then pooled equimolarly to construct the sequencing library using the NEBNext® Ultra™ II DNA Library Prep Kit (New England Biolabs, Ipswich, Massachusetts, USA) according to the manufacturer’s instructions. Paired-end sequencing (2×250) were performed on an Illumina Miseq platform by Guangzhou Genedenovo Biotechnology Co., Ltd (Guangzhou, China) following standard protocols. Among all sediment samples, 11 were successfully sequenced, with WY_1 failing to generate a sequencing library.”

[Supplementary Methods S5] “Furthermore, the size of each amplification product, including those from the blank extractions and negative controls, was verified by agarose gel electrophoresis to ensure consistency with the expected size (Supplementary Fig. S19).”

[Comment 2-29] *Section 3.8. How did you prepare the metagenomic library, which kits, which input, how sequenced (also with MiSeq)?*

[Response] Additional methodological details have been incorporated into Section 3.8. Metagenomic sequencing was performed on the DNBSEQ-T7 platform (MGI, Shenzhen, China).

[Lines 449-456] “Metagenomic sequencing was performed on representative samples DX1, LX3, and LX1, which correspond to L-MPD, M-MPD, and H-MPD, respectively (Supplementary Table S1). The concentration and integrity of DNA in these samples were assessed using a Qubit fluorometer (Thermo Fisher Scientific, Waltham, MA, USA) with the Qubit dsDNA HS Assay Kit, and by agarose gel electrophoresis. DNA libraries were then constructed using the MGIEasy Fast DNA Library Prep Kit V2.0 (MGI, Shenzhen, China) according to the manufacturer’s instructions, with an input DNA amount of 200 ng. Finally, the metagenomic sequencing was carried out using the DNBSEQ-T7 platform (MGI, Shenzhen, China) to generate 2×150 bp paired-end reads.”

[Comment 2-30] *L374: Add the database you used.*

[Response] The NCBI database was used and added as suggested.

[Lines 502-504] All 16S rRNA gene high-throughput sequencing and metagenomic sequencing data have been deposited in the NCBI database under the accession number SRP550967 for amplicon sequencing data and SRP551189 for metagenome data.

[Comment 2-31] *L375: Couldn’t find the metagenomic data in NCBI, please upload and release before publication and at least provide reviewer link meanwhile.*

[Response] Thank you for your comment. The metagenomic sequencing data has been deposited in the NCBI database under the accession number SRP551189. We have provided the reviewer link below for access to the data prior to publication. We will ensure that the data is publicly released upon publication of our manuscript.

Reviewer Link:

<https://dataview.ncbi.nlm.nih.gov/object/PRJNA1197547?reviewer=ld984qp2cqslsj59ni1pi3117ls>

[Comment 2-32] Section 3.10: Regression analysis are not mentioned as well as PCA (which distance matrix did you use). I am also not sure how exactly the replication of the study worked. Did you samples three times per site, then site needs to be random factor or did you have only 1 sample per site, then a t-test or similar is not suitable at all.

[Response] We appreciate your comment. The methods for regression analysis and PCA (standardized covariance matrix) have been provided. For water quality, dissolved N₂ and N₂O concentrations, we collected three replicate samples at each sampling point for measurement. For sediment samples, we collected one sample per site. To meet the statistical requirement for sample independence, we have re-analyzed the samples accordingly. We now use the mean values of replicates in each site for statistical analysis. We removed the previous Supplementary Fig. S3 and replaced it with Supplementary Table S7, which shows different groups with varying MPD, dissolved N₂ and ΔN₂ concentrations. For the indoor experiments, we designed paired experiments and selected the subsequent paired analysis methods based on whether the residuals met the normality assumptions. We have added explanations for the data in the legends of figures and tables. The statistical analysis in the supplementary materials has also been revised accordingly.

[Lines 475-499] “The mean values of three technical replicates at each of the 12 sampling sites were used for statistical analysis ($n = 12$ biological replicates). We quantified key environmental factors influencing *in situ* denitrification potential (dissolved N₂ and ΔN₂ concentration) through sequential statistical analyses. Spearman correlation identified factors significantly associated with denitrification potential, followed by simple linear regression. Variables meeting residual assumptions (homoscedasticity, normality) after transformation were used in stepwise multiple linear regression (bidirectional Akaike Information Criterion optimization). The optimal model was validated by screening multicollinearity (variance inflation factor < 5), re-testing residuals, and assessing predictor significance. Relative influence of factors was determined using standardized coefficients, nested model comparisons (ANOVA), and visualization (*‘MASS’*, *‘lmtest’*, *‘car’*, and *‘ggplot2’* packages in R software, version 4.0.4). Detailed code analytical procedures are provided in Supplementary Methods S7. Linear and exponential regression analyses were performed using GraphPad Prism (version 9.0.0). To test the effects of humic substance addition on MPD, denitrification rates, N₂O accumulation levels, microbial biomass, and relative expression levels of functional genes, we conducted paired analyses ($n = 8$ biological replicates). First, the Shapiro-Wilk test was used to examine the normality of the residuals of the paired samples. If the data met the normality assumptions, a paired *t*-test was applied; otherwise, the Wilcoxon matched-pairs signed rank test was used (GraphPad Prism) (Supplementary Table S13). For the analysis of microbial community based on 16S rRNA high-throughput sequencing, the *‘microeco’* package within the R software environment (version 4.0.4) was employed. Initially, “mitochondria” and “chloroplasts” were removed from the dataset. All samples were then resampled to equalize the sequence number at the minimum value observed among the samples. This resampling procedure was performed to facilitate a more precise evaluation of α-diversity indexes. Spearman correlation analysis, principal component analysis (standardized covariance matrix), and heatmap were carried out using the *‘Hmisc’*, *‘corrplot’*, *‘vegan’*, *‘ggplot2’*, and *‘pheatmap’* packages in R software environment (version 4.0.4).”

Results/Discussion:

[Comment 2-33] L98: Are these values comparable to other studies?

[Response] We have added comparisons with previous studies.

[Lines 100-107] “Consequently, the electron transfer capacity of *in situ* sediments was assessed using maximum power density (MPD) measured by microbial fuel cells (MFC) (Supplementary Fig. S2)²³. The MPD values spanned a wide range (0.05 ~ 76.65 $\mu\text{W m}^{-2}$), comparable to those measured in aquaculture pond sediment MFC (52.2 ~ 99.5 $\mu\text{W m}^{-2}$)²⁴ and plant MFC (0.28 ~ 37.0 $\mu\text{W m}^{-2}$)^{25,26}. However, these values were significantly lower than those reported for lake Taihu sediment MFC (3150 ~ 5700 $\mu\text{W m}^{-2}$)^{27,28,29}, which were attribute to the formation of mature biofilm on anode surfaces after 15 to 60 days of cultivation in prior studies. Thus, the MPD measured in the present study (within 1 h after sediment addition) better reflects *in situ* characteristics.”

[Comment 2-34] Fig. 2: Panel b,c are showing redundant content as panel a. I suggest to put into supplement. Panel d, e: I still don't understand if that was done for all 12 sites or only for 1. It would be very interesting to see the relationship between the treatments and inherent HS concentrations, does that weaken in high HS sites?

[Response] Thank you for your valuable comments! Following the reviewer's suggestions, we have moved panels b-c of Fig. 2 to Supplementary Fig. S5. In response to the previous review comments, we re-performed the ¹⁵N incubation experiments (*n* = 8 biological replicates in total, including three L-MPD samples, all M-MPD samples, and all H-MPD samples) to compare the denitrification rates, N₂O accumulation levels, and functional gene expression between the blank group and the corresponding humic substances (HS) addition groups using a paired experimental design. Notably, the effects of HS addition weaken in high MPD sites (Fig. 4c). Finally, we moved Fig. 2a to Fig. 1d to present all field data in Fig. 1 and all laboratory results with humic substances addition in Fig. 4.

Supplementary Fig. S5 | Linear regression analyses of key environmental factors with N₂ gas emission characteristics. **a** Linear regression analyses of log(NO₃⁻-N_S+1), log(NH₄⁺-N_S), maximum power density (MPD), and humic acid (HA) concentrations with dissolved N₂ concentration. **b** Linear regression analyses of 10^{-pH} in water, log(NO₃⁻-N_S+1), log(NH₄⁺-N_S), MPD, and HA concentrations with ΔN₂ concentration. *n* = 12 biological replicates. Light gray shading bordered in red represents the 95% confidence intervals for these regressions. The key physicochemical factors were identified according to the significant level (*p* < 0.05) in Spearman correlation analysis (Supplementary Fig. S3). Only linear regression with a *p*-value < 0.05 were shown.

Fig. 1 | Sampling sites, characteristics of electron shuttles, and *in situ* nitrogen removal performance. **a** Distribution of sampling sites (red stars, $n = 12$ biological replicates). The pie charts indicate the concentrations and proportions of dissolved electron shuttles, i.e., humic acid (HA) and fulvic acid (FA). The size of each pie chart is proportional to the concentration of humic substances (HS), while the color represents the excess N_2 (ΔN_2) concentration at the respective sites. **b** Linear regression analyses depicting the relationships between the concentrations of HS (HA + FA) and the electron transfer capacity, represented by maximum power density (MPD). Light gray shading with red borders indicates 95% confidence intervals. **c** Standardized coefficients of environmental factors derived from multiple linear regression model predicting dissolved N_2 concentration and ΔN_2 concentration. **d** Spearman correlations between the MPD of fresh sediments and the *in situ* emission characteristics of N_2 and N_2O . Blue and red circles denote positive and negative correlations, respectively, with color saturation representing correlation strength. $n = 12$ biological replicates. * and ** indicate p value < 0.05 and < 0.01 , respectively.

Fig. 4. Effects of humic substance (HS) addition on extracellular electron transfer (EET)-mediated denitrification. **a** Impact of HS addition on maximum power density (MPD) across all groups and the low-MPD (L-MPD) group. **b** Influence of HS addition on denitrification rates and N₂O accumulation levels. **c** Nonlinear regression analysis (one-phase decay exponential model, GraphPad Prism v9.0.0) demonstrating the quantitative relationship between HS concentration and denitrification enhancement ratio (HS-C/Blank). **d** The ratio of relative expression levels of denitrifying genes, N₂O reduction genes, and *nos/nir*, i.e., (*nosZI*+*nosZII*)/(*nirS*+*nirK*), in HS_C groups compared to Blank groups (HS_C/Blank). **e** The ratio of relative expression levels of EET genes (*omcB*, *omcS*, *pilA*) in HS_C groups compared to Blank groups (HS_C/Blank). **f** Log-transformed relative expression levels of denitrifying genes (*nirS* and *nirK*) and N₂O reduction genes (*nosZI*, and *nosZII*). **g** Log-transformed relative expression levels of EET genes (*omcB*, *omcS*, and *pilA*). All gene expression levels were normalized against 16S rRNA levels. Additional statistical information is provided in Supplementary Table S13. Relative expression levels of *nirS* were compared across all samples, while *nirK*, *nosZI*, *nosZII*, *omcB*, *omcS*, and *pilA* were analyzed specifically in the L-MPD subgroup. In violin plots, horizontal bars indicate medians, while dotted lines represent 25th and 75th percentiles.

[Comment 2-35] Fig. 3: What are representative samples, which ones did you exactly use, and how did you decide?

[Response] As addressed in our response to **[Comment 2-3]**, maximum power density (MPD) was identified as the key environmental factor governing dissolved N₂ concentration and ΔN₂ concentration through comprehensive analyses. Based on the resulting data distribution (Supplementary Fig. S5), all samples were categorized into three groups: L-MPD (4.1±6.1 μW m⁻²), M-MPD (44.0±3.5 μW m⁻²), and H-MPD (73.3±3.6 μW m⁻²) (Supplementary Table S7). From these groups, we randomly selected DX_1 (L-MPD), LX_3 (M-MPD), and LX_1 (H-MPD) as representative samples for XPS analysis (Supplementary Table S1).

[Lines 137-140] “Given that MPD significantly affects *in situ* denitrification potential (Fig. 1c), the samples were categorized into three representative groups to analyze the functional group characteristics of HS under varying MPD conditions (L-MPD: 4.1±6.1 μW m⁻², M-MPD: 44.0±3.5 μW m⁻², and H-MPD: 73.3±3.6 μW m⁻²) (Supplementary Table S7).”

[Lines 342-344] “DX_1, LX_3, and LX_1 were randomly selected as representative samples from L-MPD, M-MPD, and H-MPD, respectively (Supplementary Table S1). The components of the extracted representative HS were first analyzed using EEM fluorescence spectroscopy.”

[Fig. 2] “**Fig. 2 | The functional groups of extracted electron shuttles. a,b** X-ray photoelectron spectroscopy (XPS) measured C1s spectrum and atomic% of humic acid (a) and fulvic acid (b) in representative L-MPD, M-MPD, and H-MPD samples. DX_1, LX_3, and LX_1 were randomly selected as representative L-MPD, M-MPD, and H-MPD samples, respectively (Supplementary Table S1). MPD, maximum power density.”

[Fig. 5] “The microbial community structure at species level was analyzed using metagenomic sequencing (DX_1, LX_3, and LX_1).”

[Fig. 6] “**a** Taxonomy and relative abundance of MAGs in representative low-, medium-, and high-MPD samples (DX_1, LX_3, and LX_1).”

[Supplementary Fig. S6] “DX_1, LX_3, and LX_1 were randomly selected as representative L-MPD, M-MPD, and H-MPD samples, respectively (Supplementary Table S1).”

Fig. 1 | Sampling sites, characteristics of electron shuttles, and *in situ* nitrogen removal performance. **a** Distribution of sampling sites (red stars, $n = 12$ biological replicates). The pie charts indicate the concentrations and proportions of dissolved electron shuttles, i.e., humic acid (HA) and fulvic acid (FA). The size of each pie chart is proportional to the concentration of humic substances (HS), while the color represents the excess N_2 (ΔN_2) concentration at the respective sites. **b** Linear regression analyses depicting the relationships between the concentrations of HS (HA + FA) and the electron transfer capacity, represented by maximum power density (MPD). Light gray shading with red borders indicates 95% confidence intervals. **c** Standardized coefficients of environmental factors derived from multiple linear regression model predicting dissolved N_2 concentration and ΔN_2 concentration. **d** Spearman correlations between the MPD of fresh sediments and the *in situ* emission characteristics of N_2 and N_2O . Blue and red circles denote positive and negative correlations, respectively, with color saturation representing correlation strength. $n = 12$ biological replicates. * and ** indicate p value < 0.05 and < 0.01, respectively.

Supplementary Fig. S5 | Linear regression analyses of key environmental factors with N₂ gas emission characteristics. **a** Linear regression analyses of log(NO₃⁻-N_{S+1}), log(NH₄⁺-N_S), maximum power density (MPD), and humic acid (HA) concentrations with dissolved N₂ concentration. **b** Linear regression analyses of 10^{^(-pH)} in water, log(NO₃⁻-N_{S+1}), log(NH₄⁺-N_S), MPD, and HA concentrations with ΔN₂ concentration. *n* = 12 biological replicates. Light gray shading bordered in red represents the 95% confidence intervals for these regressions. The key physicochemical factors were identified according to the significant level (*p* < 0.05) in Spearman correlation analysis (Supplementary Fig. S3). Only linear regression with a *p*-value < 0.05 were shown.

Supplementary Table S7 | In situ denitrification potential across sampling sites categorized by maximum power density (MPD).

Groups	MPD (µW m ⁻²)	Dissolved N ₂ concentration (µmol L ⁻¹)	ΔN ₂ concentration (µmol L ⁻¹)
L-MPD	0.1~15.6	524.0±22.3	5.9±16.2
M-MPD	41.7~42.0	584.2±30.9	65.0±28.1
H-MPD	70.4~76.1	671.1±26.3	151.3±24.6

The detailed sample names for each group are provided in Supplementary Table S1. Abbreviations: L-MPD, Low-MPD; M-MPD, medium-MPD; H-MPD, high-MPD; ΔN₂, excess dissolved N₂.

[Comment 2-36] L172: Throughout the whole text please write 16S rRNA gene.

[Response] We have revised all instances of ‘16S’ to ‘16S rRNA’ throughout the manuscript.

[Comment 2-37] Fig. 5c: Where do the chloroplasts come from, does it make sense to keep them in the dataset?

[Response] We appreciate your insightful comment and agree that chloroplasts require special consideration. In lake sediment samples, these chloroplasts originate from settled algae. Due to their evolutionary origin, chloroplasts possess conserved 16S rRNA regions similar to prokaryotes, leading to potential co-amplification during PCR. Consequently, we removed both “chloroplasts” and “mitochondria” from microbial community data using the “filter_pollutants(taxa = c(“mitochondria”, “chloroplasts”))” function in R’s ‘microeco’ package. To ensure robust diversity analysis, we then performed resampling to standardize sequencing depth (based on minimum sample depth) before

recalculating α -diversity indexes. Finally, we reperformed all downstream analyses including PCA and microbial composition assessments to generate the revised Fig. 5.

[Lines 492-499] “For the analysis of microbial community based on 16S rRNA high-throughput sequencing, the ‘microeco’ package within the R software environment (version 4.0.4) was employed. Initially, “mitochondria” and “chloroplasts” were removed from the dataset. All samples were then resampled to equalize the sequence number at the minimum value observed among the samples. This resampling procedure was performed to facilitate a more precise evaluation of α -diversity indexes. Spearman correlation analysis, principal component analysis (standardized covariance matrix), and heatmap were carried out using the ‘Hmisc’, ‘corrplot’, ‘vegan’, ‘ggplot2’, and ‘pheatmap’ packages in R software environment (version 4.0.4).”

Fig. 5 | Variation of microbial community structure in estuary sediment along with increased MPD. **a** Principal component analysis of the microbial community at genus level. The dots represent microbial community of different sampling sites, and were colored based on maximum power density (MPD) and electron shuttles concentration, respectively. **b-d** The variation of dominant microbial community structure along with MPD at phylum (>1% in at least one sample), genus (>2% in at least one sample), and species (>0.5% in at least one sample) levels. The microbial community structure at phylum and genus levels were analyzed using 16S high-throughput sequencing. The microbial community structure at species level was analyzed using metagenomic sequencing (DX_1, LX_3, and LX_1). Abbreviations: HA, humic acid; FA, fulvic acid.

L185: Especially for *Quatrionococcus* it was shown before that nitrite is a likely endproduct of denitrification. How did nitrite concentration correlate to your data in general? Did you measure it in the in situ and incubation samples? It would be quite important to see whether there is an accumulation of nitrite across your MPD/HS gradient. Also your MAGs indicate that there might be a potential risk of nitrite accumulation (DX76).

[Response] Nitrite concentration in lake water and sediment was measured. Supplementary measurements of nitrite in incubation samples were conducted. Spearman correlation analysis between nitrite and denitrification potential (dissolved N_2 and ΔN_2 concentrations) is presented in Supplementary Fig. S4. The results indicate that nitrite was not a key factor affecting denitrification potential. However, humic substances (HS) addition significantly increased accumulated nitrite concentration in incubation vessels, consistent with the significant positive linear correlation observed between sediment nitrite concentration and HS/MPD concentrations (Supplementary Fig. S13). Therefore, enhanced extracellular electron transfer (EET)-coupled denitrification in sediment carries a demonstrated risk of toxic nitrite accumulation. We have supplemented the relevant content.

[Lines 271-280] “Nineteen of the forty-one MAGs were associated with denitrification (Fig. 6). However, each MAG only contained genes mediating one or two reduction steps (Fig. 6)⁶⁰. Consequently, nitrite likely becomes the terminal denitrification product in specific MAGs and the denitrifier *Quatrionococcus australiensis* due to the absence of nitrite reduction genes⁵⁶ (Fig. 5-6). Supporting this, sediment nitrite concentration showed a significant positive correlation with both MPD and humic substance concentration (Supplementary Fig. S13). Furthermore, HS addition significantly increased nitrite accumulation in L-MPD sediments (Supplementary Fig. S13d). Consequently, the EET-coupled denitrification process might enhance the potential risk of toxic nitrite accumulation in sediment. This risk should be carefully considered during ecological monitoring and application of EET for enhancing environmental bioremediation.”

Supplementary Fig. S13 | Effects of maximum power density (MPD) and humic substances on

nitrite concentration in sediment. a Linear regression between sediment nitrite concentration and MPD. **b** Linear regression between sediment nitrite concentration and HA+FA concentration. **c** Linear regression between sediment nitrite concentration and HA concentration. **d** Temporal variation of nitrite concentration in supernatant during 24-h incubation for Blank (no humic substances addition) and HS_C (commercial humic substances added) groups. * $p < 0.05$. $n = 3$ biological replicates for L-MPD samples (WX_2, WX_3, WY_1).

[Comment 2-38] L191: Please give threshold values for medium and high quality MAGs.

[Response] We have defined the threshold values for medium- and high-quality MAGs as follows: $\geq 50\%$ completeness with $< 10\%$ contamination for medium-quality, and $> 90\%$ completeness with $< 5\%$ contamination for high-quality, in accordance with the standards established by Bowers et al. (2017).

[Lines 248-250] “This study identified 41 medium- (completion $\geq 50\%$, contamination $< 10\%$) and high-quality (completion $> 90\%$, contamination $< 5\%$) metagenome-assembled genomes (MAGs) (Supplementary Data 1), some of which exhibited complete EET capacity (Fig. 6).”

Bowers R, et al. Minimum information about a single amplified genome (MISAG) and a metagenome-assembled genome (MIMAG) of bacteria and archaea. Nat. Biotechnol. 35, 725-731 (2017).

[Comment 2-39] L215: I don't think you can say that sequential denitrification is the general mode. This strongly depends on the strain, environmental conditions and can be even different for closely related species. To proof this you need more physiological testing of isolates of your system, or at least some transcription analysis over a timer series.

[Response] We agree with your comments and have revised the sentences accordingly.

[Lines 272-273] “Nineteen of the forty-one MAGs were associated with denitrification (Fig. 6). However, each MAG only contained genes mediating one or two reduction steps (Fig. 6)⁶⁰.”

[Comment 2-40] L231: I am not sure if you can talk about efficiency, that would require some kind of normalization to be able to compare across the different sites, e.g. N₂ production per mol nitrate or so.

[Response] We have revised “denitrification efficiency” to “denitrification rates”.

[Lines 296-297] “Altogether, EET processes significantly contribute to completing the denitrification process, leading to high denitrification rates and reduced N₂O emissions.”

Reviewer #2 (Remarks on code availability):

[Comment 2-41] I partly reviewed to code, which is not too helpful. The read.me did not contain a list of packages to be installed. Regarding the 16S code there is even a difference between what the code states and what the manuscript states for the truncation command.

--p-trunc-len-f 0 \ (in code)

--p-trunc-len-f 240 (in manuscript)

[Response] We sincerely apologize for this oversight. We have now added the packages list, main procedures for analysis, and precautions for both the 16S analysis project and the metagenomic analysis project in the README.txt file. We have added a new R script for multiple linear regression analysis to evaluate the contributions of environmental factors to denitrification potentials. Furthermore, detailed comments have been incorporated throughout all analytical code sections to enhance clarity. Finally, the code in Supplementary Methods S6 has been revised.

[Supplementary Methods S6] “Based on this high quality, the QIIME 2 plugin “dada2 denoise-paired” was applied with the following parameters: `--p-trim-left-f 0 --p-trim-left-r 0 --p-trunc-len-f 0 --p-trunc-len-r 06`.”

[Comment 2-42] *It would be helpful to see the error plots etc on which the truncation decision was made. As it stand now it can't be judged if that was correctly done.*

[Response] We have added the quality plot of forward reads and reverse reads (Supplementary Fig. S12) and revised Supplementary Methods S6.

[Supplementary Methods S6] “Except for the 6th base in forward reads and the 213th base in reverse reads, which showed lower quality, the 25th percentile of the quality scores for all bases reached a high value of 37 (Supplementary Fig. S20). This indicates that sequencing quality across most regions of both forward and reverse reads was consistently high, concentrated between 35 and 40, thereby meeting the requirements for subsequent analysis. Based on this high quality, the QIIME 2 plugin “dada2 denoise-paired” was applied with the following parameters: `--p-trim-left-f 0 --p-trim-left-r 0 --p-trunc-len-f 0 --p-trunc-len-r 06`. Further details on the analysis are available in the GitHub repository (<https://github.com/dengcode11/Electron-shuttling-derived-denitrification/>).”

Supplementary Fig. S20. Quality Plot of forward reads and reverse reads. **a** The quality score of sequence base in forward reads. **b** The quality score of sequence base in reverse reads. Box plots were generated using a random sampling of 10,000 sequences out of 1,221,321, without replacement. Except for the 6th base in forward reads and the 213th base in reverse reads, the 25th percentile of the quality scores for all bases is as high as 37. Given this high quality, all bases were retained for subsequent analyses.

[Comment 2-43] *For the community to learn from it, it would very helpful to add an explanation what each single step does, especially for the metagenomic code.*

[Response] All the code has been thoroughly commented.